# REPRESENTATION CONVERGENCE: MUTUAL DISTILLATION IS SECRETLY A FORM OF REGULARIZATION

## ABSTRACT

In this paper, we argue that mutual distillation between reinforcement learning policies serves as an *implicit regularization*, preventing them from overfitting to irrelevant features. We highlight two *separate* contributions: (i) Theoretically, for the first time, we provide an *end-to-end* theoretical proof that enhancing the policy robustness to irrelevant features leads to improved generalization performance. (ii) Empirically, we demonstrate that mutual distillation between policies contributes to such robustness, enabling the spontaneous emergence of *invariant representations* over pixel inputs. Ultimately, we do not claim to achieve state-of-the-art performance but rather focus on uncovering the underlying principles of generalization and deepening our understanding of its mechanisms. Our website: https://dml-rl.github.io/.

## 1 INTRODUCTION

Humans exhibit a remarkable ability to learn robustly and generalize across diverse environments. Once a skill is acquired, it often transfers seamlessly to new contexts that share the same underlying semantics, even when their visual appearance differs substantially. For example, consider a person who becomes proficient at a video game, even if the background graphics or character textures are altered, the player retains their ability to perform well, effortlessly adapting to the new setting. This suggests that human learning is not overly dependent on low-level visual details, but rather grounded in abstract representations that capture the essential structure of a task. Neuroscientific studies support this view, linking abstract reasoning to the human prefrontal cortex (Bengtsson et al., 2009; Dumontheil, 2014), and highlighting the role of inhibitory neurons in enhancing cognitive processing efficiency (Pi et al., 2013).

In stark contrast, visual reinforcement learning (VRL) agents often struggle with generalization. While they can be trained to solve complex tasks in specific environments, even minor changes, such as shifts in color schemes or background textures, can significantly degrade their performance. This sensitivity indicates that VRL agents tend to overfit to superficial visual features, failing to capture the underlying structure of the task (Cobbe et al., 2019; 2020). These limitations give rise to a fundamental question:

*What hinders reinforcement learning agents from generalizing like humans? How can we enable them to learn robust representations that drive human-like generalization behavior?*

The core reason behind the limited generalization ability of VRL agents lies in their reliance on convolutional neural networks (CNNs) as visual encoders. While CNNs are the de facto choice for processing high-dimensional visual inputs, they are notoriously sensitive to even small perturbations (Goodfellow et al., 2014). This brittleness significantly hampers the robustness of learned policies and limits their ability to generalize. To address this issue, one common strategy is to apply data augmentation (Shorten & Khoshgoftaar, 2019), which improves robustness by diversifying the training distribution and reducing dataset-induced biases. Alternatively, invariant representation learning has emerged as a principled approach to tackle generalization problem from a feature-learning perspective. It aims to extract representations that remain stable under a wide range of input transformations, thereby promoting robustness and transferability (Nguyen et al., 2021).

While data augmentation is an effective bias mitigation technique, its reliance on task-specific strategies that are manually crafted by human experts, poses a challenge for designing task-independent

solutions. In contrast, our method enables agents to generalize without any handcrafted augmentations or external priors, relying purely on training experience. Invariant representation learning is a promising approach to enhance model's cross-domain generalization. However, it relies on transformation correspondences, which are fundamentally inaccessible in the generalization scenarios of reinforcement learning due to the dynamic nature of environments. In addition, the invariant representation framework inherently separates the encoder from the model, unnecessarily complicating the theoretical analysis. Instead, our framework is theoretically and empirically end-to-end.

In this paper, we first propose a novel theoretical framework to analyze the generalization problem in reinforcement learning and show that the policy robustness to irrelevant features enhances its generalization performance. Building upon this principled insight, we then provide empirical evidence that deep mutual learning (DML) (Zhang et al., 2018b) can implicitly prevent online RL policies from overfitting to such irrelevant features, leading to a stable learning process and significant generalization improvements.

In summary, the main contributions of this paper are as follows:

- We theoretically prove that improving the policy robustness to irrelevant features enhances its generalization performance. To the best of our knowledge, we are the first to provide a rigorous proof of this intuition.
- We propose a hypothesis that deep mutual learning (DML) enhances the generalization performance of the policy by implicitly regularizing irrelevant features. We also provide intuitive insights to support this hypothesis.
- Strong empirical results support our theory and hypothesis, showing that DML technique leads to consistent improvements in generalization performance.

## 2 RELATED WORK

**The generalization of deep reinforcement learning** has been widely studied, and previous work has pointed out the overfitting problem in deep reinforcement learning (Rajeswaran et al., 2017; Zhang et al., 2018a; Justesen et al., 2018; Packer et al., 2018; Song et al., 2019; Cobbe et al., 2019; Grigsby & Qi, 2020; Cobbe et al., 2020; Yuan et al., 2023; Suau et al., 2023; Kirk et al., 2023). A natural approach to avoid the overfitting problem is to apply regularization techniques originally developed for supervised learning such as dropout (Srivastava et al., 2014; Farebrother et al., 2018; Igl et al., 2019), data augmentation (Laskin et al., 2020; Yarats et al., 2021; Zhang & Guo, 2021; Raileanu et al., 2021; Ma et al., 2022), domain randomization (Tobin et al., 2017; Yue et al., 2019; Slaoui et al., 2019; Lee et al., 2019; Mehta et al., 2020). On the other hand, in order to improve sample efficiency, previous studies encouraged the policy network and value network to share parameters (Schulman et al., 2017; Huang et al., 2022). However, recent works have explored the idea of decoupling the two and proposed additional distillation strategies (Cobbe et al., 2021; Raileanu & Fergus, 2021; Moon et al., 2022). In particular, Raileanu & Fergus (2021) demonstrated that more information is needed to accurately estimate the value function, which can lead to overfitting. Moreover, exploration has also been shown to be an effective technique for improving policy generalization (Jiang et al., 2023; Weltevrede et al., 2024), as the exploration phase effectively alters the initial state distribution and allows the policy to access more diverse trajectories (Weltevrede et al., 2024). In addition, prior works also adopt kernel complexity (Yeh et al., 2023) or causal learning perspectives (Kallus & Zhou, 2020; Suau et al., 2023) as measures of representation capacity.

**Representation learning** is another tool for improving generalization. Prior work has either leveraged bisimulation metrics to capture invariances by comparing states in terms of their reward and transition distributions (Zhang et al., 2020), or adopted self-supervised objectives that align trajectories based on behavioral similarity (Mazoure et al., 2021), which enable the encoder to learn visually robust features without relying on explicit reward signals. However, these methods introduce an additional encoder pretraining stage that is separate from the reinforcement learning process, potentially hindering sample efficiency and leading to suboptimal downstream representations, which can further limit end-to-end adaptability. Moreover, modern policy gradient algorithms such as TRPO (Schulman et al., 2015), PPO (Schulman et al., 2017), and SPO (Xie et al., 2025) typically formulate an end-to-end policy $\pi$, this further motivates us to develop a framework that is both theoretically and empirically end-to-end, while allowing easy integration into the reinforcement learning pipeline.

**Knowledge distillation** is a learning paradigm that aims to align the student network with the teacher network to achieve knowledge transfer. A commonly used practice is to distill the knowledge learned by a large model into a smaller model to reduce inference costs after deployment (Xu et al., 2024). On the other hand, distillation technique can also be used to distill a model with privileged information into a model with access to only partial information to improve its generalization ability. However, research has shown that knowledge distillation can also be applied to multiple student networks during training to encourage them to learn from each other, called deep mutual learning (DML) (Zhang et al., 2018b). Lai et al. (2020) then propose dual policy distillation, a student-student mutual distillation framework that can improve performance without requiring a pre-trained teacher. Building upon this observation, Zhao & Hospedales (2021) further demonstrate that DML can improve the generalization performance of reinforcement learning agents, yet no in-depth analysis of why this happens. In addition, recent studies suggest that aligning the student networks at the output layer may be suboptimal, and recommend alignment at the logits layer instead (Deckers et al., 2024; Vandersmissen et al.). Furthermore, Weltevrede et al. (2025) show that distilling multiple RL policies into an ensemble on diverse training states can significantly improve zero-shot generalization, yet their settings are limited to environments with rotational symmetry. We extend mutual distillation as a form of regularization and propose a more general end-to-end generalization theory.

## 3 PRELIMINARIES

In this section, we introduce reinforcement learning under the generalization setting in Section 3.1, as well as the DML technique in Section 3.2.

### 3.1 MARKOV DECISION PROCESS AND GENERALIZATION

Markov decision process (MDP) is a mathematical framework for sequential decision-making, which is defined by a tuple $\mathcal{M} = (\mathcal{S}, \mathcal{A}, r, \mathcal{P}, \rho, \gamma)$, where $\mathcal{S}$ and $\mathcal{A}$ represent the state space and action space, $r : \mathcal{S} \times \mathcal{A} \mapsto \mathbb{R}$ is the reward function, $\mathcal{P} : \mathcal{S} \times \mathcal{A} \times \mathcal{S} \mapsto [0, 1]$ is the dynamics, $\rho : \mathcal{S} \mapsto [0, 1]$ is the initial state distribution, and $\gamma \in (0, 1)$ is the discount factor.

Define a policy $\mu : \mathcal{S} \times \mathcal{A} \mapsto [0, 1]$, the action-value function and value function are defined as

$$Q^\mu(s_t, a_t) = \mathbb{E}_\mu \left[ \sum_{k=0}^\infty \gamma^k r(s_{t+k}, a_{t+k}) \right], \quad V^\mu(s_t) = \mathbb{E}_{a_t \sim \mu(\cdot|s_t)} \left[ Q^\mu(s_t, a_t) \right]. \tag{1}$$

Given $Q^\mu$ and $V^\mu$, the advantage function can be expressed as $A^\mu(s_t, a_t) = Q^\mu(s_t, a_t) - V^\mu(s_t)$.

In our generalization setting, we introduce a rendering function (Smallwood & Sondik, 1973) $f : \mathcal{S} \mapsto \mathcal{O}_f \subset \mathcal{O}$ to obfuscate the agent's actual observations, which is a *bijection*[1] from $\mathcal{S}$ to $\mathcal{O}_f$. We now define the MDP induced by the underlying MDP $\mathcal{M}$ and the rendering function $f$, denote it as $\mathcal{M}_f = (\mathcal{O}_f, \mathcal{A}, r_f, \mathcal{P}_f, \rho_f, \gamma)$, where $\mathcal{O}_f$ represents the observation space, $r_f : \mathcal{O}_f \times \mathcal{A} \mapsto \mathbb{R}$ is the reward function, $\mathcal{P}_f : \mathcal{O}_f \times \mathcal{A} \times \mathcal{O}_f \mapsto [0, 1]$ is the dynamics, and $\rho_f : \mathcal{O}_f \mapsto [0, 1]$ is the initial observation distribution. We present the following assumptions:

**Assumption 3.1.** Assume that $f$ can be sampled from a distribution $p : \mathcal{F} \mapsto [0, 1]$, where $f \in \mathcal{F}$, which means that $\int_\mathcal{F} p(f) \mathrm{d} f = 1$ is naturally satisfied.

**Assumption 3.2.** Given any $f \in \mathcal{F}$, $o_0^f, o_t^f, o_{t+1}^f \in \mathcal{O}_f$ and $a_t \in \mathcal{A}$, assume that $r_f(o_t^f, a_t) = r(f^{-1}(o_t^f), a_t), \mathcal{P}_f(o_{t+1}^f|o_t^f, a_t) = \mathcal{P}(f^{-1}(o_{t+1}^f)|f^{-1}(o_t^f), a_t), \rho_f(o_0^f) = \rho(f^{-1}(o_0^f))$.

**Explanation.** Assumption 3.2 states that all $\mathcal{M}_f$ share a common underlying MDP $\mathcal{M}$, in which the agent's observations are perturbed by different rendering functions while all other components remain unchanged, much like different painters depicting the same scene in their own styles.

Next, consider an agent interacting with $\mathcal{M}_f$ following the policy $\pi : \mathcal{O} \times \mathcal{A} \mapsto [0, 1]$ to obtain a trajectory

$$\tau_f = (o_0^f, a_0, r_0^f, o_1^f, a_1, r_1^f, \ldots, o_t^f, a_t, r_t^f, \ldots), \tag{2}$$

---

[1]We define $\mathcal{O}_f := \{f(s)|s \in \mathcal{S}\}$, which means for any $s_1 \neq s_2$, we have $f(s_1) \neq f(s_2)$.

where $o_0^f \sim \rho_f(\cdot)$, $a_t \sim \pi(\cdot|o_t^f)$, $r_t^f = r_f(o_t^f, a_t)$ and $o_{t+1}^f \sim \mathcal{P}_f(\cdot|o_t^f, a_t)$, we simplify the notation to $\tau_f \sim \pi$. During training, the agent is only allowed to access a subset of all MDPs, which is $\{\mathcal{M}_f | f \in \mathcal{F}_{\text{train}} \subset \mathcal{F}\}$, and then tests its generalization performance across all MDPs. Thus, denote $p_{\text{train}} : \mathcal{F}_{\text{train}} \mapsto [0, 1]$ as the distribution over $\mathcal{F}_{\text{train}}$, the agent's training performance $\eta(\pi)$ and generalization performance $\zeta(\pi)$ can be expressed as

$$\eta(\pi) = \mathbb{E}_{f \sim p_{\text{train}}(\cdot), \tau_f \sim \pi} \left[ \sum_{t=0}^{\infty} \gamma^t r_f(o_t^f, a_t) \right], \quad \zeta(\pi) = \mathbb{E}_{f \sim p(\cdot), \tau_f \sim \pi} \left[ \sum_{t=0}^{\infty} \gamma^t r_f(o_t^f, a_t) \right]. \quad (3)$$

The goal of the agent is to learn a policy $\pi$ that maximizes the generalization performance $\zeta(\pi)$.

## 3.2 DEEP MUTUAL LEARNING

Deep mutual learning (DML) (Zhang et al., 2018b) is a mutual distillation technique in supervised learning. Unlike the traditional teacher-student distillation strategy, DML aligns the probability distributions of multiple student networks by minimizing the KL divergence loss during training, allowing them to learn from each other. Specifically,

$$\mathcal{L}_{\text{DML}} = \mathcal{L}_{\text{SL}} + \alpha \mathcal{L}_{\text{KL}}, \quad (4)$$

where $\mathcal{L}_{\text{SL}}$ and $\mathcal{L}_{\text{KL}}$ represent the supervised learning loss and the KL divergence loss, respectively, $\alpha$ is the weight. Using DML, the student cohort effectively pools their collective estimate of the next most likely classes. Finding out and matching the other most likely classes for each training instance according to their peers increases each student's posterior entropy, which helps them converge to a more robust representation, leading to better generalization.

## 4 THEORETICAL RESULTS

In this section, we present the main results of this paper, demonstrating that enhancing the agent's robustness to irrelevant features will improve its generalization performance.

A key issue is that we do not exactly know the probability distribution $p_{\text{train}}$. Note that $\mathcal{F}_{\text{train}}$ is a subset of $\mathcal{F}$, we naturally assume that the probability distribution $p_{\text{train}}$ can be derived from the normalized probability distribution $p$.

**Assumption 4.1.** For any $f \in \mathcal{F}$, assume that

$$p_{\text{train}}(f) = \frac{p(f) \cdot \mathbb{I}(f \in \mathcal{F}_{\text{train}})}{Z}, \quad p_{\text{eval}}(f) = \frac{p(f) \cdot \mathbb{I}(f \in \mathcal{F}_{\text{eval}})}{1 - Z}, \quad (5)$$

where $Z = \int_{\mathcal{F}_{\text{train}}} p(f) \mathrm{d}f$ and $1 - Z$ is the normalization term, $\mathcal{F}_{\text{eval}} = \mathcal{F} - \mathcal{F}_{\text{train}}$, $\mathbb{I}(\cdot)$ denotes the indicator function.

An interesting fact is that, for a specific policy $\pi$, if we only consider its interaction with $\mathcal{M}_f$, we can establish a bijection between this policy and a certain underlying policy that directly interacts with $\mathcal{M}$. We now denote it as $\mu_f(\cdot|s_t) = \pi(\cdot|f(s_t))$. By further defining the normalized discounted visitation distribution $d^\mu(s) = (1 - \gamma) \sum_{t=0}^{\infty} \gamma^t \mathbb{P}(s_t = s|\mu)$, we can use this underlying policy $\mu_f$ to replace the training and generalization performance of the policy $\pi$. Specifically, we have the following connection:

**Lemma 4.2.** *For any given policy $\pi$, define its underlying policy as $\mu_f(\cdot|s_t) = \pi(\cdot|f(s_t))$, then*

$$\eta(\pi) = \frac{1}{1 - \gamma} \mathbb{E}_{\substack{f \sim p_{\text{train}}(\cdot) \\ s \sim d^{\mu_f}(\cdot) \\ a \sim \mu_f(\cdot|s)}} [r(s, a)], \quad \zeta(\pi) = \frac{1}{1 - \gamma} \mathbb{E}_{\substack{f \sim p(\cdot) \\ s \sim d^{\mu_f}(\cdot) \\ a \sim \mu_f(\cdot|s)}} [r(s, a)]. \quad (6)$$

*Proof.* See Appendix F.1. □

We can thus analyze the generalization problem using the underlying policy $\mu_f$. Then, we define $L_\pi(\tilde{\pi}) = \eta(\pi) + \frac{1}{1-\gamma} \mathbb{E}_{f \sim p_{\text{train}}(\cdot), s \sim d^{\mu_f}(\cdot), a \sim \tilde{\mu}_f(\cdot|s)} [A^{\mu_f}(s, a)]$ as the first-order approximation of $\eta$ (Schulman et al., 2015), we can derive the following lower bounds:

**Theorem 4.3** (Training performance lower bound). *Given any two policies, $\tilde{\pi}$ and $\pi$, the following bound holds:*

$$\eta(\tilde{\pi}) \geq L_\pi(\tilde{\pi}) - \frac{2\gamma\epsilon_{\text{train}}}{(1-\gamma)^2} \underset{\substack{f\sim p_{\text{train}}(\cdot) \\ s\sim d^{\mu_f}(\cdot)}}{\mathbb{E}} [D_{\text{TV}}(\tilde{\mu}_f\|\mu_f)[s]], \tag{7}$$

*where $\epsilon_{\text{train}} = \max_{f\in\mathcal{F}_{\text{train}}}\left\{\max_s\left|\mathbb{E}_{a\sim\tilde{\mu}_f(\cdot|s)}[A^{\mu_f}(s,a)]\right|\right\}$.*

*Proof.* See Appendix F.3. ☐

**Theorem 4.4** (Generalization performance lower bound). *Given any two policies, $\tilde{\pi}$ and $\pi$, the following bound holds:*

$$\zeta(\tilde{\pi}) \geq L_\pi(\tilde{\pi}) - \frac{2r_{\max}(1-Z)}{1-\gamma} - \frac{2\gamma\epsilon_{\text{train}}}{(1-\gamma)^2} \underset{\substack{f\sim p_{\text{train}}(\cdot) \\ s\sim d^{\mu_f}(\cdot)}}{\mathbb{E}} [D_{\text{TV}}(\tilde{\mu}_f\|\mu_f)[s]]$$

$$- \frac{2\delta_{\text{train}}(1-Z)}{1-\gamma} \underset{\substack{f\sim p_{\text{train}}(\cdot) \\ s\sim d^{\tilde{\mu}_f}(\cdot)}}{\mathbb{E}} [D_{\text{TV}}(\tilde{\mu}_f\|\mu_f)[s]] - \frac{2\delta_{\text{eval}}(1-Z)}{1-\gamma} \underset{\substack{f\sim p_{\text{eval}}(\cdot) \\ s\sim d^{\tilde{\mu}_f}(\cdot)}}{\mathbb{E}} [D_{\text{TV}}(\tilde{\mu}_f\|\mu_f)[s]],$$

$$\tag{8}$$

*where $r_{\max} = \max_{s,a}|r(s,a)|$, $\delta_{\text{train}} = \max_{f\in\mathcal{F}_{\text{train}}}\{\max_{s,a}|A^{\mu_f}(s,a)|\}$, and $\delta_{\text{eval}} = \max_{f\in\mathcal{F}_{\text{eval}}}\{\max_{s,a}|A^{\mu_f}(s,a)|\}$.*

*Proof.* See Appendix F.2. ☐

**Explanation.** Building upon Theorems 4.3 and 4.4, we observe that, in contrast to the lower bound on training performance, the lower bound on generalization performance incorporates three additional terms, scaled by the common coefficient $(1-Z)$. This implies that increasing $Z$ contributes to improved generalization performance, with the special case of $Z=1$ resulting in alignment between generalization and training performance. Notably, this theoretical insight was also validated in Figure 2 of Cobbe et al. (2020).

However, once the training level is fixed (i.e., $\mathcal{F}_{\text{train}}$), $Z$ is a constant, improving generalization performance requires constraining the following three terms:

$$\underbrace{\underset{\substack{f\sim p_{\text{train}}(\cdot) \\ s\sim d^{\tilde{\mu}_f}(\cdot)}}{\mathbb{E}} [D_{\text{TV}}(\tilde{\mu}_f\|\mu_f)[s]]}_{\text{denote it as } \mathfrak{D}_1}, \quad \underbrace{\underset{\substack{f\sim p_{\text{eval}}(\cdot) \\ s\sim d^{\tilde{\mu}_f}(\cdot)}}{\mathbb{E}} [D_{\text{TV}}(\tilde{\mu}_f\|\mu_f)[s]]}_{\text{denote it as } \mathfrak{D}_2}, \quad \underbrace{\underset{\substack{f\sim p_{\text{train}}(\cdot) \\ s\sim d^{\mu_f}(\cdot)}}{\mathbb{E}} [D_{\text{TV}}(\tilde{\mu}_f\|\mu_f)[s]]}_{\text{denote it as } \mathfrak{D}_{\text{train}}}. \tag{9}$$

During the training process, we can only empirically bound $\mathfrak{D}_{\text{train}}$. Next, we establish the upper bounds of $\mathfrak{D}_1$ and $\mathfrak{D}_2$. Specifically, we propose the following theorem:

**Theorem 4.5.** *Given any two policies, $\tilde{\pi}$ and $\pi$, the following bound holds:*

$$\mathfrak{D}_1 \leq \left(1 + \frac{2\gamma\sigma_{\text{train}}}{1-\gamma}\right)\mathfrak{D}_{\text{train}}, \quad \mathfrak{D}_2 \leq \left(1 + \frac{2\gamma\sigma_{\text{eval}}}{1-\gamma}\right)\underbrace{\underset{\substack{f\sim p_{\text{eval}}(\cdot) \\ s\sim d^{\mu_f}(\cdot)}}{\mathbb{E}} [D_{\text{TV}}(\tilde{\mu}_f\|\mu_f)[s]]}_{\text{denote it as } \mathfrak{D}_{\text{eval}}}, \tag{10}$$

*where $\sigma_{\text{train}} = \max_{f\in\mathcal{F}_{\text{train}}}\{D_{\text{TV}}^{\max}(\tilde{\mu}_f\|\mu_f)[s]\}$ and $\sigma_{\text{eval}} = \max_{f\in\mathcal{F}_{\text{eval}}}\{D_{\text{TV}}^{\max}(\tilde{\mu}_f\|\mu_f)[s]\}$, $D_{\text{TV}}^{\max}(\tilde{\mu}_f\|\mu_f)[s]$ is defined as $\max_s D_{\text{TV}}(\tilde{\mu}_f\|\mu_f)[s]$.*

*Proof.* See Appendix F.4. ☐

The only problem now is finding the relationship between $\mathfrak{D}_{\text{eval}}$ and $\mathfrak{D}_{\text{train}}$. To achieve this, we would like to first introduce the following definition, which represents the policy robustness to irrelevant features.

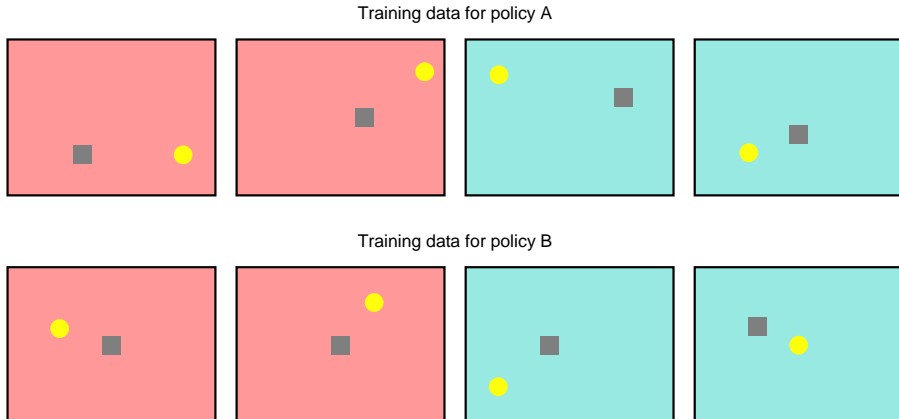

Figure 1: This is a toy environment where the gray agent's goal is to pick up coins.

**Definition 4.6** ($\mathcal{R}$-robust). We say that the policy $\pi$ is $\mathcal{R}$-robust if it satisfies

$$\sup_{s \in \mathcal{S}, \tilde{f}, f \in \mathcal{F}} D_{\mathrm{TV}}(\mu_{\tilde{f}} \| \mu_f)[s] = \mathcal{R}. \tag{11}$$

**Explanation.** This definition demonstrates how the policy $\pi$ is influenced by two different rendering functions, $\tilde{f}$ and $f$, for any given underlying state $s$. If $\mathcal{R} = 0$, it indicates that $D_{\mathrm{TV}}(\mu_{\tilde{f}} \| \mu_f)[s] \equiv 0$, which means that the policy is no longer affected by any irrelevant features.

Our intention in this definition is not to derive the tightest possible bound but rather to demonstrate how policy robustness to irrelevant features can contribute to improved generalization. Subsequently, leveraging Definition 4.6, we establish an upper bound for $\mathfrak{D}_{\mathrm{eval}}$.

**Theorem 4.7.** *Given any two policies, $\tilde{\pi}$ and $\pi$, assume that $\tilde{\pi}$ is $\mathcal{R}_{\tilde{\pi}}$-robust, and $\pi$ is $\mathcal{R}_{\pi}$-robust, then the following bound holds:*

$$\mathfrak{D}_{\mathrm{eval}} \leq \left(1 + \frac{2\gamma\sigma_{\mathrm{train}}}{1-\gamma}\right)\mathcal{R}_{\pi} + \mathcal{R}_{\tilde{\pi}} + \mathfrak{D}_{\mathrm{train}}. \tag{12}$$

*Proof.* See Appendix F.5. $\square$

Altogether, by combining Theorems 4.4, 4.5, and 4.7, we can derive the following corollary:

**Corollary 4.8.** *Given any two policies, $\tilde{\pi}$ and $\pi$, the following bound holds:*

$$\zeta(\tilde{\pi}) \geq L_{\pi}(\tilde{\pi}) - C_{\mathrm{train}}\mathfrak{D}_{\mathrm{train}} - C_{\pi}\mathcal{R}_{\pi} - C_{\tilde{\pi}}\mathcal{R}_{\tilde{\pi}} - C, \tag{13}$$

*where*

$$C_{\mathrm{train}} = \frac{2\delta_{\mathrm{train}}(1-Z)}{1-\gamma}\left(1 + \frac{2\gamma\sigma_{\mathrm{train}}}{1-\gamma}\right) + \frac{2\delta_{\mathrm{eval}}(1-Z)}{1-\gamma}\left(1 + \frac{2\gamma\sigma_{\mathrm{eval}}}{1-\gamma}\right) + \frac{2\gamma\epsilon_{\mathrm{train}}}{(1-\gamma)^2},$$

$$C_{\pi} = \frac{2\delta_{\mathrm{eval}}(1-Z)}{1-\gamma}\left(1 + \frac{2\gamma\sigma_{\mathrm{eval}}}{1-\gamma}\right)\left(1 + \frac{2\gamma\sigma_{\mathrm{train}}}{1-\gamma}\right), \tag{14}$$

$$C_{\tilde{\pi}} = \frac{2\delta_{\mathrm{eval}}(1-Z)}{1-\gamma}\left(1 + \frac{2\gamma\sigma_{\mathrm{eval}}}{1-\gamma}\right), \quad C = \frac{2r_{\mathrm{max}}(1-Z)}{1-\gamma}.$$

**Explanation.** This represents our central theoretical result, demonstrating that enhancing generalization performance requires not only minimizing $\mathfrak{D}_{\mathrm{train}}$ during training but also improving policy robustness to irrelevant features, specifically by reducing $\mathcal{R}_{\pi}$ and $\mathcal{R}_{\tilde{\pi}}$. Furthermore, we emphasize that these results rely solely on the mild Assumptions 3.1, 3.2, and 4.1. Consequently, this constitutes a novel contribution that is broadly applicable to a wide range of algorithms.

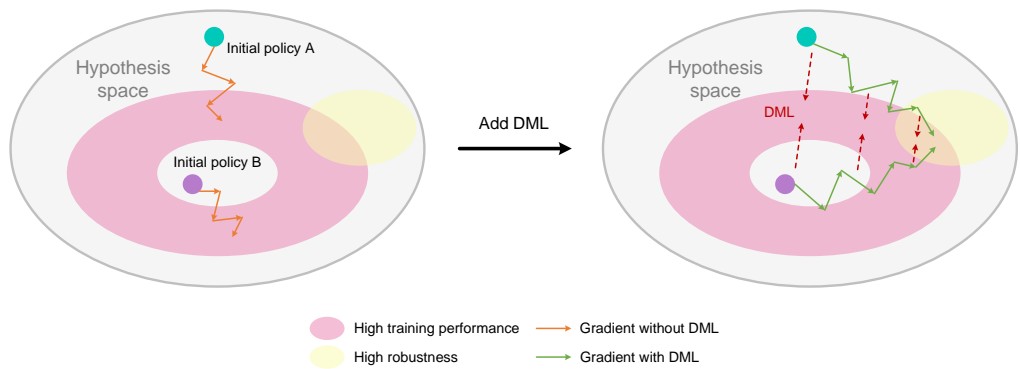

Figure 2: (Left) Independently trained reinforcement learning policies may overfit to irrelevant features. (Right) Through mutual distillation via DML, two policies regularize each other to converge toward a more robust hypothesis space, ultimately improving generalization performance.

## 5 DISTILLATION AS REGULARIZATION

Despite the theoretical advancements, in typical generalization settings, both the underlying MDP and the rendering function remain unknown. Next, we begin by introducing a minimal toy example in Section 5.1, which we then provide an in-depth analyze in Section 5.2 to motivate our hypothesis.

### 5.1 TOY EXAMPLE

Let's consider a simple environment where the agent attempts to pick up coins to earn rewards (see Figure 1). The agent's observations are the current pixels. It is clear that the agent's true objective is to pick up the coins, and the background color is a spurious feature. However, upon observing the training data for policy A, we can see that in the red background, the coins are always on the right side of the agent, while in the cyan background, the coins are always on the left side. As a result, when training policy A using reinforcement learning algorithms, it is likely to exhibit overfitting behavior, such as moving to the right in a red background and to the left in a cyan background.

However, the overfitting of policy A to the background color will fail in the training data of policy B, because in policy B's training data, regardless of whether the background color is red or cyan, the coin can appear either on the left or right side of the agent. Therefore, through DML, policy A is regularized by the behavior of policy B, effectively preventing policy A from overfitting to the background color. In other words, any irrelevant features learned by policy A could lead to suboptimal performance of policy B, and vice versa. Thus, we hypothesize that this process will force both policies to learn the true underlying semantics, ultimately improving generalization performance.

### 5.2 HYPOTHESIS

Motivated by Section 5.1, DML can be viewed as a form of implicit regularization against irrelevant features, as demonstrated in Figure 2, which illustrates two randomly initialized policies independently trained using reinforcement learning algorithms. In this case, since the training samples only include a portion of all possible MDPs, the policies are likely to overfit to irrelevant features and fail to converge to a robust hypothesis space.

Applying DML to the training process of both policies facilitates mutual learning, which can mitigate overfitting to irrelevant features. Due to the randomness of parameter initialization and the interaction process, they generate different training samples, DML encourages both policies to make consistent decisions based on the same observations. As discussed in Section 5.1, any irrelevant features learned by policy A are likely to degrade the performance of policy B, and vice versa. As training progresses, DML will drive both policies to learn more meaningful and useful representations, gradually reducing the divergence between them. Ideally, we hypothesize that both policies will capture the essential aspects of high-dimensional observations as time grows.

# 6 EXPERIMENTS

This section presents our main empirical results. Section 6.1 introduces the implementation details, Section 6.2 validates the effectiveness of DML technique for improving generalization performance, Section 6.3 verifies our central hypothesis, and Section 6.4 confirms our theoretical results.

## 6.1 IMPLEMENTATION DETAILS

We use Procgen (Cobbe et al., 2019; 2020) as the experimental benchmark for testing generalization performance. Procgen is a suite of 16 procedurally generated game-like environments designed to benchmark both sample efficiency and generalization in reinforcement learning, and it has been widely used to test the generalization performance of various reinforcement learning algorithms (Wang et al., 2020; Raileanu & Fergus, 2021; Raileanu et al., 2021; Lyle et al., 2022; Rahman & Xue, 2023; Jesson & Jiang, 2024).

We employ the Proximal Policy Optimization (PPO) (Schulman et al., 2017; Cobbe et al., 2020) as our baseline. Specifically, given a parameterized policy $\pi_\theta$ ($\theta$ represents the parameters), the objective of $\pi_\theta$ is to maximize

$$J(\theta) = \mathbb{E}_{(o_t, a_t) \sim \pi_{\theta_{\text{old}}}} \left\{ \min \left[ r_t(\theta) \cdot \hat{A}(o_t, a_t), \text{clip} \left( r_t(\theta), 1 - \epsilon, 1 + \epsilon \right) \cdot \hat{A}(o_t, a_t) \right] \right\}, \quad (15)$$

where $\hat{A}$ is the advantage estimate, and $r_t(\theta) = \pi_\theta(a_t|o_t)/\pi_{\theta_{\text{old}}}(a_t|o_t)$ is the probability ratio, where $\pi_{\theta_{\text{old}}}$ and $\pi_\theta$ denote the old and current policies, respectively.

We randomly initialize two agents to interact with the environment and collect data separately. Similar to the DML loss (4) used in supervised learning, we also introduce an additional KL divergence loss term, which leads to

$$\mathcal{L}_{\text{DML}} = \mathcal{L}_{\text{RL}} + \alpha \mathcal{L}_{\text{KL}}, \quad (16)$$

where $\mathcal{L}_{\text{RL}}$ is the reinforcement learning loss and $\mathcal{L}_{\text{KL}}$ is the KL divergence loss, $\alpha$ is the weight. And then we optimize the total loss of both agents, which is the average of their DML losses, as shown in Algorithm 1, which we name Mutual Distillation Policy Optimization (MDPO).

---

**Algorithm 1** Mutual Distillation Policy Optimization (MDPO)

---

1: **Initialize:** Two agents $\pi_1, \pi_2$, PPO algorithm $\mathcal{A}$, KL divergence weight $\alpha$
2: **while** training **do**
3:     **for** $i = 1, 2$ **do**
4:         Collect training data: $\mathcal{D}_i \sim \pi_i$
5:         Compute RL loss: $\mathcal{L}_{\text{RL}}^{(i)} \leftarrow \mathcal{A}(\mathcal{D}_i)$
6:         Compute KL loss: $\mathcal{L}_{\text{KL}}^{(i)} \leftarrow D_{\text{KL}}(\pi_{3-i} \| \pi_i)$
7:         Compute DML loss: $\mathcal{L}_{\text{DML}}^{(i)} \leftarrow \mathcal{L}_{\text{RL}}^{(i)} + \alpha \mathcal{L}_{\text{KL}}^{(i)}$
8:     **end for**
9:     Compute total loss: $\mathcal{L} \leftarrow \frac{1}{2} \left( \mathcal{L}_{\text{DML}}^{(1)} + \mathcal{L}_{\text{DML}}^{(2)} \right)$
10:    Optimize $\mathcal{L}$ using gradient descent algorithm
11: **end while**

---

Ultimately, we do not claim to achieve state-of-the-art (SOTA) performance, but rather provide empirical evidence for the non-trivial insight that DML serves as an implicit regularization against irrelevant features, leading to consistent improvements in generalization performance. We also acknowledge the methodological similarities with prior work such as Zhao & Hospedales (2021); despite that, we introduce *representation convergence* (Section 5.2), a novel insight with further supported by strong theoretical analysis (Section 4), constituting our additional contributions.

## 6.2 EMPIRICAL RESULTS

We compare the generalization performance of our MDPO against the PPO baseline on the Procgen benchmark, under the hard-level settings (Cobbe et al., 2020), the results are illustrated in Figure 3. It

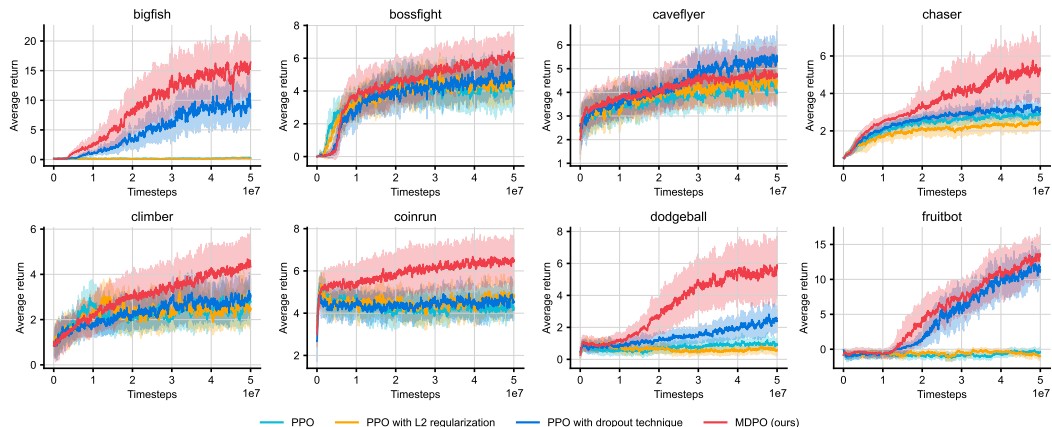

Figure 3: Generalization performance from 500 levels in Procgen benchmark with different methods. The mean and standard deviation are shown across 5 random seeds.

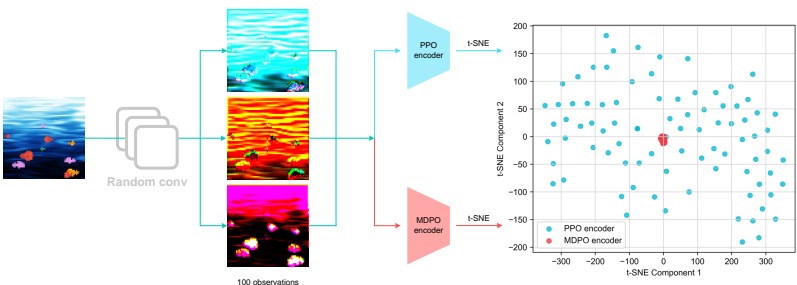

Figure 4: To test the robustness of the trained policy, we obfuscate the agent's observations using convolutional layers randomly initialized with a standard Gaussian distribution.

can be observed that DML technique indeed leads to consistent improvements in generalization performance across all environments. Notably, for the bigfish, dodgeball, and fruitbot environments, we have observed significant improvements. Moreover, the full experimental results for all environments, including training and generalization performance, are provided in Appendix E.

A natural concern arises: how can we determine whether DML improves generalization performance by enhancing the policy robustness against irrelevant features, or simply due to the additional information sharing between these two agents during training (each agent receives additional information than it would from training alone)? To answer this question, we conducted robustness testing in Section 6.3 and added an ablation study in Section 6.4 to support our theory and hypothesis.

### 6.3 ROBUSTNESS TESTING

We design a novel approach to test policy robustness against irrelevant features. For a given frame, we generate *adversarial samples* using random CNNs initialized with a standard Gaussian distribution, as shown in Figure 4. Notably, the feature extraction of MDPO encoder is highly stable and focused (red points), whereas the features extracted by the original PPO encoder are significantly dispersed (blue points).

Moreover, we design a practical measure of $\mathcal{R}$-robustness defined in Definition 4.6. Specifically, for each environment, we run the trained policy (PPO and MDPO) in the environment for 100 steps and obtain observations

| Algo\Env | caveflyer | chaser | climber | fruitbot |
|----------|-----------|--------|---------|----------|
| PPO | 1.0000 | 1.0000 | 1.0000 | 1.0000 |
| MDPO | **0.9877** | **0.9982** | **0.8344** | **0.6973** |

| Algo\Env | heist | jumper | leaper | plunder |
|----------|-------|--------|--------|---------|
| PPO | 0.9683 | 0.9699 | 1.0000 | 1.0000 |
| MDPO | **0.9142** | **0.9313** | **0.9423** | **0.9431** |

Table 1: A simple practical measure of $\mathcal{R}$-robustness defined in Definition 4.6.

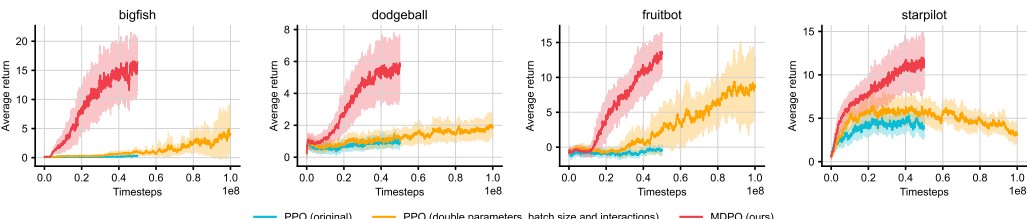

Figure 5: Generalization performance of PPO baseline with double model size, batch size, and total number of interactions, compared to original PPO and MDPO (for training results, see Figure 8).

$o_1, o_2, \ldots, o_{100}$. Then, for each $o_i$ we use 100 random CNNs to simulate rendering function samples $f_1^{(i)}, f_2^{(i)}, \ldots, f_{100}^{(i)}$ and compute the TV divergence of the policy between the adversarial samples and the original observations, i.e., $D_{\mathrm{TV}}(\pi_\theta(\cdot|o_i)\|\pi_\theta(\cdot|f_j^{(i)}(o_i)))$, where $i, j = 1, 2, \ldots, 100$. We then take the maximum of these values as a simple practical measure of $\mathcal{R}$-robustness:

$$\hat{\mathcal{R}} := \max_{i,j} D_{\mathrm{TV}}(\pi_\theta(\cdot|o_i)\|\pi_\theta(\cdot|f_j^{(i)}(o_i))), \tag{17}$$

the results are shown in Table 1. We can see that MDPO achieves a significantly lower $\hat{\mathcal{R}}$ than PPO, showing that DML effectively improves the policy robustness to irrelevant features, which serves as further strong evidence for our hypothesis.

## 6.4 ABLATION STUDY

We design additional ablation experiments. Specifically, we *double* the model size, batch size, and total number of interactions for the PPO baseline, as shown in Figure 5. It can be seen that PPO baseline still fails to match the performance of MDPO, demonstrating that naively scaling up the PPO baseline does not lead to stable improvements in generalization performance.

Furthermore, we retrain a PPO *linear probe* on top of the *frozen* encoders of the trained PPO and MDPO policies, training for only 1M steps (2% of the original training steps), the final generalization performance during the last 10% steps is shown in Table 2. It can be seen that the PPO linear probe

| Algo\Env | bigfish | chaser | dodgeball | fruitbot |
|---|---|---|---|---|
| PPO (PPO encoder) | $0.19^{\pm 0.14}$ | $2.57^{\pm 0.28}$ | $0.71^{\pm 0.34}$ | $-0.39^{\pm 0.46}$ |
| PPO (MDPO encoder) | $\mathbf{22.67}^{\pm 6.40}$ | $\mathbf{6.22}^{\pm 1.36}$ | $\mathbf{4.70}^{\pm 1.91}$ | $\mathbf{11.22}^{\pm 2.16}$ |

Table 2: Generalization performance of PPO linear probe on top of the *frozen* encoders.

| $\alpha$ in MDPO\Env | bigfish | chaser | dodgeball | fruitbot |
|---|---|---|---|---|
| 0 (baseline) | $0.26^{\pm 0.23}$ | $0.92^{\pm 0.46}$ | $-0.50^{\pm 0.81}$ | $3.99^{\pm 0.21}$ |
| 0.1 | $9.87^{\pm 4.57}$ | $4.35^{\pm 1.63}$ | $11.94^{\pm 2.96}$ | $10.97^{\pm 2.72}$ |
| 1 | $\mathbf{16.11}^{\pm 4.63}$ | $\mathbf{5.66}^{\pm 1.98}$ | $\mathbf{13.23}^{\pm 3.04}$ | $\mathbf{11.28}^{\pm 3.04}$ |
| 10 | $7.69^{\pm 3.65}$ | $4.35^{\pm 1.48}$ | $2.31^{\pm 2.41}$ | $8.54^{\pm 2.27}$ |

Table 3: Generalization performance of MDPO under different KL divergence weights.

trained on the MDPO encoder achieves significantly better generalization performance, indicating that DML helps the policy learn better (more robust) representations. Moreover, we add a sensitivity analysis of the KL divergence weight $\alpha$, and the results are presented in Table 3.

## 7 CONCLUSION

In this paper, we provide a novel theoretical framework to explain the generalization problem in deep reinforcement learning. We further hypothesize that DML, as a form of implicit regularization, effectively prevents the policy from overfitting to irrelevant features. Strong empirical results support our central theory and hypothesis, demonstrating that our approach can improve the generalization performance of reinforcement learning systems by enhancing robustness against irrelevant features. Our work provides valuable insights and elegant solutions into the development of more adaptable and robust policies capable of generalizing across diverse environments.

ETHICS STATEMENT

All authors have read and adhere to the ICLR Code of Ethics. This paper does not involve studies with human subjects, dataset releases, potentially harmful insights, methodologies or applications, conflicts of interest, or concerns related to discrimination, bias, or fairness.

REPRODUCIBILITY STATEMENT

We provide the full reproducible code in the supplementary materials, which is fully consistent with the hyperparameters listed in Appendix C. All theorems are fully proven in the appendix. Anyone can easily reproduce the results of our paper based on the provided code and hyperparameter settings.

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

## A  LLM USAGE

In this work, large language models (LLMs) were used to assist in refining and polishing the writing.

## B  LIMITATIONS

While our method demonstrates that mutual distillation improves robustness and generalization, it inevitably introduces additional computational costs. Specifically, MDPO requires *twice* the number of trainable parameters and roughly twice the environment interaction steps compared to a single-policy baseline. Consequently, the method may be less practical in settings with limited computational resources or when sample efficiency is critical. Addressing these efficiency concerns, such as via parameter sharing or selective distillation, is an interesting direction for future work.

## C  HYPERPARAMETERS

Table 4 shows the detailed hyperparameter settings in our code, with the main hyperparameters consistent with the hard-level settings in Cobbe et al. (2020), except that we train for 50M steps instead of 200M. We train the policy on the initial 500 levels and then test its generalization performance across the full distribution of levels.

Table 4: Detailed hyperparameters in Procgen.

| Hyperparameter\Algorithm | PPO (Schulman et al., 2017) | MDPO (ours) |
|---|---|---|
| Number of workers | 64 | 64 |
| Horizon | 256 | 256 |
| Learning rate | 0.0005 | 0.0005 |
| Learning rate decay | No | No |
| Optimizer | Adam | Adam |
| Total interaction steps | 50M | 50M |
| Update epochs | 3 | 3 |
| Mini-batches | 8 | 8 |
| Batch size | 16384 | 16384 |
| Mini-batch size | 2048 | 2048 |
| Discount factor $\gamma$ | 0.999 | 0.999 |
| GAE parameter $\lambda$ | 0.95 | 0.95 |
| Value loss coefficient $c_1$ | 0.5 | 0.5 |
| Entropy loss coefficient $c_2$ | 0.01 | 0.01 |
| Clipping parameter $\epsilon$ | 0.2 | 0.2 |
| KL divergence weight $\alpha$ | - | 1.0 |

# D  THE REPRESENTATION CONVERGENCE PHENOMENON

To further demonstrate that mutual distillation indeed promotes representation convergence, we conducted the following experiment: we compared the *Centered Kernel Alignment* (CKA) of two agents in MDPO on the same batch of adversarial examples at different training stages, under different KL divergence weight $\alpha$, the results are shown in the Table 5 below:

Table 5: CKA of two MDPO policies under different $\alpha$.

| Algo\Training stage | 0% | 25% | 50% | 75% | 100% |
|---|---|---|---|---|---|
| MDPO ($\alpha = 1.0$) | 0.649 | 0.769 | 0.797 | 0.850 | 0.867 |
| MDPO ($\alpha = 0.0$) | 0.649 | 0.185 | 0.131 | 0.146 | 0.004 |

It is evident that after mutual distillation ($\alpha = 1.0$), the two agents learned more robust representations, as their representations of the same batch of adversarial examples became increasingly similar. In contrast, when the distillation weight $\alpha = 0.0$, their representations diverge over time. We further evaluated the *cosine similarity* of the representations of adversarial examples encoded by PPO and MDPO across four environments, as shown in the Table 6.

Table 6: Cosine similarity of the representations.

| Algo\Env | coinrun | dodgeball | fruitbot | starpilot |
|---|---|---|---|---|
| PPO encoder | 0.301 | -0.006 | 0.180 | 0.027 |
| MDPO encoder | **0.781** | **0.585** | **0.547** | **0.718** |

We can see that MDPO achieves significantly higher cosine similarity for the adversarial samples, showing that MDPO has learned more robust representations with respect to irrelevant features.

# E MORE RESULTS

## E.1 FULL RESULTS

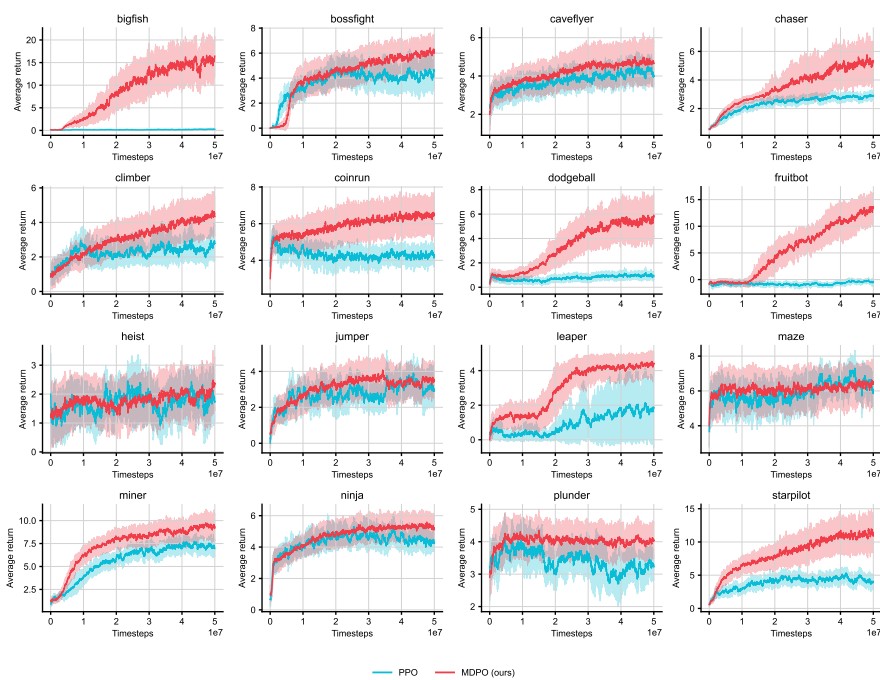

Figure 6: Generalization performance of PPO and MDPO from 500 levels in each environment.

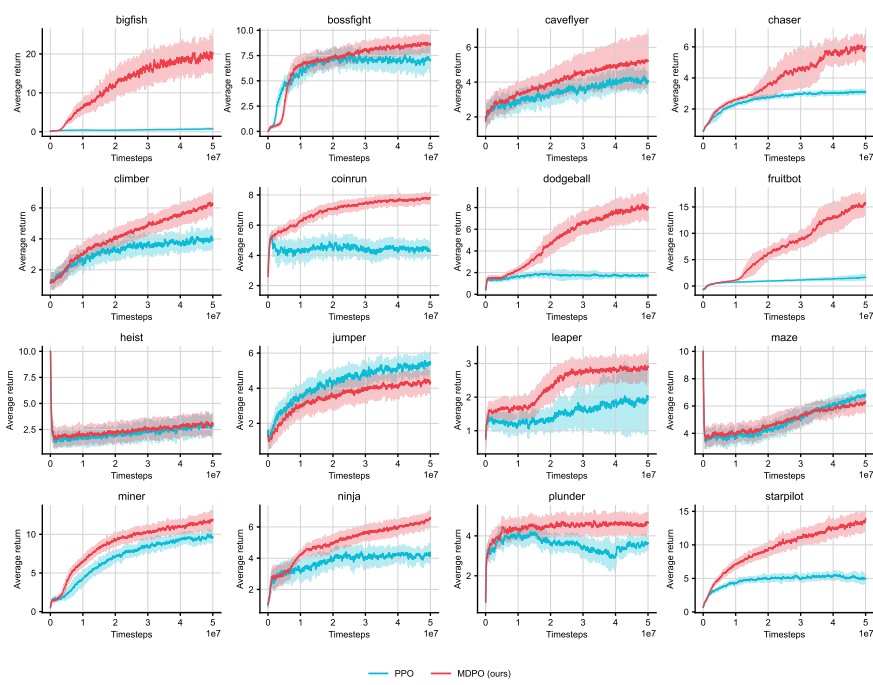

Figure 7: Training performance of PPO and MDPO from 500 levels in each environment.

### E.2 MORE ABLATION RESULTS

Here, we additionally present the training curves from the Ablation Study (Section 6.4), as shown in Figure 8.

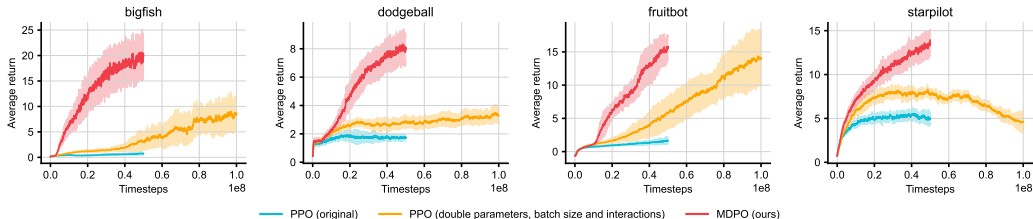

Figure 8: Training performance of PPO baseline with double model size, batch size, and total number of interactions, compared to original PPO and MDPO.

Interestingly, although the scaled-up PPO nearly matches MDPO in training performance during the final stage of training in the fruitbot environment, there remains a substantial gap in their generalization performance (as shown in Figure 5). This provides further strong evidence that DML effectively enhances the policy robustness to irrelevant features, as MDPO achieves significantly better generalization performance despite comparable training performance.

### E.3 ADDITIONAL VISUALIZATIONS

We also generate adversarial samples by adjusting the brightness, contrast, saturation, and hue of the images, and test the robustness of the PPO encoder and our MDPO encoder, as shown in Figure 9.

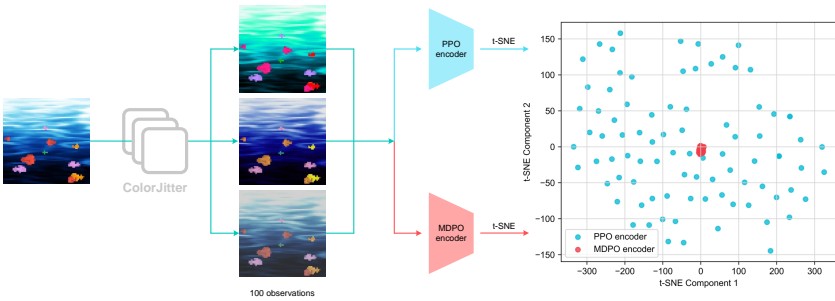

Figure 9: The robustness of PPO and MDPO to brightness, contrast, saturation, and hue.

We can see that the MDPO policy has also learned robustness representations to these irrelevant factors, while the PPO policy remains sensitive to them. Additionally, we present adversarial samples generated by random CNNs, as shown in Figure 10, as well as those generated by randomly adjusting brightness, contrast, saturation, and hue, as can be seen from Figure 11.

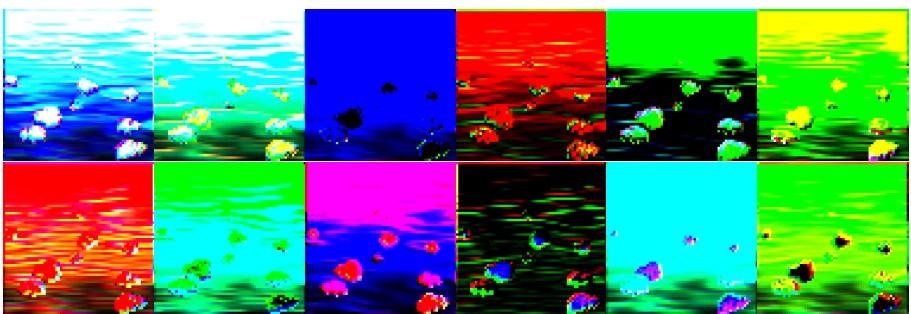

Figure 10: Adversarial samples generated by random CNNs.

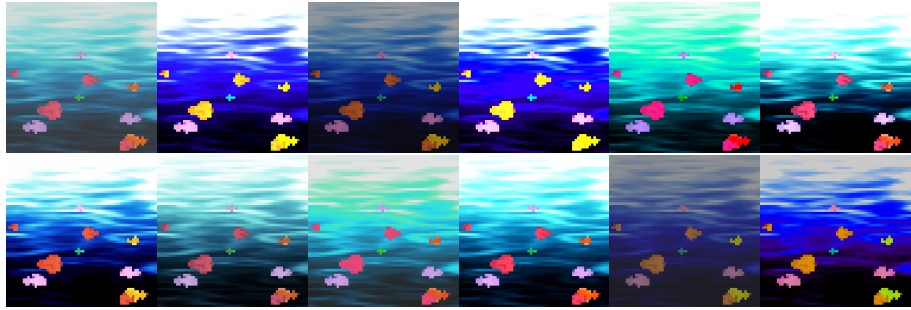

Figure 11: Adversarial samples generated by different brightness, contrast, saturation, and hue.

# F  PROOFS

Let's start with some useful lemmas.

**Lemma F.1** (Performance difference). *Let $\mu_f(\cdot|s_t) = \pi(\cdot|f(s_t))$ and $\tilde{\mu}_f(\cdot|s_t) = \tilde{\pi}(\cdot|f(s_t))$, define training and generalization performance as*

$$\eta(\pi) = \frac{1}{1-\gamma} \underset{\substack{f \sim p_{\text{train}}(\cdot) \\ s \sim d^{\mu_f}(\cdot) \\ a \sim \mu_f(\cdot|s)}}{\mathbb{E}} [r(s,a)], \quad \zeta(\pi) = \frac{1}{1-\gamma} \underset{\substack{f \sim p(\cdot) \\ s \sim d^{\mu_f}(\cdot) \\ a \sim \mu_f(\cdot|s)}}{\mathbb{E}} [r(s,a)]. \tag{18}$$

*Then the differences in training and generalization performance can be expressed as*

$$\eta(\tilde{\pi}) - \eta(\pi) = \frac{1}{1-\gamma} \underset{\substack{f \sim p_{\text{train}}(\cdot) \\ s \sim d^{\tilde{\mu}_f}(\cdot) \\ a \sim \tilde{\mu}_f(\cdot|s)}}{\mathbb{E}} [A^{\mu_f}(s,a)], \quad \zeta(\tilde{\pi}) - \zeta(\pi) = \frac{1}{1-\gamma} \underset{\substack{f \sim p(\cdot) \\ s \sim d^{\tilde{\mu}_f}(\cdot) \\ a \sim \tilde{\mu}_f(\cdot|s)}}{\mathbb{E}} [A^{\mu_f}(s,a)]. \tag{19}$$

*Proof.* This result can be directly derived from Kakade & Langford (2002). □

**Lemma F.2.** *The divergence between two normalized discounted visitation distribution, $\|d^{\tilde{\mu}} - d^{\mu}\|_1$, is bounded by an average divergence of $\tilde{\mu}$ and $\mu$:*

$$\|d^{\tilde{\mu}} - d^{\mu}\|_1 \le \frac{\gamma}{1-\gamma} \underset{s \sim d^{\mu}(\cdot)}{\mathbb{E}} [\|\tilde{\mu} - \mu\|_1] = \frac{2\gamma}{1-\gamma} \underset{s \sim d^{\mu}(\cdot)}{\mathbb{E}} [D_{\text{TV}}(\tilde{\mu}\|\mu)[s]], \tag{20}$$

*where $D_{\text{TV}}(\tilde{\mu}\|\mu)[s] = \frac{1}{2}\sum_{a \in \mathcal{A}} |\tilde{\mu}(a|s) - \mu(a|s)|$ represents the Total Variation (TV) distance.*

*Proof.* See Achiam et al. (2017). □

**Lemma F.3.** *Given any state $s \in \mathcal{S}$, any two policies $\tilde{\mu}$ and $\mu$, the average advantage, $\mathbb{E}_{a \sim \tilde{\mu}(\cdot|s)}[A^{\mu}(s,a)]$, is bounded by*

$$\left|\mathbb{E}_{a \sim \tilde{\mu}(\cdot|s)}[A^{\mu}(s,a)]\right| \le 2D_{\text{TV}}(\tilde{\mu}\|\mu)[s] \cdot \max_a |A^{\mu}(s,a)|. \tag{21}$$

*Proof.* Note that

$$\begin{aligned}
\mathbb{E}_{a \sim \mu(\cdot|s)}[A^{\mu}(s,a)] &= \mathbb{E}_{a \sim \mu(\cdot|s)}[Q^{\mu}(s,a) - V^{\mu}(s)] \\
&= \mathbb{E}_{a \sim \mu(\cdot|s)}[Q^{\mu}(s,a)] - V^{\mu}(s) \\
&= V^{\mu}(s) - V^{\mu}(s) \\
&= 0,
\end{aligned} \tag{22}$$

thus,

$$\begin{aligned}
\left|\mathbb{E}_{a \sim \tilde{\mu}(\cdot|s)}[A^{\mu}(s,a)]\right| &= \left|\mathbb{E}_{a \sim \tilde{\mu}(\cdot|s)}[A^{\mu}(s,a)] - \mathbb{E}_{a \sim \mu(\cdot|s)}[A^{\mu}(s,a)]\right| \\
&\le \|\tilde{\mu} - \mu\|_1 \cdot \|A^{\mu}(s,a)\|_{\infty} \\
&= 2D_{\text{TV}}(\tilde{\mu}\|\mu)[s] \cdot \max_a |A^{\mu}(s,a)|.
\end{aligned} \tag{23}$$

This is a widely used trick (Schulman et al., 2015; Zhuang et al., 2023; Gan et al., 2024). □

In addition, using the above lemmas, the following corollary can be obtained, which will be repeatedly used in our proof.

**Corollary F.4.** *Given any two policies, $\tilde{\mu}$ and $\mu$, the following bound holds:*

$$\left| \underset{\substack{s \sim d^{\tilde{\mu}}(\cdot) \\ a \sim \tilde{\mu}(\cdot|s)}}{\mathbb{E}} [A^{\mu}(s,a)] - \underset{\substack{s \sim d^{\mu}(\cdot) \\ a \sim \tilde{\mu}(\cdot|s)}}{\mathbb{E}} [A^{\mu}(s,a)] \right| \le \frac{2\epsilon\gamma}{1-\gamma} \underset{s \sim d^{\mu}(\cdot)}{\mathbb{E}} [D_{\text{TV}}(\tilde{\mu}\|\mu)[s]], \tag{24}$$

*where $\epsilon = \max_s \left| \mathbb{E}_{a \sim \tilde{\mu}(\cdot|s)} [A^{\mu}(s,a)] \right|$.*

*Proof.* We rewrite the expectation as

$$
\left| \underset{\substack{s \sim d^{\tilde{\mu}}(\cdot) \\ a \sim \tilde{\mu}(\cdot|s)}}{\mathbb{E}} [A^{\mu}(s,a)] - \underset{\substack{s \sim d^{\mu}(\cdot) \\ a \sim \tilde{\mu}(\cdot|s)}}{\mathbb{E}} [A^{\mu}(s,a)] \right| = \left| \underset{s \sim d^{\tilde{\mu}}(\cdot)}{\mathbb{E}} \left\{ \underset{a \sim \tilde{\mu}(\cdot|s)}{\mathbb{E}} [A^{\mu}(s,a)] \right\} - \underset{s \sim d^{\mu}(\cdot)}{\mathbb{E}} \left\{ \underset{a \sim \tilde{\mu}(\cdot|s)}{\mathbb{E}} [A^{\mu}(s,a)] \right\} \right|,
$$

(25)

where the expectation $\mathbb{E}_{a \sim \tilde{\mu}(\cdot|s)} [A^{\mu}(s,a)]$ is a function of $s$, then

$$
\left| \underset{s \sim d^{\tilde{\mu}}(\cdot)}{\mathbb{E}} \left\{ \underset{a \sim \tilde{\mu}(\cdot|s)}{\mathbb{E}} [A^{\mu}(s,a)] \right\} - \underset{s \sim d^{\mu}(\cdot)}{\mathbb{E}} \left\{ \underset{a \sim \tilde{\mu}(\cdot|s)}{\mathbb{E}} [A^{\mu}(s,a)] \right\} \right| \leq \left\| d^{\tilde{\mu}} - d^{\mu} \right\|_1 \cdot \left\| \underset{a \sim \tilde{\mu}(\cdot|s)}{\mathbb{E}} [A^{\mu}(s,a)] \right\|_\infty.
$$

(26)

Next, according to Lemma F.2, we have

$$
\left\| d^{\tilde{\mu}} - d^{\mu} \right\|_1 \cdot \left\| \underset{a \sim \tilde{\mu}(\cdot|s)}{\mathbb{E}} [A^{\mu}(s,a)] \right\|_\infty = \left\| d^{\tilde{\mu}} - d^{\mu} \right\|_1 \cdot \epsilon \leq \frac{2\epsilon\gamma}{1-\gamma} \underset{s \sim d^{\mu}(\cdot)}{\mathbb{E}} [D_{\text{TV}}(\tilde{\mu} \| \mu)[s]],
$$

(27)

concluding the proof. $\square$

### F.1 PROOF OF LEMMA 4.2

**Lemma 4.2.** *For any given policy $\pi$, define its underlying policy as $\mu_f(\cdot|s_t) = \pi(\cdot|f(s_t))$, then*

$$
\eta(\pi) = \frac{1}{1-\gamma} \underset{\substack{f \sim p_{\text{train}}(\cdot) \\ s \sim d^{\mu_f}(\cdot) \\ a \sim \mu_f(\cdot|s)}}{\mathbb{E}} [r(s,a)], \quad \zeta(\pi) = \frac{1}{1-\gamma} \underset{\substack{f \sim p(\cdot) \\ s \sim d^{\mu_f}(\cdot) \\ a \sim \mu_f(\cdot|s)}}{\mathbb{E}} [r(s,a)].
$$

(28)

*Proof.* According to the definition of training and generalization performance in (3), we have

$$
\eta(\pi) = \mathbb{E}_{f \sim p_{\text{train}}(\cdot), \tau_f \sim \pi} \left[ \sum_{t=0}^{\infty} \gamma^t r_f(o_t^f, a_t) \right], \quad \zeta(\pi) = \mathbb{E}_{f \sim p(\cdot), \tau_f \sim \pi} \left[ \sum_{t=0}^{\infty} \gamma^t r_f(o_t^f, a_t) \right].
$$

(29)

To prove Lemma 4.2, we only need to show that for any given $f \in \mathcal{F}$, the following equation holds:

$$
\frac{1}{1-\gamma} \underset{\substack{s \sim d^{\mu_f}(\cdot) \\ a \sim \mu_f(\cdot|s)}}{\mathbb{E}} [r(s,a)] = \mathbb{E}_{\tau_f \sim \pi} \left[ \sum_{t=0}^{\infty} \gamma^t r_f(o_t^f, a_t) \right].
$$

(30)

According to the definition of the normalized discounted visitation distribution $d^{\mu}(s) = (1 - \gamma) \sum_{t=0}^{\infty} \gamma^t \mathbb{P}(s_t = s|\mu)$, we have

$$
\frac{1}{1-\gamma} \underset{\substack{s \sim d^{\mu_f}(\cdot) \\ a \sim \mu_f(\cdot|s)}}{\mathbb{E}} [r(s,a)] = \frac{1}{1-\gamma} \sum_{s \in \mathcal{S}} (1-\gamma) \sum_{t=0}^{\infty} \gamma^t \mathbb{P}(s_t = s|\mu_f) \sum_{a \in \mathcal{A}} \mu_f(a|s) \cdot r(s,a)
$$

(31)

$$
= \sum_{t=0}^{\infty} \sum_{s \in \mathcal{S}} \mathbb{P}(s_t = s|\mu_f) \sum_{a \in \mathcal{A}} \mu_f(a|s) \cdot \gamma^t r(s,a)
$$

Next, according to Assumption 3.2, we have

$$
\frac{1}{1-\gamma} \underset{\substack{s \sim d^{\mu_f}(\cdot) \\ a \sim \mu_f(\cdot|s)}}{\mathbb{E}} [r(s,a)] = \sum_{t=0}^{\infty} \sum_{s \in \mathcal{S}} \mathbb{P}(s_t = s|\mu_f) \sum_{a \in \mathcal{A}} \mu_f(a|s) \cdot \gamma^t r(s,a)
$$

$$
= \sum_{t=0}^{\infty} \sum_{s \in \mathcal{S}} \mathbb{P}(f(s_t) = f(s)|\mu_f) \sum_{a \in \mathcal{A}} \pi(a|f(s)) \cdot \gamma^t r_f(f(s), a)
$$

(32)

$$
\overset{f(s) = o^f, f(s_t) = o_t^f}{=} \sum_{t=0}^{\infty} \sum_{o^f \in \mathcal{O}_f} \mathbb{P}(o_t^f = o^f|\pi) \sum_{a \in \mathcal{A}} \pi(a|o^f) \cdot \gamma^t r_f(o^f, a)
$$

$$
= \mathbb{E}_{\tau_f \sim \pi} \left[ \sum_{t=0}^{\infty} \gamma^t r_f(o_t^f, a_t) \right],
$$

concluding the proof. □

## F.2   Proof of Theorem 4.4

**Theorem 4.4.** *Given any two policies, $\tilde{\pi}$ and $\pi$, the following bound holds:*

$$\zeta(\tilde{\pi}) \geq L_\pi(\tilde{\pi}) - \frac{2r_{\max}(1-Z)}{1-\gamma} - \frac{2\gamma\epsilon_{\text{train}}}{(1-\gamma)^2} \mathop{\mathbb{E}}_{\substack{f \sim p_{\text{train}}(\cdot) \\ s \sim d^{\mu_f}(\cdot)}} [D_{\text{TV}}(\tilde{\mu}_f \| \mu_f)[s]]$$

$$- \frac{2\delta_{\text{train}}(1-Z)}{1-\gamma} \mathop{\mathbb{E}}_{\substack{f \sim p_{\text{train}}(\cdot) \\ s \sim d^{\tilde{\mu}_f}(\cdot)}} [D_{\text{TV}}(\tilde{\mu}_f \| \mu_f)[s]] - \frac{2\delta_{\text{eval}}(1-Z)}{1-\gamma} \mathop{\mathbb{E}}_{\substack{f \sim p_{\text{eval}}(\cdot) \\ s \sim d^{\tilde{\mu}_f}(\cdot)}} [D_{\text{TV}}(\tilde{\mu}_f \| \mu_f)[s]]. \tag{33}$$

*Proof.* Let's start with the first-order approximation of the training performance (Schulman et al., 2015), denote it as

$$L_\pi(\tilde{\pi}) = \eta(\pi) + \frac{1}{1-\gamma} \mathop{\mathbb{E}}_{\substack{f \sim p_{\text{train}}(\cdot) \\ s \sim d^{\mu_f}(\cdot) \\ a \sim \tilde{\mu}_f(\cdot|s)}} [A^{\mu_f}(s,a)]. \tag{34}$$

Then, we are trying to bound the difference between $\zeta(\tilde{\pi})$ and $L_\pi(\tilde{\pi})$, according to Lemma F.1, that is,

$$|\zeta(\tilde{\pi}) - L_\pi(\tilde{\pi})|$$

$$= \left| \zeta(\pi) - \eta(\pi) + \frac{1}{1-\gamma} \mathop{\mathbb{E}}_{\substack{f \sim p(\cdot) \\ s \sim d^{\tilde{\mu}_f}(\cdot) \\ a \sim \tilde{\mu}_f(\cdot|s)}} [A^{\mu_f}(s,a)] - \frac{1}{1-\gamma} \mathop{\mathbb{E}}_{\substack{f \sim p_{\text{train}}(\cdot) \\ s \sim d^{\mu_f}(\cdot) \\ a \sim \tilde{\mu}_f(\cdot|s)}} [A^{\mu_f}(s,a)] \right|$$

$$= \frac{1}{1-\gamma} \left| \mathop{\mathbb{E}}_{\substack{f \sim p(\cdot) \\ s \sim d^{\mu_f}(\cdot) \\ a \sim \mu_f(\cdot|s)}} [r(s,a)] - \mathop{\mathbb{E}}_{\substack{f \sim p_{\text{train}}(\cdot) \\ s \sim d^{\mu_f}(\cdot) \\ a \sim \mu_f(\cdot|s)}} [r(s,a)] + \mathop{\mathbb{E}}_{\substack{f \sim p(\cdot) \\ s \sim d^{\tilde{\mu}_f}(\cdot) \\ a \sim \tilde{\mu}_f(\cdot|s)}} [A^{\mu_f}(s,a)] - \mathop{\mathbb{E}}_{\substack{f \sim p_{\text{train}}(\cdot) \\ s \sim d^{\mu_f}(\cdot) \\ a \sim \tilde{\mu}_f(\cdot|s)}} [A^{\mu_f}(s,a)] \right|$$

$$\leq \frac{1}{1-\gamma} \left\{ \left| \mathop{\mathbb{E}}_{\substack{f \sim p(\cdot) \\ s \sim d^{\mu_f}(\cdot) \\ a \sim \mu_f(\cdot|s)}} [r(s,a)] - \mathop{\mathbb{E}}_{\substack{f \sim p_{\text{train}}(\cdot) \\ s \sim d^{\mu_f}(\cdot) \\ a \sim \mu_f(\cdot|s)}} [r(s,a)] \right| + \left| \mathop{\mathbb{E}}_{\substack{f \sim p(\cdot) \\ s \sim d^{\tilde{\mu}_f}(\cdot) \\ a \sim \tilde{\mu}_f(\cdot|s)}} [A^{\mu_f}(s,a)] - \mathop{\mathbb{E}}_{\substack{f \sim p_{\text{train}}(\cdot) \\ s \sim d^{\mu_f}(\cdot) \\ a \sim \tilde{\mu}_f(\cdot|s)}} [A^{\mu_f}(s,a)] \right| \right\}. \tag{35}$$

We can bound these two terms separately. Simplifying the notation, denote $g(f) = \mathbb{E}_{s \sim d^{\mu_f}(\cdot), a \sim \mu_f(\cdot|s)} [r(s,a)]$, we can thus rewrite the first term as

$$\left| \mathop{\mathbb{E}}_{\substack{f \sim p(\cdot) \\ s \sim d^{\mu_f}(\cdot) \\ a \sim \mu_f(\cdot|s)}} [r(s,a)] - \mathop{\mathbb{E}}_{\substack{f \sim p_{\text{train}}(\cdot) \\ s \sim d^{\mu_f}(\cdot) \\ a \sim \mu_f(\cdot|s)}} [r(s,a)] \right| = \left| \mathop{\mathbb{E}}_{f \sim p(\cdot)} [g(f)] - \mathop{\mathbb{E}}_{f \sim p_{\text{train}}(\cdot)} [g(f)] \right|, \tag{36}$$

then

$$\left| \mathop{\mathbb{E}}_{f \sim p(\cdot)} [g(f)] - \mathop{\mathbb{E}}_{f \sim p_{\text{train}}(\cdot)} [g(f)] \right| = \left| \int_{\mathcal{F}} p(f) \cdot g(f) \mathrm{d}f - \int_{\mathcal{F}_{\text{train}}} p_{\text{train}}(f) \cdot g(f) \mathrm{d}f \right|. \tag{37}$$

Next, according to Assumption 4.1,

$$
\left| \int_{\mathcal{F}} p(f) \cdot g(f) \mathrm{d}f - \int_{\mathcal{F}_{\text{train}}} p_{\text{train}}(f) \cdot g(f) \mathrm{d}f \right| = \left| \int_{\mathcal{F}} p(f) \cdot g(f) \mathrm{d}f - \int_{\mathcal{F}_{\text{train}}} \frac{p(f)}{Z} \cdot g(f) \mathrm{d}f \right|
$$

$$
= \left| \int_{\mathcal{F}_{\text{train}}} p(f) \cdot g(f) \mathrm{d}f - \int_{\mathcal{F}_{\text{train}}} \frac{p(f)}{Z} \cdot g(f) \mathrm{d}f + \int_{\mathcal{F} - \mathcal{F}_{\text{train}}} p(f) \cdot g(f) \mathrm{d}f \right|
$$

$$
= \left| \int_{\mathcal{F}_{\text{train}}} \frac{Z-1}{Z} p(f) \cdot g(f) \mathrm{d}f + \int_{\mathcal{F} - \mathcal{F}_{\text{train}}} p(f) \cdot g(f) \mathrm{d}f \right|,
\tag{38}
$$

where $Z = \int_{\mathcal{F}_{\text{train}}} p(f) \mathrm{d}f \leq 1$, thus,

$$
\left| \int_{\mathcal{F}_{\text{train}}} \frac{Z-1}{Z} p(f) \cdot g(f) \mathrm{d}f + \int_{\mathcal{F} - \mathcal{F}_{\text{train}}} p(f) \cdot g(f) \mathrm{d}f \right|
$$

$$
\leq \left| \int_{\mathcal{F}_{\text{train}}} \frac{Z-1}{Z} p(f) \cdot g(f) \mathrm{d}f \right| + \left| \int_{\mathcal{F} - \mathcal{F}_{\text{train}}} p(f) \cdot g(f) \mathrm{d}f \right|
\tag{39}
$$

$$
\leq \frac{1-Z}{Z} \left| \int_{\mathcal{F}_{\text{train}}} p(f) \cdot g(f) \mathrm{d}f \right| + \left| \int_{\mathcal{F} - \mathcal{F}_{\text{train}}} p(f) \cdot g(f) \mathrm{d}f \right|.
$$

Meanwhile,

$$
|g(f)| = \left| \mathop{\mathbb{E}}_{\substack{s \sim d^{\mu_f}(\cdot) \\ a \sim \mu_f(\cdot|s)}} [r(s,a)] \right| = \left| \sum_{s \in \mathcal{S}} (1-\gamma) \sum_{t=0}^{\infty} \gamma^t \mathbb{P}(s_t = s | \mu_f) \sum_{a \in \mathcal{A}} \mu_f(a|s) \cdot r(s,a) \right|
$$

$$
\leq (1-\gamma) \sum_{t=0}^{\infty} \sum_{s \in \mathcal{S}} \mathbb{P}(s_t = s | \mu_f) \sum_{a \in \mathcal{A}} \mu_f(a|s) \cdot \gamma^t |r(s,a)|
\tag{40}
$$

$$
\leq (1-\gamma) \sum_{t=0}^{\infty} \gamma^t r_{\max} = r_{\max},
$$

where $r_{\max} = \max_{s,a} |r(s,a)|$, then we can bound the first term as

$$
\left| \mathop{\mathbb{E}}_{\substack{f \sim p(\cdot) \\ s \sim d^{\mu_f}(\cdot) \\ a \sim \mu_f(\cdot|s)}} [r(s,a)] - \mathop{\mathbb{E}}_{\substack{f \sim p_{\text{train}}(\cdot) \\ s \sim d^{\mu_f}(\cdot) \\ a \sim \mu_f(\cdot|s)}} [r(s,a)] \right| \leq \frac{1-Z}{Z} \left| \int_{\mathcal{F}_{\text{train}}} p(f) \cdot g(f) \mathrm{d}f \right| + \left| \int_{\mathcal{F} - \mathcal{F}_{\text{train}}} p(f) \cdot g(f) \mathrm{d}f \right|
$$

$$
\leq \frac{1-Z}{Z} \int_{\mathcal{F}_{\text{train}}} p(f) \cdot |g(f)| \, \mathrm{d}f + \int_{\mathcal{F} - \mathcal{F}_{\text{train}}} p(f) \cdot |g(f)| \, \mathrm{d}f
$$

$$
\leq \frac{(1-Z) r_{\max}}{Z} \int_{\mathcal{F}_{\text{train}}} p(f) \mathrm{d}f + r_{\max} \int_{\mathcal{F} - \mathcal{F}_{\text{train}}} p(f) \mathrm{d}f
$$

$$
= \frac{(1-Z) r_{\max}}{Z} \cdot Z + r_{\max} \cdot (1-Z) = 2 r_{\max}(1-Z).
\tag{41}
$$

Now we are trying to bound the second term, which can be expressed as

$$
\left| \mathop{\mathbb{E}}_{\substack{f \sim p(\cdot) \\ s \sim d^{\tilde{\mu}_f}(\cdot) \\ a \sim \tilde{\mu}_f(\cdot|s)}} \left[ A^{\mu_f}(s,a) \right] - \mathop{\mathbb{E}}_{\substack{f \sim p_{\text{train}}(\cdot) \\ s \sim d^{\mu_f}(\cdot) \\ a \sim \tilde{\mu}_f(\cdot|s)}} \left[ A^{\mu_f}(s,a) \right] \right|
$$

$$
= \left| \mathop{\mathbb{E}}_{\substack{f \sim p(\cdot) \\ s \sim d^{\tilde{\mu}_f}(\cdot) \\ a \sim \tilde{\mu}_f(\cdot|s)}} \left[ A^{\mu_f}(s,a) \right] - \mathop{\mathbb{E}}_{\substack{f \sim p_{\text{train}}(\cdot) \\ s \sim d^{\tilde{\mu}_f}(\cdot) \\ a \sim \tilde{\mu}_f(\cdot|s)}} \left[ A^{\mu_f}(s,a) \right] + \mathop{\mathbb{E}}_{\substack{f \sim p_{\text{train}}(\cdot) \\ s \sim d^{\tilde{\mu}_f}(\cdot) \\ a \sim \tilde{\mu}_f(\cdot|s)}} \left[ A^{\mu_f}(s,a) \right] - \mathop{\mathbb{E}}_{\substack{f \sim p_{\text{train}}(\cdot) \\ s \sim d^{\mu_f}(\cdot) \\ a \sim \tilde{\mu}_f(\cdot|s)}} \left[ A^{\mu_f}(s,a) \right] \right|
$$

$$
\leq \underbrace{\left| \mathop{\mathbb{E}}_{\substack{f \sim p(\cdot) \\ s \sim d^{\tilde{\mu}_f}(\cdot) \\ a \sim \tilde{\mu}_f(\cdot|s)}} \left[ A^{\mu_f}(s,a) \right] - \mathop{\mathbb{E}}_{\substack{f \sim p_{\text{train}}(\cdot) \\ s \sim d^{\tilde{\mu}_f}(\cdot) \\ a \sim \tilde{\mu}_f(\cdot|s)}} \left[ A^{\mu_f}(s,a) \right] \right|}_{\text{denote as } \Phi} + \underbrace{\left| \mathop{\mathbb{E}}_{\substack{f \sim p_{\text{train}}(\cdot) \\ s \sim d^{\tilde{\mu}_f}(\cdot) \\ a \sim \tilde{\mu}_f(\cdot|s)}} \left[ A^{\mu_f}(s,a) \right] - \mathop{\mathbb{E}}_{\substack{f \sim p_{\text{train}}(\cdot) \\ s \sim d^{\mu_f}(\cdot) \\ a \sim \tilde{\mu}_f(\cdot|s)}} \left[ A^{\mu_f}(s,a) \right] \right|}_{\text{denote as } \Psi} .
$$

$$(42)$$

Using Corollary F.4, $\Psi$ can be bounded by

$$
\Psi = \left| \mathop{\mathbb{E}}_{f \sim p_{\text{train}}(\cdot)} \left\{ \mathop{\mathbb{E}}_{\substack{s \sim d^{\tilde{\mu}_f}(\cdot) \\ a \sim \tilde{\mu}_f(\cdot|s)}} \left[ A^{\mu_f}(s,a) \right] - \mathop{\mathbb{E}}_{\substack{s \sim d^{\mu_f}(\cdot) \\ a \sim \tilde{\mu}_f(\cdot|s)}} \left[ A^{\mu_f}(s,a) \right] \right\} \right|
$$

$$
\leq \mathop{\mathbb{E}}_{f \sim p_{\text{train}}(\cdot)} \left\{ \left| \mathop{\mathbb{E}}_{\substack{s \sim d^{\tilde{\mu}_f}(\cdot) \\ a \sim \tilde{\mu}_f(\cdot|s)}} \left[ A^{\mu_f}(s,a) \right] - \mathop{\mathbb{E}}_{\substack{s \sim d^{\mu_f}(\cdot) \\ a \sim \tilde{\mu}_f(\cdot|s)}} \left[ A^{\mu_f}(s,a) \right] \right| \right\} \tag{43}
$$

$$
\leq \mathop{\mathbb{E}}_{f \sim p_{\text{train}}(\cdot)} \left\{ \frac{2\epsilon\gamma}{1-\gamma} \mathop{\mathbb{E}}_{s \sim d^{\mu_f}(\cdot)} \left[ D_{\text{TV}}(\tilde{\mu}_f \| \mu_f)[s] \right] \right\},
$$

where $\epsilon = \max_s \left| \mathbb{E}_{a \sim \tilde{\mu}_f(\cdot|s)} \left[ A^{\mu_f}(s,a) \right] \right|$, denote $\epsilon_{\text{train}} = \max_{f \in \mathcal{F}_{\text{train}}} \{\epsilon\}$, we obtain

$$
\Psi \leq \frac{2\gamma\epsilon_{\text{train}}}{1-\gamma} \mathop{\mathbb{E}}_{\substack{f \sim p_{\text{train}}(\cdot) \\ s \sim d^{\mu_f}(\cdot)}} \left[ D_{\text{TV}}(\tilde{\mu}_f \| \mu_f)[s] \right]. \tag{44}
$$

Next, with a little abuse of notation $g(f)$, denote

$$
g(f) = \mathop{\mathbb{E}}_{\substack{s \sim d^{\tilde{\mu}_f}(\cdot) \\ a \sim \tilde{\mu}_f(\cdot|s)}} \left[ A^{\mu_f}(s,a) \right], \tag{45}
$$

we can rewrite $\Phi$ as

$$
\Phi = \left| \mathop{\mathbb{E}}_{f \sim p(\cdot)} \left[ g(f) \right] - \mathop{\mathbb{E}}_{f \sim p_{\text{train}}(\cdot)} \left[ g(f) \right] \right|, \tag{46}
$$

then, similar to (37), (38), (39) and (41),

$$
\Phi \leq \frac{1-Z}{Z} \int_{\mathcal{F}_{\text{train}}} p(f) \cdot |g(f)| \, df + \int_{\mathcal{F} - \mathcal{F}_{\text{train}}} p(f) \cdot |g(f)| \, df. \tag{47}
$$

According to Lemma F.3, we can bound $g(f)$, which can be expressed as

$$
g(f) = \mathop{\mathbb{E}}_{\substack{s \sim d^{\tilde{\mu}_f}(\cdot) \\ a \sim \tilde{\mu}_f(\cdot|s)}} \left[ A^{\mu_f}(s,a) \right] = \mathop{\mathbb{E}}_{s \sim d^{\tilde{\mu}_f}(\cdot)} \left\{ \mathop{\mathbb{E}}_{a \sim \tilde{\mu}_f(\cdot|s)} \left[ A^{\mu_f}(s,a) \right] \right\}, \tag{48}
$$

thus,

$$|g(f)| \leq \mathop{\mathbb{E}}_{s \sim d^{\tilde{\mu}_f}(\cdot)} \left\{ \left| \mathop{\mathbb{E}}_{a \sim \tilde{\mu}_f(\cdot|s)} [A^{\mu_f}(s,a)] \right| \right\} \leq \mathop{\mathbb{E}}_{s \sim d^{\tilde{\mu}_f}(\cdot)} \left\{ 2 D_{\mathrm{TV}}(\tilde{\mu}_f \| \mu_f)[s] \cdot \max_a |A^{\mu_f}(s,a)| \right\}. \tag{49}$$

Denote $\delta = \max_{s,a} |A^{\mu_f}(s,a)|$, then we have

$$|g(f)| \leq 2\delta \mathop{\mathbb{E}}_{s \sim d^{\tilde{\mu}_f}(\cdot)} [D_{\mathrm{TV}}(\tilde{\mu}_f \| \mu_f)[s]], \tag{50}$$

which means that

$$\begin{aligned}
\Phi \leq & \frac{1-Z}{Z} \int_{\mathcal{F}_{\mathrm{train}}} p(f) \cdot |g(f)| \, \mathrm{d}f + \int_{\mathcal{F} - \mathcal{F}_{\mathrm{train}}} p(f) \cdot |g(f)| \, \mathrm{d}f \\
\leq & \frac{2\delta_{\mathrm{train}}(1-Z)}{Z} \int_{\mathcal{F}_{\mathrm{train}}} p(f) \cdot \mathop{\mathbb{E}}_{s \sim d^{\tilde{\mu}_f}(\cdot)} [D_{\mathrm{TV}}(\tilde{\mu}_f \| \mu_f)[s]] \, \mathrm{d}f \\
& + 2\delta_{\mathrm{eval}} \int_{\mathcal{F} - \mathcal{F}_{\mathrm{train}}} p(f) \cdot \mathop{\mathbb{E}}_{s \sim d^{\tilde{\mu}_f}(\cdot)} [D_{\mathrm{TV}}(\tilde{\mu}_f \| \mu_f)[s]] \, \mathrm{d}f \\
= & 2\delta_{\mathrm{train}}(1-Z) \int_{\mathcal{F}_{\mathrm{train}}} \frac{p(f)}{Z} \cdot \mathop{\mathbb{E}}_{s \sim d^{\tilde{\mu}_f}(\cdot)} [D_{\mathrm{TV}}(\tilde{\mu}_f \| \mu_f)[s]] \, \mathrm{d}f \\
& + 2\delta_{\mathrm{eval}}(1-Z) \int_{\mathcal{F} - \mathcal{F}_{\mathrm{train}}} \frac{p(f)}{1-Z} \cdot \mathop{\mathbb{E}}_{s \sim d^{\tilde{\mu}_f}(\cdot)} [D_{\mathrm{TV}}(\tilde{\mu}_f \| \mu_f)[s]] \, \mathrm{d}f \\
= & 2\delta_{\mathrm{train}}(1-Z) \mathop{\mathbb{E}}_{\substack{f \sim p_{\mathrm{train}}(\cdot) \\ s \sim d^{\tilde{\mu}_f}(\cdot)}} [D_{\mathrm{TV}}(\tilde{\mu}_f \| \mu_f)[s]] + 2\delta_{\mathrm{eval}}(1-Z) \mathop{\mathbb{E}}_{\substack{f \sim p_{\mathrm{eval}}(\cdot) \\ s \sim d^{\tilde{\mu}_f}(\cdot)}} [D_{\mathrm{TV}}(\tilde{\mu}_f \| \mu_f)[s]],
\end{aligned} \tag{51}$$

where $\delta_{\mathrm{train}} = \max_{f \in \mathcal{F}_{\mathrm{train}}} \{\max_{s,a} |A^{\mu_f}(s,a)|\}$ and $\delta_{\mathrm{eval}} = \max_{f \in \mathcal{F}_{\mathrm{eval}}} \{\max_{s,a} |A^{\mu_f}(s,a)|\}$.

Finally, combining (35), (41), (42), (44), and (51), we have

$$\begin{aligned}
|\zeta(\tilde{\pi}) - L_\pi(\tilde{\pi})| \leq & \frac{2r_{\max}(1-Z)}{1-\gamma} + \frac{2\gamma\epsilon_{\mathrm{train}}}{(1-\gamma)^2} \mathop{\mathbb{E}}_{\substack{f \sim p_{\mathrm{train}}(\cdot) \\ s \sim d^{\mu_f}(\cdot)}} [D_{\mathrm{TV}}(\tilde{\mu}_f \| \mu_f)[s]] \\
& + \frac{2\delta_{\mathrm{train}}(1-Z)}{1-\gamma} \mathop{\mathbb{E}}_{\substack{f \sim p_{\mathrm{train}}(\cdot) \\ s \sim d^{\tilde{\mu}_f}(\cdot)}} [D_{\mathrm{TV}}(\tilde{\mu}_f \| \mu_f)[s]] + \frac{2\delta_{\mathrm{eval}}(1-Z)}{1-\gamma} \mathop{\mathbb{E}}_{\substack{f \sim p_{\mathrm{eval}}(\cdot) \\ s \sim d^{\tilde{\mu}_f}(\cdot)}} [D_{\mathrm{TV}}(\tilde{\mu}_f \| \mu_f)[s]],
\end{aligned} \tag{52}$$

thus, the generalization performance lower bound is

$$\begin{aligned}
\zeta(\tilde{\pi}) \geq & L_\pi(\tilde{\pi}) - \frac{2r_{\max}(1-Z)}{1-\gamma} - \frac{2\gamma\epsilon_{\mathrm{train}}}{(1-\gamma)^2} \mathop{\mathbb{E}}_{\substack{f \sim p_{\mathrm{train}}(\cdot) \\ s \sim d^{\mu_f}(\cdot)}} [D_{\mathrm{TV}}(\tilde{\mu}_f \| \mu_f)[s]] \\
& - \frac{2\delta_{\mathrm{train}}(1-Z)}{1-\gamma} \mathop{\mathbb{E}}_{\substack{f \sim p_{\mathrm{train}}(\cdot) \\ s \sim d^{\tilde{\mu}_f}(\cdot)}} [D_{\mathrm{TV}}(\tilde{\mu}_f \| \mu_f)[s]] - \frac{2\delta_{\mathrm{eval}}(1-Z)}{1-\gamma} \mathop{\mathbb{E}}_{\substack{f \sim p_{\mathrm{eval}}(\cdot) \\ s \sim d^{\tilde{\mu}_f}(\cdot)}} [D_{\mathrm{TV}}(\tilde{\mu}_f \| \mu_f)[s]],
\end{aligned} \tag{53}$$

concluding the proof. $\square$

### F.3 PROOF OF THEOREM 4.3

**Theorem 4.3.** *Given any two policies, $\tilde{\pi}$ and $\pi$, the following bound holds:*

$$\eta(\tilde{\pi}) \geq L_\pi(\tilde{\pi}) - \frac{2\gamma\epsilon_{\mathrm{train}}}{(1-\gamma)^2} \mathop{\mathbb{E}}_{\substack{f \sim p_{\mathrm{train}}(\cdot) \\ s \sim d^{\mu_f}(\cdot)}} [D_{\mathrm{TV}}(\tilde{\mu}_f \| \mu_f)[s]]. \tag{54}$$

*Proof.* Since

$$|\eta(\tilde{\pi}) - L_\pi(\tilde{\pi})| = \frac{1}{1-\gamma} \left| \mathop{\mathbb{E}}_{\substack{f\sim p_{\text{train}}(\cdot) \\ s\sim d^{\tilde{\mu}_f}(\cdot) \\ a\sim\tilde{\mu}_f(\cdot|s)}} [A^{\mu_f}(s,a)] - \mathop{\mathbb{E}}_{\substack{f\sim p_{\text{train}}(\cdot) \\ s\sim d^{\mu_f}(\cdot) \\ a\sim\tilde{\mu}_f(\cdot|s)}} [A^{\mu_f}(s,a)] \right| = \frac{\Psi}{1-\gamma} \tag{55}$$

$$\leq \frac{2\gamma\epsilon_{\text{train}}}{(1-\gamma)^2} \mathop{\mathbb{E}}_{\substack{f\sim p_{\text{train}}(\cdot) \\ s\sim d^{\mu_f}(\cdot)}} [D_{\text{TV}}(\tilde{\mu}_f\|\mu_f)[s]],$$

thus,

$$\eta(\tilde{\pi}) \geq L_\pi(\tilde{\pi}) - \frac{2\gamma\epsilon_{\text{train}}}{(1-\gamma)^2} \mathop{\mathbb{E}}_{\substack{f\sim p_{\text{train}}(\cdot) \\ s\sim d^{\mu_f}(\cdot)}} [D_{\text{TV}}(\tilde{\mu}_f\|\mu_f)[s]], \tag{56}$$

concluding the proof. □

### F.4 PROOF OF THEOREM 4.5

**Theorem 4.5.** *Given any two policies, $\tilde{\pi}$ and $\pi$, the following bound holds:*

$$\mathfrak{D}_1 \leq \left(1 + \frac{2\gamma\sigma_{\text{train}}}{1-\gamma}\right)\mathfrak{D}_{\text{train}}, \quad \mathfrak{D}_2 \leq \left(1 + \frac{2\gamma\sigma_{\text{eval}}}{1-\gamma}\right) \underbrace{\mathop{\mathbb{E}}_{\substack{f\sim p_{\text{eval}}(\cdot) \\ s\sim d^{\mu_f}(\cdot)}} [D_{\text{TV}}(\tilde{\mu}_f\|\mu_f)[s]]}_{\text{denote it as } \mathfrak{D}_{\text{eval}}}, \tag{57}$$

*where $\sigma_{\text{train}} = \max_{f\in\mathcal{F}_{\text{train}}}\{D_{\text{TV}}^{\max}(\tilde{\mu}_f\|\mu_f)[s]\}$ and $\sigma_{\text{eval}} = \max_{f\in\mathcal{F}_{\text{eval}}}\{D_{\text{TV}}^{\max}(\tilde{\mu}_f\|\mu_f)[s]\}$, $D_{\text{TV}}^{\max}(\tilde{\mu}_f\|\mu_f)[s]$ represents $\max_s D_{\text{TV}}(\tilde{\mu}_f\|\mu_f)[s]$.*

*Proof.* According to Lemma F.2, we have

$$|\mathfrak{D}_1 - \mathfrak{D}_{\text{train}}| = \left| \mathop{\mathbb{E}}_{\substack{f\sim p_{\text{train}}(\cdot) \\ s\sim d^{\tilde{\mu}_f}(\cdot)}} [D_{\text{TV}}(\tilde{\mu}_f\|\mu_f)[s]] - \mathop{\mathbb{E}}_{\substack{f\sim p_{\text{train}}(\cdot) \\ s\sim d^{\mu_f}(\cdot)}} [D_{\text{TV}}(\tilde{\mu}_f\|\mu_f)[s]] \right|$$

$$= \left| \mathop{\mathbb{E}}_{f\sim p_{\text{train}}(\cdot)} \left\{ \mathop{\mathbb{E}}_{s\sim d^{\tilde{\mu}_f}(\cdot)} [D_{\text{TV}}(\tilde{\mu}_f\|\mu_f)[s]] - \mathop{\mathbb{E}}_{s\sim d^{\mu_f}(\cdot)} [D_{\text{TV}}(\tilde{\mu}_f\|\mu_f)[s]] \right\} \right|$$

$$\leq \mathop{\mathbb{E}}_{f\sim p_{\text{train}}(\cdot)} \left\{ \left| \mathop{\mathbb{E}}_{s\sim d^{\tilde{\mu}_f}(\cdot)} [D_{\text{TV}}(\tilde{\mu}_f\|\mu_f)[s]] - \mathop{\mathbb{E}}_{s\sim d^{\mu_f}(\cdot)} [D_{\text{TV}}(\tilde{\mu}_f\|\mu_f)[s]] \right| \right\} \tag{58}$$

$$\leq \mathop{\mathbb{E}}_{f\sim p_{\text{train}}(\cdot)} \left\{ \left\|d^{\tilde{\mu}_f} - d^{\mu_f}\right\|_1 \cdot \left\|D_{\text{TV}}(\tilde{\mu}_f\|\mu_f)[s]\right\|_\infty \right\}$$

$$\leq \mathop{\mathbb{E}}_{f\sim p_{\text{train}}(\cdot)} \left\{ \frac{2\gamma}{1-\gamma} \mathop{\mathbb{E}}_{s\sim d^{\mu_f}(\cdot)} [D_{\text{TV}}(\tilde{\mu}_f\|\mu_f)[s]] \cdot \max_s D_{\text{TV}}(\tilde{\mu}_f\|\mu_f)[s] \right\}$$

$$\leq \frac{2\gamma\sigma_{\text{train}}}{1-\gamma} \mathop{\mathbb{E}}_{\substack{f\sim p_{\text{train}}(\cdot) \\ s\sim d^{\mu_f}(\cdot)}} [D_{\text{TV}}(\tilde{\mu}_f\|\mu_f)[s]] = \frac{2\gamma\sigma_{\text{train}}}{1-\gamma} \cdot \mathfrak{D}_{\text{train}},$$

as a result,

$$\mathfrak{D}_1 \leq \left(1 + \frac{2\gamma\sigma_{\text{train}}}{1-\gamma}\right)\mathfrak{D}_{\text{train}}. \tag{59}$$

Similarly, using Lemma F.2 again, we have

$$
\begin{aligned}
|\mathfrak{D}_2 - \mathfrak{D}_{\mathrm{eval}}| &= \left| \mathop{\mathbb{E}}_{\substack{f \sim p_{\mathrm{eval}}(\cdot) \\ s \sim d^{\tilde{\mu}_f}(\cdot)}} [D_{\mathrm{TV}}(\tilde{\mu}_f \| \mu_f)[s]] - \mathop{\mathbb{E}}_{\substack{f \sim p_{\mathrm{eval}}(\cdot) \\ s \sim d^{\mu_f}(\cdot)}} [D_{\mathrm{TV}}(\tilde{\mu}_f \| \mu_f)[s]] \right| \\
&= \left| \mathop{\mathbb{E}}_{f \sim p_{\mathrm{eval}}(\cdot)} \left\{ \mathop{\mathbb{E}}_{s \sim d^{\tilde{\mu}_f}(\cdot)} [D_{\mathrm{TV}}(\tilde{\mu}_f \| \mu_f)[s]] - \mathop{\mathbb{E}}_{s \sim d^{\mu_f}(\cdot)} [D_{\mathrm{TV}}(\tilde{\mu}_f \| \mu_f)[s]] \right\} \right| \\
&\leq \mathop{\mathbb{E}}_{f \sim p_{\mathrm{eval}}(\cdot)} \left\{ \left| \mathop{\mathbb{E}}_{s \sim d^{\tilde{\mu}_f}(\cdot)} [D_{\mathrm{TV}}(\tilde{\mu}_f \| \mu_f)[s]] - \mathop{\mathbb{E}}_{s \sim d^{\mu_f}(\cdot)} [D_{\mathrm{TV}}(\tilde{\mu}_f \| \mu_f)[s]] \right| \right\} \qquad (60) \\
&\leq \mathop{\mathbb{E}}_{f \sim p_{\mathrm{eval}}(\cdot)} \left\{ \left\| d^{\tilde{\mu}_f} - d^{\mu_f} \right\|_1 \cdot \left\| D_{\mathrm{TV}}(\tilde{\mu}_f \| \mu_f)[s] \right\|_\infty \right\} \\
&\leq \mathop{\mathbb{E}}_{f \sim p_{\mathrm{eval}}(\cdot)} \left\{ \frac{2\gamma}{1-\gamma} \mathop{\mathbb{E}}_{s \sim d^{\mu_f}(\cdot)} [D_{\mathrm{TV}}(\tilde{\mu}_f \| \mu_f)[s]] \cdot \max_s D_{\mathrm{TV}}(\tilde{\mu}_f \| \mu_f)[s] \right\} \\
&\leq \frac{2\gamma \sigma_{\mathrm{eval}}}{1-\gamma} \mathop{\mathbb{E}}_{\substack{f \sim p_{\mathrm{eval}}(\cdot) \\ s \sim d^{\mu_f}(\cdot)}} [D_{\mathrm{TV}}(\tilde{\mu}_f \| \mu_f)[s]] = \frac{2\gamma \sigma_{\mathrm{eval}}}{1-\gamma} \cdot \mathfrak{D}_{\mathrm{eval}},
\end{aligned}
$$

as a result,

$$
\mathfrak{D}_2 \leq \left( 1 + \frac{2\gamma \sigma_{\mathrm{eval}}}{1-\gamma} \right) \mathfrak{D}_{\mathrm{eval}}, \qquad (61)
$$

concluding the proof. $\qquad \square$

### F.5 PROOF OF THEOREM 4.7

**Theorem 4.7.** *Given any two policies, $\tilde{\pi}$ and $\pi$, assume that $\tilde{\pi}$ is $\mathcal{R}_{\tilde{\pi}}$-robust, and $\pi$ is $\mathcal{R}_\pi$-robust, then the following bound holds:*

$$
\mathfrak{D}_{\mathrm{eval}} \leq \left( 1 + \frac{2\gamma \sigma_{\mathrm{train}}}{1-\gamma} \right) \mathcal{R}_\pi + \mathcal{R}_{\tilde{\pi}} + \mathfrak{D}_{\mathrm{train}}. \qquad (62)
$$

*Proof.* Let's first rewrite $\mathfrak{D}_{\mathrm{eval}}$ as

$$
\mathfrak{D}_{\mathrm{eval}} = \mathop{\mathbb{E}}_{\substack{\tilde{f} \sim p_{\mathrm{eval}}(\cdot) \\ s \sim d^{\mu}\tilde{f}(\cdot)}} \left[ D_{\mathrm{TV}}(\tilde{\mu}_{\tilde{f}} \| \mu_{\tilde{f}})[s] \right]. \qquad (63)
$$

For another $f \in \mathcal{F}_{\mathrm{train}}$, by repeatedly using the triangle inequality of the TV distance, we have

$$
\begin{aligned}
\mathfrak{D}_{\mathrm{eval}} &= \mathop{\mathbb{E}}_{\substack{\tilde{f} \sim p_{\mathrm{eval}}(\cdot) \\ s \sim d^{\mu}\tilde{f}(\cdot)}} \left[ D_{\mathrm{TV}}(\tilde{\mu}_{\tilde{f}} \| \mu_{\tilde{f}})[s] \right] \\
&\leq \mathop{\mathbb{E}}_{\substack{\tilde{f} \sim p_{\mathrm{eval}}(\cdot) \\ s \sim d^{\mu}\tilde{f}(\cdot)}} \left[ D_{\mathrm{TV}}(\tilde{\mu}_{\tilde{f}} \| \tilde{\mu}_f)[s] + D_{\mathrm{TV}}(\tilde{\mu}_f \| \mu_f)[s] + D_{\mathrm{TV}}(\mu_f \| \mu_{\tilde{f}})[s] \right] \\
&= \mathop{\mathbb{E}}_{\substack{\tilde{f} \sim p_{\mathrm{eval}}(\cdot) \\ s \sim d^{\mu}\tilde{f}(\cdot)}} \left[ D_{\mathrm{TV}}(\tilde{\mu}_{\tilde{f}} \| \tilde{\mu}_f)[s] \right] + \mathop{\mathbb{E}}_{\substack{\tilde{f} \sim p_{\mathrm{eval}}(\cdot) \\ s \sim d^{\mu}\tilde{f}(\cdot)}} [D_{\mathrm{TV}}(\tilde{\mu}_f \| \mu_f)[s]] + \mathop{\mathbb{E}}_{\substack{\tilde{f} \sim p_{\mathrm{eval}}(\cdot) \\ s \sim d^{\mu}\tilde{f}(\cdot)}} \left[ D_{\mathrm{TV}}(\mu_f \| \mu_{\tilde{f}})[s] \right],
\end{aligned}
$$
$$
(64)
$$

taking the expectation of both sides of the inequality with respect to $f \sim p_{\mathrm{train}}(\cdot)$, we obtain

$$
\mathop{\mathbb{E}}_{f \sim p_{\mathrm{train}}(\cdot)} [\mathfrak{D}_{\mathrm{eval}}] \leq \mathop{\mathbb{E}}_{\substack{f \sim p_{\mathrm{train}}(\cdot) \\ \tilde{f} \sim p_{\mathrm{eval}}(\cdot) \\ s \sim d^{\mu}\tilde{f}(\cdot)}} \left[ D_{\mathrm{TV}}(\tilde{\mu}_{\tilde{f}} \| \tilde{\mu}_f)[s] \right] + \mathop{\mathbb{E}}_{\substack{f \sim p_{\mathrm{train}}(\cdot) \\ \tilde{f} \sim p_{\mathrm{eval}}(\cdot) \\ s \sim d^{\mu}\tilde{f}(\cdot)}} [D_{\mathrm{TV}}(\tilde{\mu}_f \| \mu_f)[s]] + \mathop{\mathbb{E}}_{\substack{f \sim p_{\mathrm{train}}(\cdot) \\ \tilde{f} \sim p_{\mathrm{eval}}(\cdot) \\ s \sim d^{\mu}\tilde{f}(\cdot)}} \left[ D_{\mathrm{TV}}(\mu_f \| \mu_{\tilde{f}})[s] \right].
$$
$$
(65)
$$

Since $\mathfrak{D}_{\text{eval}}$ is independent of $f$, it becomes a constant after taking the expectation, which is

$$\mathfrak{D}_{\text{eval}} \leq \mathop{\mathbb{E}}_{\substack{f\sim p_{\text{train}}(\cdot) \\ \tilde{f}\sim p_{\text{eval}}(\cdot) \\ s\sim d^{\mu_{\tilde{f}}}(\cdot)}} \left[ D_{\text{TV}}(\tilde{\mu}_{\tilde{f}}\|\tilde{\mu}_f)[s] \right] + \mathop{\mathbb{E}}_{\substack{f\sim p_{\text{train}}(\cdot) \\ \tilde{f}\sim p_{\text{eval}}(\cdot) \\ s\sim d^{\mu_{\tilde{f}}}(\cdot)}} \left[ D_{\text{TV}}(\tilde{\mu}_f\|\mu_f)[s] \right] + \mathop{\mathbb{E}}_{\substack{f\sim p_{\text{train}}(\cdot) \\ \tilde{f}\sim p_{\text{eval}}(\cdot) \\ s\sim d^{\mu_{\tilde{f}}}(\cdot)}} \left[ D_{\text{TV}}(\mu_f\|\mu_{\tilde{f}})[s] \right].$$

$$(66)$$

Note that $\tilde{\pi}$ is $\mathcal{R}_{\tilde{\pi}}$-robust, and $\pi$ is $\mathcal{R}_{\pi}$-robust, we can thus bound the first term:

$$\mathop{\mathbb{E}}_{\substack{f\sim p_{\text{train}}(\cdot) \\ \tilde{f}\sim p_{\text{eval}}(\cdot) \\ s\sim d^{\mu_{\tilde{f}}}(\cdot)}} \left[ D_{\text{TV}}(\tilde{\mu}_{\tilde{f}}\|\tilde{\mu}_f)[s] \right] = \mathop{\mathbb{E}}_{\substack{f\sim p_{\text{train}}(\cdot) \\ \tilde{f}\sim p_{\text{eval}}(\cdot)}} \left[ \sum_{s\in\mathcal{S}} d^{\mu_{\tilde{f}}}(s) \cdot D_{\text{TV}}(\tilde{\mu}_{\tilde{f}}\|\tilde{\mu}_f)[s] \right]$$

$$\leq \mathop{\mathbb{E}}_{\substack{f\sim p_{\text{train}}(\cdot) \\ \tilde{f}\sim p_{\text{eval}}(\cdot)}} \left[ \sum_{s\in\mathcal{S}} d^{\mu_{\tilde{f}}}(s) \cdot \mathcal{R}_{\tilde{\pi}} \right] = \mathcal{R}_{\tilde{\pi}} \mathop{\mathbb{E}}_{\substack{f\sim p_{\text{train}}(\cdot) \\ \tilde{f}\sim p_{\text{eval}}(\cdot)}} \left[ \sum_{s\in\mathcal{S}} d^{\mu_{\tilde{f}}}(s) \right] = \mathcal{R}_{\tilde{\pi}}.$$

$$(67)$$

Similarly, we can bound the third term:

$$\mathop{\mathbb{E}}_{\substack{f\sim p_{\text{train}}(\cdot) \\ \tilde{f}\sim p_{\text{eval}}(\cdot) \\ s\sim d^{\mu_{\tilde{f}}}(\cdot)}} \left[ D_{\text{TV}}(\mu_{\tilde{f}}\|\mu_f)[s] \right] = \mathop{\mathbb{E}}_{\substack{f\sim p_{\text{train}}(\cdot) \\ \tilde{f}\sim p_{\text{eval}}(\cdot)}} \left[ \sum_{s\in\mathcal{S}} d^{\mu_{\tilde{f}}}(s) \cdot D_{\text{TV}}(\mu_{\tilde{f}}\|\mu_f)[s] \right]$$

$$\leq \mathop{\mathbb{E}}_{\substack{f\sim p_{\text{train}}(\cdot) \\ \tilde{f}\sim p_{\text{eval}}(\cdot)}} \left[ \sum_{s\in\mathcal{S}} d^{\mu_{\tilde{f}}}(s) \cdot \mathcal{R}_{\pi} \right] = \mathcal{R}_{\pi} \mathop{\mathbb{E}}_{\substack{f\sim p_{\text{train}}(\cdot) \\ \tilde{f}\sim p_{\text{eval}}(\cdot)}} \left[ \sum_{s\in\mathcal{S}} d^{\mu_{\tilde{f}}}(s) \right] = \mathcal{R}_{\pi}.$$

$$(68)$$

Next, we are trying to bound the second term, which is similar to $\mathfrak{D}_{\text{train}}$. Note that $\mathfrak{D}_{\text{train}}$ is independent of $\tilde{f}$, we can thus rewrite it as

$$\mathfrak{D}_{\text{train}} = \mathop{\mathbb{E}}_{\substack{f\sim p_{\text{train}}(\cdot) \\ s\sim d^{\mu_f}(\cdot)}} \left[ D_{\text{TV}}(\tilde{\mu}_f\|\mu_f)[s] \right] = \mathop{\mathbb{E}}_{\substack{f\sim p_{\text{train}}(\cdot) \\ \tilde{f}\sim p_{\text{eval}}(\cdot) \\ s\sim d^{\mu_f}(\cdot)}} \left[ D_{\text{TV}}(\tilde{\mu}_f\|\mu_f)[s] \right], \qquad (69)$$

then

$$\left| \mathop{\mathbb{E}}_{\substack{f\sim p_{\text{train}}(\cdot) \\ \tilde{f}\sim p_{\text{eval}}(\cdot) \\ s\sim d^{\mu_{\tilde{f}}}(\cdot)}} \left[ D_{\text{TV}}(\tilde{\mu}_f\|\mu_f)[s] \right] - \mathfrak{D}_{\text{train}} \right|$$

$$= \left| \mathop{\mathbb{E}}_{\substack{f\sim p_{\text{train}}(\cdot) \\ \tilde{f}\sim p_{\text{eval}}(\cdot) \\ s\sim d^{\mu_{\tilde{f}}}(\cdot)}} \left[ D_{\text{TV}}(\tilde{\mu}_f\|\mu_f)[s] \right] - \mathop{\mathbb{E}}_{\substack{f\sim p_{\text{train}}(\cdot) \\ \tilde{f}\sim p_{\text{eval}}(\cdot) \\ s\sim d^{\mu_f}(\cdot)}} \left[ D_{\text{TV}}(\tilde{\mu}_f\|\mu_f)[s] \right] \right|$$

$$= \left| \int_{\mathcal{F}_{\text{train}}} p_{\text{train}}(f) \int_{\mathcal{F}_{\text{eval}}} p_{\text{eval}}(\tilde{f}) \left\{ \mathop{\mathbb{E}}_{s\sim d^{\mu_{\tilde{f}}}(\cdot)} \left[ D_{\text{TV}}(\tilde{\mu}_f\|\mu_f)[s] \right] - \mathop{\mathbb{E}}_{s\sim d^{\mu_f}(\cdot)} \left[ D_{\text{TV}}(\tilde{\mu}_f\|\mu_f)[s] \right] \right\} \mathrm{d}\tilde{f}\mathrm{d}f \right|$$

$$\leq \int_{\mathcal{F}_{\text{train}}} p_{\text{train}}(f) \int_{\mathcal{F}_{\text{eval}}} p_{\text{eval}}(\tilde{f}) \left\{ \left| \mathop{\mathbb{E}}_{s\sim d^{\mu_{\tilde{f}}}(\cdot)} \left[ D_{\text{TV}}(\tilde{\mu}_f\|\mu_f)[s] \right] - \mathop{\mathbb{E}}_{s\sim d^{\mu_f}(\cdot)} \left[ D_{\text{TV}}(\tilde{\mu}_f\|\mu_f)[s] \right] \right| \right\} \mathrm{d}\tilde{f}\mathrm{d}f.$$

$$(70)$$

Note that,

$$\left| \mathop{\mathbb{E}}_{s\sim d^{\mu_{\tilde{f}}}(\cdot)} \left[ D_{\text{TV}}(\tilde{\mu}_f\|\mu_f)[s] \right] - \mathop{\mathbb{E}}_{s\sim d^{\mu_f}(\cdot)} \left[ D_{\text{TV}}(\tilde{\mu}_f\|\mu_f)[s] \right] \right| \leq \left\| d^{\mu_{\tilde{f}}} - d^{\mu_f} \right\|_1 \cdot \left\| D_{\text{TV}}(\tilde{\mu}_f\|\mu_f)[s] \right\|_\infty.$$

$$(71)$$

According to Lemma F.2,

$$\|d^{\mu_{\tilde{f}}} - d^{\mu_f}\|_1 \le \frac{2\gamma}{1-\gamma} \mathop{\mathbb{E}}_{s \sim d^{\mu_f}(\cdot)} \left[ D_{\mathrm{TV}}(\mu_{\tilde{f}} \| \mu_f)[s] \right], \tag{72}$$

$\pi$ is $\mathcal{R}_\pi$-robust, so,

$$\|d^{\mu_{\tilde{f}}} - d^{\mu_f}\|_1 \le \frac{2\gamma}{1-\gamma} \mathop{\mathbb{E}}_{s \sim d^{\mu_f}(\cdot)} \left[ D_{\mathrm{TV}}(\mu_{\tilde{f}} \| \mu_f)[s] \right] = \frac{2\gamma}{1-\gamma} \sum_{s \in \mathcal{S}} d^{\mu_f}(s) \cdot D_{\mathrm{TV}}(\mu_{\tilde{f}} \| \mu_f)[s] \le \frac{2\gamma}{1-\gamma} \mathcal{R}_\pi. \tag{73}$$

As a result,

$$\left| \mathop{\mathbb{E}}_{\substack{f \sim p_{\mathrm{train}}(\cdot) \\ \tilde{f} \sim p_{\mathrm{eval}}(\cdot) \\ s \sim d^{\mu_{\tilde{f}}}(\cdot)}} [D_{\mathrm{TV}}(\tilde{\mu}_f \| \mu_f)[s]] - \mathfrak{D}_{\mathrm{train}} \right|$$

$$\le \int_{\mathcal{F}_{\mathrm{train}}} p_{\mathrm{train}}(f) \int_{\mathcal{F}_{\mathrm{eval}}} p_{\mathrm{eval}}(\tilde{f}) \cdot \left\{ \left| \mathop{\mathbb{E}}_{s \sim d^{\mu_{\tilde{f}}}(\cdot)} [D_{\mathrm{TV}}(\tilde{\mu}_f \| \mu_f)[s]] - \mathop{\mathbb{E}}_{s \sim d^{\mu_f}(\cdot)} [D_{\mathrm{TV}}(\tilde{\mu}_f \| \mu_f)[s]] \right| \right\} \mathrm{d}\tilde{f} \mathrm{d}f$$

$$\le \int_{\mathcal{F}_{\mathrm{train}}} p_{\mathrm{train}}(f) \int_{\mathcal{F}_{\mathrm{eval}}} p_{\mathrm{eval}}(\tilde{f}) \cdot \left\{ \frac{2\gamma}{1-\gamma} \mathcal{R}_\pi \cdot \max_s D_{\mathrm{TV}}(\tilde{\mu}_f \| \mu_f)[s] \right\} \mathrm{d}\tilde{f} \mathrm{d}f$$

$$= \int_{\mathcal{F}_{\mathrm{train}}} p_{\mathrm{train}}(f) \cdot \left\{ \frac{2\gamma}{1-\gamma} \mathcal{R}_\pi \cdot \max_s D_{\mathrm{TV}}(\tilde{\mu}_f \| \mu_f)[s] \right\} \cdot \int_{\mathcal{F}_{\mathrm{eval}}} p_{\mathrm{eval}}(\tilde{f}) \mathrm{d}\tilde{f} \mathrm{d}f$$

$$= \int_{\mathcal{F}_{\mathrm{train}}} p_{\mathrm{train}}(f) \cdot \left\{ \frac{2\gamma}{1-\gamma} \mathcal{R}_\pi \cdot \max_s D_{\mathrm{TV}}(\tilde{\mu}_f \| \mu_f)[s] \right\} \mathrm{d}f = \frac{2\gamma}{1-\gamma} \mathcal{R}_\pi \int_{\mathcal{F}_{\mathrm{train}}} p_{\mathrm{train}}(f) \cdot \max_s D_{\mathrm{TV}}(\tilde{\mu}_f \| \mu_f)[s] \mathrm{d}f. \tag{74}$$

We previously defined $\sigma_{\mathrm{train}} = \max_{f \in \mathcal{F}_{\mathrm{train}}} \{\max_s D_{\mathrm{TV}}(\tilde{\mu}_f \| \mu_f)[s]\}$, so that

$$\left| \mathop{\mathbb{E}}_{\substack{f \sim p_{\mathrm{train}}(\cdot) \\ \tilde{f} \sim p_{\mathrm{eval}}(\cdot) \\ s \sim d^{\mu_{\tilde{f}}}(\cdot)}} [D_{\mathrm{TV}}(\tilde{\mu}_f \| \mu_f)[s]] - \mathfrak{D}_{\mathrm{train}} \right| \le \frac{2\gamma}{1-\gamma} \mathcal{R}_\pi \int_{\mathcal{F}_{\mathrm{train}}} p_{\mathrm{train}}(f) \cdot \max_s D_{\mathrm{TV}}(\tilde{\mu}_f \| \mu_f)[s] \mathrm{d}f$$

$$\le \frac{2\gamma\sigma_{\mathrm{train}}}{1-\gamma} \mathcal{R}_\pi \int_{\mathcal{F}_{\mathrm{train}}} p_{\mathrm{train}}(f) \mathrm{d}f = \frac{2\gamma\sigma_{\mathrm{train}}}{1-\gamma} \mathcal{R}_\pi, \tag{75}$$

thus, the second term is bounded by

$$\mathop{\mathbb{E}}_{\substack{f \sim p_{\mathrm{train}}(\cdot) \\ \tilde{f} \sim p_{\mathrm{eval}}(\cdot) \\ s \sim d^{\mu_{\tilde{f}}}(\cdot)}} [D_{\mathrm{TV}}(\tilde{\mu}_f \| \mu_f)[s]] \le \frac{2\gamma\sigma_{\mathrm{train}}}{1-\gamma} \mathcal{R}_\pi + \mathfrak{D}_{\mathrm{train}}. \tag{76}$$

Finally, combining (67), (68) and (76), we have

$$\mathfrak{D}_{\mathrm{eval}} \le \left( 1 + \frac{2\gamma\sigma_{\mathrm{train}}}{1-\gamma} \right) \mathcal{R}_\pi + \mathcal{R}_{\tilde{\pi}} + \mathfrak{D}_{\mathrm{train}}, \tag{77}$$

concluding the proof. $\qquad\square$

