# OpenReview forum: "Representation Convergence: Mutual Distillation is Secretly a Form of Regularization"
_ICLR.cc/2026/Conference — ICLR 2026 Conference Withdrawn Submission_

### Official Review · Reviewer_kbQ7 · 2025-10-27

**Soundness:** 3
**Presentation:** 2
**Contribution:** 2
**Rating:** 4
**Confidence:** 4

**Summary:**

This paper studies generalization in reinforcement learning by examining the robustness of learned features. It proposes (1) a theoretical framework showing how policies sensitive to “rendering” features that do not generalizable suffer from worse performance in expected returns and (2) the conjecture that mutual distillation between two agents regularizes representation learning toward such robust, invariant features and thus improves generalization.

The paper derives bounds relating expected test returns to terms measuring representation robustness and train/test differences, and empirically investigates whether mutual distillation indeed yields more robust policies on Procgen environments. The empirical results exhibit strong performance gains of the mutual distillation variant over PPO and several controls.

**Strengths:**

- The paper is easy to follow for the most part and has a clear motivated given a widespread interest in generalization and representation learning in RL.
- exploring generalization in RL through distillation is well-motivated and interesting. As far as I know, this technique is not exhaustively researched / understood in the context of RL.
- The formalization of rendering families and the decomposition of generalization error into robustness and train-test divergence components is intuitive and provides a neat mathematical framing. I believe this framing ends being equivalent to classical POMDP definitions, so it may be worth to align language (e.g., rendering function - emission function / observation function).
- MDPO performs consistently better than PPO and other baselines on Procgen tasks, even when controlling for model size or training budget (at least partly).

**Weaknesses:**

**Theoretical novelty and completeness**:

I found that the novelty of the theoretical exposition is overstated in serveral places. One of the main theorems (Theorem 3.3) mirrors the first-order performance difference bound from Schulman et al. (2015, TRPO) so closely that I believe it should be cited as a known existing bound.
The paper furthermore claims to be the first to “ first to provide a rigorous proof of {the} intuition {that robustness to irrelevant features enhances generalization performance}", which is an overstatement in this phrasing in my view. Several prior works in the literature provide generalization (some as part of sample-complexity) bounds explicitly dependent on representation adequacy. A non-exhaustive list:

- kernel RL and kernel complexity as a measure of representation capacity: Yeh et al. (2023)
- from a causality perspective: Kallus et al. (2020), Suau et al. (2023)
- from a symmetry and invariance perspective: Weltevrede et al. (2025)

The latter of the above even considers distillation for deep RL policies and should most certaintly be discussed in this work.

**Disconnect between theory and practice**
The biggest weakness of this paper in my view is the strong disparity between the derived theoretical claims and the conjectured effect of mutual distillation. The argument or intuition for how mutual distillation learns more robust features is very handwavy. For example it is not obvious at all that policies from different intialization or different batch orders or other factors will learn significantly different features. The work above indeed does show a related effect theoretically using neural tangent kernel theory (Weltevrede et al.). Knowing that feature learning is difficult to treat theoretically in deep learning I would suggest an alternative path would be to support the argument more empirically. For example with experiments showing indeed that different initialization / different exploration seeds / different batch orders etc. lead to feature learning of distinct spurious features which can then be eliminated through mutual distillation.

**Presentation**
The paper has several issues with presentation and clariity. Apart from several writing / language issues, the are also a number of ambiguities:
- Inconsistency in robustness metric. The robustness term is defined as a maximum total variation distance max TV distance, yet Table 1 lists values larger than 1.
- Algorithm 1 implies two agents collecting data separately; it is unclear whether the 50M total steps refer to per agent or shared. The appendix does not explicitly resolve this. Without this clarification, comparisons to PPO may reflect more total experience rather than representational effects.
- Adversarial/robustness experiments underspecified. The procedure for generating “adversarial renderings” (random CNNs) lacks some detail, and, importantly, motivation. I find this a very perculiar way of generating adversarial examples, could the authors elaborate this choice?
- some quantities (e.g., $L_\pi$) is not defined in such a way that the paper is self-sufficient. One should not need to refer to TRPO to recall its definition.
- Despite being in the title, the term "representation convergence" is never formally defined. I'm not following what this term aims to imply, what converges to what?


- Yeh, Sing-Yuan, et al. "Sample complexity of kernel-based q-learning." International Conference on Artificial Intelligence and Statistics. PMLR, 2023.
- Kallus, Nathan, and Angela Zhou. "Confounding-robust policy evaluation in infinite-horizon reinforcement learning." Advances in neural information processing systems 33 (2020): 22293-22304.
- Suau, Miguel, Matthijs TJ Spaan, and Frans A. Oliehoek. "Bad habits: Policy confounding and out-of-trajectory generalization in RL." arXiv preprint arXiv:2306.02419 (2023).
- Weltevrede, Max, et al. "How Ensembles of Distilled Policies Improve Generalisation in Reinforcement Learning." arXiv preprint arXiv:2505.16581 (2025).

**Questions:**

- can you provide a theoretical statement (e.g., in linearized form) showing that mutual distillation dynamics reduce the $R$-robustness term?
- how do you define the “representation convergence” concept?
- do the mutual distillation learners together use the same amount of samples as PPO baselines? If not, how is data shared or counted?
- could you describe the random CNNs used to generate alternate renderings and the motivation behind it?

---

> ### Author Response · Authors · 2025-11-26
>
> Dear Reviewer kbQ7, thank you for your detailed feedback on our paper. We believe it is valuable for improving the readability and quality of our work.
>
> We are especially delighted to receive your high praise:
> - _Exploring generalization in RL through distillation is well-motivated and interesting._
> - _The formalization of rendering families and the decomposition of generalization error into robustness and train-test divergence components is intuitive and provides a neat mathematical framing._
> - _MDPO performs consistently better than PPO and other baselines on Procgen tasks, even when controlling for model size or training budget._
>
> We will provide a comprehensive response to your comments below.
>
> >I found that the novelty of the theoretical exposition is overstated in serveral places. One of the main theorems (Theorem 3.3) mirrors the first-order performance difference bound from TRPO [1] so closely that I believe it should be cited as a known existing bound.
>
> >some quantities (e.g., $L_\pi$) is not defined in such a way that the paper is self-sufficient. One should not need to refer to TRPO to recall its definition.
>
> Thank you for your feedback. However, we respectfully disagree with your point. First, we **did** cite TRPO [1] when introducing the first-order approximation term $L_{\pi}$ before Theorem 3.3 (line 163). Second, although Theorem 3.3 is indeed related to TRPO, the lower bound in TRPO involves the **maximum KL divergence** over all states $s$, whereas our bound relies on the **TV divergence expectation** over states $s\sim d^{\mu_f}(\cdot)$, and we additionally introduce a rendering function $f\in\mathcal{F}$. Therefore, directly citing it as the original TRPO bound is not appropriate. Finally, Theorems 3.4, 3.5, 3.7, as well as Definition 3.6 and Corollary 3.8, constitute the main theoretical contributions of our paper, especially Corollary 3.8.
>
> Thank you for your suggestion regarding writing clarity. We will provide the definition of $L_{\pi}$ when introducing Theorem 3.3 in the revised PDF. In the original text, its explicit form is given in the proof of Theorem 3.4 in the appendix, equation **(34)**.

---

> > ### Author Response · Authors · 2025-11-26
> >
> > >The paper furthermore claims to be the first to ''first to provide a rigorous proof of {the} intuition {that robustness to irrelevant features enhances generalization performance'', which is an overstatement in this phrasing in my view. Several prior works [2, 3, 4, 5] in the literature provide generalization (some as part of sample-complexity) bounds explicitly dependent on representation adequacy.
> >
> > >[5] even considers distillation for deep RL policies and should most certaintly be discussed in this work.
> >
> > Thank you for your comments. We have summarized the key contributions of the related works you listed as follows:
> >
> > | Paper | Key Contribution |
> > |-------|----------------|
> > | [2] | Proposed a kernel-based Q-learning method, and provided finite sample complexity bounds |
> > | [3] | Proposed robust off-policy evaluation in infinite-horizon MDPs, providing upper and lower bounds on policy performance |
> > | [4] | Defined ''policy confounding'' and analyzed how narrow training trajectory distributions hurt generalization |
> > | [5] | Proposed ensembles of distilled policies, gave theoretical generalization bounds, and showed more diverse data and ensemble can improve generalization |
> >
> > However, we note that [2] primarily studies sample complexity, and [3] mainly introduces a robust policy evaluation method. We believe these works do not overlap with our main theoretical result, namely that ''policy robustness helps generalization.'' These works do not directly demonstrate how policy robustness affects generalization performance.
> >
> > Works [4] and [5] are more closely related to ours. However, [4] formalizes the problem mainly from a representation learning perspective, i.e., defining the state representation $\Phi:\mathcal{S}\mapsto\bar{\mathcal{S}}$ in Definition 3, which inherently **decouples** the encoder from the policy, whereas our theoretical framework is theoretically and empirically **end-to-end**.
> >
> > Moreover, [5] is indeed related to our work through the proposal of a policy distillation ensemble method, but its core motivation is to improve generalization via increased data diversity and ensemble distillation. It still does not directly establish a connection between generalization performance and policy robustness; instead, it studies certain symmetry groups in the training environment and expects the policy to be invariant to these transformations. In contrast, our framework **does not** require assumptions about the symmetry of the rendering function. This indicates that our theoretical framework is more general. Additionally, the experiments in [5] are **limited to toy examples** such as Reacher with rotational symmetry, whereas the Procgen benchmark provides a much more challenging generalization setting. We will discuss this work in the related work section.
> >
> > In summary, we respectfully disagree that our theoretical contributions overlap with these works. To avoid overstating our theoretical contributions, we will additionally clarify in the revised PDF that our theoretical framework is end-to-end, i.e., it does not require approaching from a feature learning perspective.

---

> ### Author Response · Authors · 2025-11-26
>
> >Disconnect between theory and practice The biggest weakness of this paper in my view is the strong disparity between the derived theoretical claims and the conjectured effect of mutual distillation. The argument or intuition for how mutual distillation learns more robust features is very handwavy. For example it is not obvious at all that policies from different intialization or different batch orders or other factors will learn significantly different features. The work above indeed does show a related effect theoretically using neural tangent kernel theory [5]. Knowing that feature learning is difficult to treat theoretically in deep learning I would suggest an alternative path would be to support the argument more empirically. For example with experiments showing indeed that different initialization / different exploration seeds / different batch orders etc. lead to feature learning of distinct spurious features which can then be eliminated through mutual distillation.
>
> Thank you for your feedback. We would like to emphasize again that our theoretical framework **does not** require decoupling the encoder from the policy, nor does it rely on feature learning. To further demonstrate how mutual distillation can learn more robust features, we also evaluated the $\mathcal{R}$-robustness of PPO and MDPO. We proposed two different metrics: one is the maximum $\mathcal{R}$-robustness metric $\mathcal{R}_ {\mathrm{max}}$ from the original paper, and the other is the average $\mathcal{R}$-robustness metric $\mathcal{R}_ {\mathrm{average}}$, defined as follows:
>
> $$\mathcal{R}_ {\mathrm{max}}=\max_ {i=1,\dots,1000;j=1,\dots,100}D_ {\mathrm{TV}}(\pi_ {\theta}(\cdot|o_i)\Vert\pi_ {\theta}(\cdot|f_j^{(i)}(o_i)))$$
>
> $$\mathcal{R}_ {\mathrm{average}}=\frac{1}{100000}\sum_ {i=1}^{1000}\sum_ {j=1}^{100}D_ {\mathrm{TV}}(\pi_ {\theta}(\cdot|o_i)\Vert\pi_ {\theta}(\cdot|f_j^{(i)}(o_i)))$$
>
> Here, $\mathcal{R}_ {\mathrm{max}}$ is a direct measure of the policy robustness from the original paper, but it is relatively loose. We can easily extend the theorem results from the paper to the form of $\mathcal{R}_ {\mathrm{average}}$, which provides a tighter bound, simply by rewriting Definition 3.6 as an expectation over a specific set of underlying states $s\in\mathcal{S}$. The results are shown in the table below:
>
> | training stage | 0% | 25% | 50% | 75% | 100% |
> |------|------|------|------|------|------|
> |   PPO ($\mathcal{R}_ {\mathrm{max}}$)  |   0.1100   |   0.9989   |   0.9959   |   1.0   |   1.0   |
> |   MDPO ($\mathcal{R}_ {\mathrm{max}}$)  |   0.0781   |   0.9936   |   0.9987   |   0.9997   |   0.9993   |
>
> | training stage | 0% | 25% | 50% | 75% | 100% |
> |------|------|------|------|------|------|
> |   PPO ($\mathcal{R}_ {\mathrm{average}}$)  |   0.0239   |   0.3058   |   0.3853   |   0.7669   |   0.8953   |
> |   MDPO ($\mathcal{R}_ {\mathrm{average}}$)  |   0.0172   |   0.5949   |   0.5858   |   0.5576   |   0.4899   |
>
> We can see that under the tighter metric, MDPO's $\mathcal{R}_ {\mathrm{average}}$ gradually **decreases** during training, while PPO's **increases**, showing that mutual distillation effectively improves policy robustness.
>
> Furthermore, we respectfully believe that generalization performance is the most direct reflection of whether a model has learned spurious features. If a policy has learned the true representations of the environment, it will naturally exhibit better generalization.
>
> To further demonstrate that mutual distillation indeed eliminates irrelevant features, we conducted the following experiment: we compared the **Centered Kernel Alignment (CKA)** of two agents in MDPO on the same batch of adversarial examples at different training stages, under different distillation loss weights ($\alpha=1.0$ and $\alpha=0.0$). The results are shown in the table below:
>
> | training stage | 0% | 25% | 50% | 75% | 100% |
> |------|------|------|------|------|------|
> |   MDPO ($\alpha=1.0$)  |   0.649   |   0.769   |   0.797   |   0.850   |   0.867   |
> |   MDPO ($\alpha=0.0$)  |   0.649   |   0.185   |   0.131   |   0.146   |   0.004   |
>
> It is evident that after mutual distillation ($\alpha=1.0$), the two agents learned more robust representations, as their representations of the same batch of adversarial examples became increasingly **similar**. In contrast, when the distillation weight $\alpha=0.0$, the representations of the two agents on this batch of adversarial examples **diverge** over time, indicating that they learned different spurious features.

---

> > ### Author Response · Authors · 2025-11-26
> >
> > >Inconsistency in robustness metric. The robustness term is defined as a maximum total variation distance max TV distance, yet Table 1 lists values larger than 1.
> >
> > Thank you for your careful review. Indeed, in discrete action spaces, the upper bound of the TV distance between any two policies is 1. We have re-tested $\mathcal{R}_ {\mathrm{max}}$ and $\mathcal{R}_ {\mathrm{average}}$ in eight environments, and the results are shown in the table below:
> >
> > | $\mathcal{R}_{\mathrm{max}}\downarrow$ | caveflyer | chaser | climber | fruitbot | heist | jumper | leaper | plunder |
> > |------------------------------|-----------|--------|---------|----------|-------|--------|--------|---------|
> > | PPO  | 1.0000 | 1.0000 | 1.0000 | 1.0000 | 0.9683 | 0.9699 | 1.0000 | 1.0000 |
> > | MDPO | **0.9877** | **0.9982** | **0.8344** | **0.6973** | **0.9142** | **0.9313** | **0.9423** | **0.9431** |
> >
> > | $\mathcal{R}_{\mathrm{average}}\downarrow$ | caveflyer | chaser | climber | fruitbot | heist | jumper | leaper | plunder |
> > |----------------------------------|-----------|--------|---------|----------|-------|--------|--------|---------|
> > | PPO  | 0.8632 | 0.8897 | 0.6221 | 0.9977 | 0.5255 | 0.7878 | 0.7579 | 0.7789 |
> > | MDPO | **0.5044** | **0.6322** | **0.2363** | **0.4524** | **0.1138** | **0.3870** | **0.4476** | **0.4165** |
> >
> > It can be seen that the MDPO policies achieved significantly lower $\mathcal{R}_ {\mathrm{max}}$ and $\mathcal{R}_ {\mathrm{average}}$, showing that **mutual distillation effectively improved policy robustness**.
> >
> > >Algorithm 1 implies two agents collecting data separately; it is unclear whether the 50M total steps refer to per agent or shared. The appendix does not explicitly resolve this. Without this clarification, comparisons to PPO may reflect more total experience rather than representational effects.
> >
> > >do the mutual distillation learners together use the same amount of samples as PPO baselines? If not, how is data shared or counted?
> >
> > Thank you for your comment. The two policies indeed double the actual number of interactions. However, when computing their respective losses, the two agents cannot directly access each other's training data for RL training. Specifically, assume that agent A (denote as $\pi_A$) collects a batch of training data $\mathcal{D}_A=((o_1^A,a_1^A,r_1^A),\dots,(o_k^A,a_k^A,r_k^A))$, agent B (denote as $\pi_B$) collects a same batch of training data $\mathcal{D}_B=((o_1^B,a_1^B,r_1^B),\dots,(o_k^B,a_k^B,r_k^B))$. Then, the total loss for agent A is:
> >
> > $$\mathcal{L}_A=\text{RL loss}(\mathcal{D}_A)+\alpha\cdot\frac{1}{K}\sum_i\text{KL}(\pi_B(\cdot|o_i^A)\Vert\pi_A(\cdot|o_i^A))$$
> >
> > which only involves agent A's **own** dataset $\mathcal{D}_A$, while the KL loss serves as a regularization term. Therefore, from each agent's own perspective, the number of interactions remains consistent with the baseline. In addition, **Figure 5** includes a comparison at **equal training cost**, showing that even when scaling the PPO baseline to match MDPO's total training cost, MDPO still achieves better generalization performance.
> >
> > >Adversarial/robustness experiments underspecified. The procedure for generating “adversarial renderings” (random CNNs) lacks some detail, and, importantly, motivation. I find this a very perculiar way of generating adversarial examples, could the authors elaborate this choice?
> >
> > >could you describe the random CNNs used to generate alternate renderings and the motivation behind it?
> >
> > Thank you for your question. We chose random CNNs to generate adversarial examples because they **do not** change the underlying semantics of the current observation, yet introduce irrelevant features (see Figure 10 in the appendix). Moreover, adversarial examples generated by random CNNs are more challenging than those created by adjusting brightness, contrast, saturation, or hue in Figure 11, as they induce significant visual changes in the current observation. Therefore, we use random CNNs to generate adversarial examples to test the learned policy's robustness to irrelevant features. The pytorch style initialization of the random CNNs is:
> >
> > ```python
> > random_conv = nn.Conv2d(3, 3, kernel_size=3, padding=1, stride=1).to(device)
> > random_conv.weight.data = torch.randn_like(random_conv.weight.data).to(device)
> > ```
> >
> > >Despite being in the title, the term "representation convergence" is never formally defined. I'm not following what this term aims to imply, what converges to what?
> >
> > >how do you define the “representation convergence” concept?
> >
> > Thank you for your question. You may also refer to our response to Reviewer 7ZFX. Representation convergence indicates that the two agents in MDPO eventually learned more fundamental, robust representations that are not influenced by irrelevant features. Figure 2 provides an intuitive illustration of this process: the two policies converge to a more robust hypothesis space through mutual regularization, which eventually leads to improved generalization performance.

---

> ### Author Response · Authors · 2025-11-26
>
> >can you provide a theoretical statement (e.g., in linearized form) showing that mutual distillation dynamics reduce the $\mathcal{R}$-robustness term?
>
> We may not be able to provide a theoretical statement under a linear setting soon. However, we are glad to offer another **simpler and more intuitive** explanation.
>
> For different rendering functions $f\in\mathcal{F}$, we can define the set of representations that **can solve the current task** as follows:
>
> $$\Phi^f:=(\phi_1^f,\dots,\phi_{k_f}^f)$$
>
> Clearly, the **optimal** representation $\phi^*$ must be the intersection of all $\Phi^f$, as this representation can absolutely solve the task under all $f\in\mathcal{F}$, i.e.,
>
> $$\phi^*=\cap_{f\in\mathcal{F}}\Phi^f$$
>
> Thus, assuming that the two policies correspond to $f_1\in\mathcal{F}$ and $f_2\in\mathcal{F}$ for their current observations, there exist many representations (in total $k_{f_1}$ and $k_{f_2}$, respectively) that can solve their **own** tasks.
>
> **(a) So what exactly does mutual distillation do?**
>
> It turns out that mutual distillation forces these two policies to output the **same** action distribution for the same observation. Therefore, mutual distillation essentially encourages them to learn a representation that can **simultaneously** solve the tasks under **both** $f_1$ and $f_2$, i.e.,
>
> $$\phi_{\mathrm{DML}}=\Phi^{f_1}\cap\Phi^{f_2}$$
>
> This process causes the representation space that the agents can learn to become **increasingly constrained** and eventually converge to the most fundamental representation:
>
> $$\phi_{\mathrm{DML}}\rightarrow\phi^*$$
>
> **(b) So how do we measure policy robustness?**
>
> Clearly, if the representation $\phi$ learned by the policy can solve the task under a more **diverse** set of $f$, then the policy is more robust. Conversely, if the representation $\phi$ can solve the task only under a **limited** set of $f$, that indicates the policy has learned some spurious features, since this representation cannot solve all tasks.
>
> As a result, from this perspective, the representation $\phi_{\mathrm{DML}}$ learned through mutual distillation will be more robust. We hope this provides you with some intuition.
>
> **We will upload the revised PDF version as soon as possible.**
>
> ---
> **References**
>
> [1] J Schulman et al. Trust region policy optimization.
>
> [2] SY Yeh et al. Sample complexity of kernel-based q-learning.
>
> [3] N Kallus et al. Confounding-robust policy evaluation in infinite-horizon reinforcement learning.
>
> [4] M Suau et al. Bad habits: Policy confounding and out-of-trajectory generalization in RL.
>
> [5] M Weltevrede et al. How Ensembles of Distilled Policies Improve Generalisation in Reinforcement Learning.

---

### Official Review · Reviewer_zv9o · 2025-10-30

**Soundness:** 2
**Presentation:** 1
**Contribution:** 3
**Rating:** 2
**Confidence:** 3

**Summary:**

The paper presents a theoretical and empirical study arguing that mutual distillation between reinforcement learning policies acts as an implicit regularization to prevent overfitting to irrelevant features, thereby improving generalization. It claims to be the first to theoretically prove that enhancing policy robustness to irrelevant features leads to better generalization. The paper introduces Mutual Distillation Policy Optimization (MDPO), which uses Deep Mutual Learning (DML) via a KL divergence loss between to (otherwise independently) trained PPO agents. Empirically, MDPO shows improved generalization performance over a PPO baseline on the hard level configuration of the Procgen benchmark and demonstrates enhanced robustness to visual disturbances like random convolutions or changes in brightness, contrast, saturation or hue.

**Recommendation:**\
I recommend to reject the paper because it is lacking in certain fundamental areas: insufficient positioning with respect to related work,  issues with clarity regarding the significance of theoretical results, empirical results, and the novelty of the ideas/approach.

**Strengths:**

- The central idea of using mutual distillation to induce robustness to spurious correlations in the training data is interesting.
- The empirical results for MDPO look promising.
- The analytical experiments of the robustness of the MDPO policy to visual disturbances and the quality of the learned representations in Sections 5.3 and 5.4 are very insightful.

**Weaknesses:**

- The positioning of this work is severely lacking, especially concerning previous literature.
	-  The paper seems to miss several related works on topics such as:
		- Representation learning in RL (for example, [1,2])
		- Policy distillation for generalization (for example, [3,4])
		- Mutual distillation (for example, [5])
		- Overfitting to training data in RL (for example, [6,7,8])
		- The above are not necessarily exhaustive.
	- The claims regarding theoretical novelty are strong, but difficult for me to verify due to insufficient discussion of existing work, and my own lack of knowledge on this specific topic.
	- The novelty of the motivation in Section 4 is difficult to judge, since it lacks comparison with related ideas in the existing literature (for example, see [7,8]).
- The theoretical contribution has some issues:
	- The core theoretical claim is formulated too strongly. Corollary 3.8 only shows an increase in the _lower bound_ on generalization performance as robustness increases. However, increasing a lower bound does not guarantee an increase in actualized performance, which is what is promised in the abstract and introduction.
	- Clarity of the theoretical framework is lacking. For example, the proofs depend on two policies $\pi$ and $\tilde{\pi}$, but there is no mention of what the significance of these two policies are or how they are related to the overall narrative of the paper.
	- A discussion on the significance of the theoretical results is missing. Whether increasing a lower bound has any bearing on the realized performance depends on whether the bound is vacuous or not. This means the bounds derived in Section 3 would benefit from analysis or discussion on how tight they are. For example, the bound's dependence on the relative performance of two arbitrary policies $\pi$ and $\tilde{\pi}$ complicates the interpretation of its significance.
- The motivation for mutual distillation seems conceptually incomplete. The logic presented in Sections 4.1 and 4.2 hinges on the policies encountering different spurious correlations due to collecting different trajectories as a result of different initializations. However, the DML mechanism's goal is to regularize the two policies to converge to the same policy. This seems to introduce a paradox: in order to benefit from the two policies collecting different training data, they need to be regularized to be the same.
- Experimental details are missing, making it difficult to judge the validity and significance of the results. For example, the paper claims significant improvements, but there is no mention of number of seeds used, what the shaded areas in the figures denote, or how significance was tested.

**Questions:**

- Does it matter how $\pi$ and $\tilde{\pi}$ are related? Can we view them through the lens of the two policies used in mutual distillation somehow?
- The policy $\pi$, $\tilde{\pi}$ and the linear approximation $L_\pi(\tilde{\pi})$ (the only positive term in the lower bound in Corollary 3.8), seem to originate from [9], which assumes $\pi$ and $\tilde{\pi}$ are close to each other. Does this mean the lower bound will only be non-vacuous if $\pi$ and $\tilde{\pi}$ are similar? How does this impact the significance of the derived bounds?
- Line 200: "During the training process, we can only empirically bound $\mathfrak{D}_{train}$" Why?
- Figure 2: "(Right) Through mutual distillation via DML, two policies regularize each other to converge toward a more robust hypothesis space,..." I agree that they are regularized towards each other, but why would that be toward a more robust hypothesis space, and not just any other part of the non-robust hypothesis space?
- Why is an algorithm that regularizes two policies towards each other, the solution for the problem identified in Figure 1 that requires the two policies to collect substantially different data?
- For the experiments, how many seeds are used and what do the shaded regions indicate? Also, how is significance of the results determined?


**Things to improve that did not impact decision:**
- There many missing words, making sentences incomplete (for example, line 58 and 65 of the introduction).
- Section 2.1: Stating that a function $f: X \to [0,1]$ does not sufficiently define it as a probability distribution (it is missing the constraint that all the probabilities sum up to 1).
- Section 2.1: The authors seem to introduce a new MDP framework with the rendering function in the background section. Either this is new, which means it should not be in the background section, or it is an existing framework from somewhere else, which means it is missing a citation.
	- The framework also seems reminiscent of a specific type of contextual MDP [10], perhaps it could be useful to frame this work within the CMDP framework.
- Table 2: It would be interesting to also include the training performance for this experiment.
- Figure 3: Does the x-axis for MDPO include the timesteps of both agents (since they collect data independently)? In other words, at timestep 50 million in the figure, the individual MDPO policies will only have trained on 25 million steps each? If not, I feel this figure is slightly misleading.

**References:**\
[1] Learning Invariant Representations for Reinforcement Learning Without Reconstruction. Zhang et al. 2021\
[2] Cross-Trajectory Representation Learning for Zero-Shot Generalization in RL. Mazoure et al. 2022\
[3] Learning Dynamics and Generalization in Reinforcement Learning. Lyle et al. 2022\
[4] How Ensembles of Distilled Policies Improve Generalisation in Reinforcement Learning. Weltevrede et al. 2025\
[5] Dual Policy Distillation. Lai et al. 2020\
[6] On the Importance of Exploration for Generalization in Reinforcement Learning. Jiang et al. 2023\
[7] Policy Confounding and Out-of-Trajectory Generalization in RL. Suau et al. 2024\
[8] Exploration Implies Data Augmentation: Reachability and Generalisation in Contextual MDPs. Weltevrede et al. 2025\
[9] Trust Region Policy Optimization. Schulman et al. 2015\
[10] A survey of zero-shot generalisation in deep reinforcement learning. Kirk et al. 2023

---

> ### Author Response · Authors · 2025-11-24
>
> Dear reviewer zv9o, thank you for your extremely detailed and insightful comments on our paper. We believe they are very helpful for improving the quality of our paper.
>
> We are especially delighted to receive your high praise:
>
> - _The central idea is interesting._
> - _The empirical results for MDPO look promising._
> - _Sections 5.3 and 5.4 are very insightful._
>
> Below, we address your concerns one by one, and we hope this will enable you to re-evaluate the value of our work.
>
> >The paper seems to miss several related works on topics such as [1-8].
>
> >The claims regarding theoretical novelty are strong, but difficult for me to verify due to insufficient discussion of existing work, and my own lack of knowledge on this specific topic.
>
> Thank you for pointing this out. We will include a discussion of these works in the Related Work section.
>
> We are delighted to hear that you consider our theoretical claims to be strong, and we fully understand your concerns. However, we would like to respectfully point out that prior work on RL generalization typically approaches the problem from a representation learning perspective [1,2], and these frameworks inevitably **decouple** the encoder from the policy.
>
> In contrast, the results in our paper are both theoretically and empirically end-to-end, and we make no assumptions about the distribution of the rendering function class $p:\mathcal{F}\mapsto [0,1]$. Notably, on the Procgen benchmark, previous work largely focused on **empirical studies** [14, 15, 16, 17, 18, 19], whereas we are the first to build a theoretical connection between generalization performance $\eta$ and policy robustness $\mathcal{R}$. We are also pleased that this theoretical insight has been recognized by the other three reviewers:
>
> - `7ZFX`: _Generalisation in RL is a major and actively researched topic. The insights provided by the paper will have far reaching impact._
> - `xv4v`: _This work presents a formal proof of a long standing assumption that robustness of policy against irrelevant features improves generalization._
> - `kbQ7`: _The formalization of rendering families and the decomposition of generalization error into robustness and train-test divergence components is intuitive and provides a neat mathematical framing._
>
> To avoid overclaiming our theoretical contributions, we have slightly revised the wording in the abstract. We further clarify that our theoretical framework is **end-to-end**, which represents a fundamental distinction from prior results.
>
> >The novelty of the motivation in Section 4 is difficult to judge, since it lacks comparison with related ideas in the existing literature (for example, see [7,8]).
>
> Thank you for your comments, but we respectfully disagree with your point. The main results in [7] are approached from a **representation learning perspective**. Specifically, Definition 5 defines the _Minimal State Representation_, which is introduced via a state representation function $\Phi:\mathcal{S}\mapsto\bar{\mathcal{S}}$. Reference [8] primarily presents the insight that introducing an exploration phase before the RL stage can effectively alter the initial state distribution without affecting the value function estimates, thus improving generalization. We believe this shares no conceptual similarity to our motivation in Section 4, i.e., **distillation as regularization**.
>
> >The core theoretical claim is formulated too strongly. Corollary 3.8 only shows an increase in the lower bound on generalization performance as robustness increases. However, increasing a lower bound does not guarantee an increase in actualized performance, which is what is promised in the abstract and introduction.
>
> We acknowledge your point that empirically, many factors—such as data bias, neural network training, and various hyperparameter settings—can cause an improvement in the lower bound not to translate into better generalization performance. However, it is undeniable that this theoretical result is indeed important, as it provides a theoretical guarantee for all existing methods based on data augmentation or invariant representation learning. Moreover, we have prefixed this statement with ''Theoretically'' to clarify the context.
>
> In fact, as an **important paper in policy gradient methods**, TRPO [9] **only theoretically guarantees** policy monotonic improvement. However, we **cannot** dismiss its theoretical contribution based on empirical observations that the policy may not always improve monotonically. We wish to respectfully point out this lack of objectivity on your part.

---

> ### Author Response · Authors · 2025-11-24
>
> >Clarity of the theoretical framework is lacking. For example, the proofs depend on two policies
>  and $\pi$ and $\tilde{\pi}$, but there is no mention of what the significance of these two policies are or how they are related to the overall narrative of the paper.
>
> Thank you for your feedback. **Theorem 3.3** (Training performance lower bound) and **Theorem 3.4** (Generalization performance lower bound) both state that $\pi$ and $\tilde{\pi}$ are **any two** policies. You can also refer to the original TRPO [9] and CPO [11] papers, which feature similar forms of lower bounds.
>
> >A discussion on the significance of the theoretical results is missing. Whether increasing a lower bound has any bearing on the realized performance depends on whether the bound is vacuous or not. This means the bounds derived in Section 3 would benefit from analysis or discussion on how tight they are. For example, the bound's dependence on the relative performance of two arbitrary policies $\pi$ and $\tilde{\pi}$ complicates the interpretation of its significance.
>
> Thank you for your feedback. However, we did include a discussion in the **Explanation** below **Corollary 3.8**. **Corollary 3.8** shows that for any two policies $\pi$ and $\tilde{\pi}$,
>
> $$\xi(\tilde{\pi})\geq \underbrace{L_ {\pi}(\tilde{\pi})-C_ {\mathrm{train}}\mathfrak{D}_ {\mathrm{train}}}_ {\text{TRPO lower bound style}}-\underbrace{C_ {\pi}\mathcal{R}_ {\pi}-C_ {\tilde{\pi}}\mathcal{R}_ {\tilde{\pi}}-C}_ {\text{policy robustness term}}$$
>
> It is clear that improving policy robustness term can lead to improved generalization performance. If the policy is no longer influenced by any irrelevant features, then
>
> $$\text{policy robustness term}=0$$
>
> since $\mathcal{R}_ {\pi}=\mathcal{R}_ {\tilde{\pi}}=0$, and our theoretical framework degenerates into a **standard MDP** because the policy is no longer influenced by the rendering function $f$. As a result, the policy produces the **same distribution** under any $f$, so $f$ no longer needs to be formalized. This theoretical result then reduces to a TRPO-style trust region form [9, 11].
>
> >The motivation for mutual distillation seems conceptually incomplete. The logic presented in Sections 4.1 and 4.2 hinges on the policies encountering different spurious correlations due to collecting different trajectories as a result of different initializations. However, the DML mechanism's goal is to regularize the two policies to converge to the same policy. This seems to introduce a paradox: in order to benefit from the two policies collecting different training data, they need to be regularized to be the same.
>
> >Why is an algorithm that regularizes two policies towards each other, the solution for the problem identified in Figure 1 that requires the two policies to collect substantially different data?
>
> Thank you for your comments, but we respectfully disagree with your point. Mutual distillation does encourage the behaviors of two policies to converge; however, even if we have two **identical** policies $\pi_1=\pi_2=\pi$, they will still collect completely different trajectories $\tau_1=(s_0,a_0,s_1,a_1,\dots,s_T)$ and $\tau_2=(s_0,a_0',s_1',a_1',\dots,s_T')$ when interacting with the environment, due to its inherent stochasticity. Moreover, it can be shown that the probability of them collecting identical trajectories decreases **exponentially** with the trajectory length $T$:
>
> $$\mathrm{Pr}(\tau_1=\tau_2)=\prod_ {t=0}^{T-1}\sum _{a_t\in\mathcal{A}}\pi(a_t|s_t)^2\leq\lambda^T,\enspace \lambda\leq 1$$
>
> Here, we assume that the environment dynamics are deterministic for simplicity. Therefore, the fact that the two policies collect different data does not contradict the convergence of their behaviors, because even for the same policy, the probability of collecting identical trajectories is very low.
>
> >Experimental details are missing, making it difficult to judge the validity and significance of the results. For example, the paper claims significant improvements, but there is no mention of number of seeds used, what the shaded areas in the figures denote, or how significance was tested.
>
> >For the experiments, how many seeds are used and what do the shaded regions indicate? Also, how is significance of the results determined?
>
> Thank you for your feedback. We indeed overlooked mentioning the number of seeds—we used 5 random seeds and visualized the mean and standard deviation of both training and generalization performance throughout the process. We respectfully disagree with your comment that ''_it is difficult to judge the validity and significance of the results_'' because our method clearly achieves superior generalization performance and policy robustness. **Please refer to Figures 3, 4, and 5, Tables 1, 2, and 3 in the main text, as well as the additional complete experiments in the appendix for detailed evidence.**

---

> ### Author Response · Authors · 2025-11-24
>
> >Does it matter how $\pi$ and $\tilde{\pi}$ are related? Can we view them through the lens of the two policies used in mutual distillation somehow?
>
> This is a **fundamental misunderstanding** of our method. $\pi$ and $\tilde{\pi}$ represent the policy of a single agent before and after each training iteration, while the two agents regularize each other and improve robustness through mutual distillation. **$\pi$ and $\tilde{\pi}$ do not represent the two agents in our MDPO algorithm**.
>
> >The policy $\pi$ and $\tilde{\pi}$ and the linear approximation $L_{\pi}(\tilde{\pi})$ (the only positive term in the lower bound in Corollary 3.8), seem to originate from [9], which assumes $\pi$ and $\tilde{\pi}$ are close to each other. Does this mean the lower bound will only be non-vacuous if $\pi$ and $\tilde{\pi}$ are similar? How does this impact the significance of the derived bounds?
>
> We respectfully disagree with your view. $\pi$ and $\tilde{\pi}$ are **two arbitrary policies**, and the $L_{\pi}(\tilde{\pi})$ term is unaffected. The reason we want $\pi$ and $\tilde{\pi}$ to be close to each other is that we aim to minimize the $\mathfrak{D}_ {\mathrm{train}}$ term, which is the well-known trust region constraint that widely used in TRPO [9], PPO [12], and SPO [13].
>
> >Line 200: "During the training process, we can only empirically bound $\mathfrak{D}_ {\mathrm{train}}$, Why?
>
> We would like to respectfully remind you that $\mathfrak{D}_ {\mathrm{train}}$ represents the trust region, which is implemented through the ratio clipping function in the PPO objective. And because $\mathfrak{D}_{\mathrm{train}}$ depends only on the **current** policy $\mu_f$, while the states in $\mathfrak{D}_1$ and $\mathfrak{D}_2$ depend on the **new** policy $\tilde{\mu}_f$, so we can only bound this term for now and subsequently attempt to find upper bounds for $\mathfrak{D}_1$ and $\mathfrak{D}_2$.
>
> >Figure 2: "(Right) Through mutual distillation via DML, two policies regularize each other to converge toward a more robust hypothesis space,..." I agree that they are regularized towards each other, but why would that be toward a more robust hypothesis space, and not just any other part of the non-robust hypothesis space?
>
> This is the core motivation of our paper. We empirically demonstrate that they indeed converge to a more robust hypothesis space, rather than other part of the non-robust hypothesis space. Please refer to **Figure 4 and Table 1 in the main text, as well as Figure 10 in the appendix**. These provide strong evidence supporting our claim.
>
> >There many missing words, making sentences incomplete (for example, line 58 and 65 of the introduction).
>
> Thank you for your suggestion. We will improve the readability of the paper in the revised version.
>
> >Section 2.1: Stating that a function $p:\mathcal{F}\mapsto[0,1]$ does not sufficiently define it as a probability distribution (it is missing the constraint that all the probabilities sum up to 1).
>
> Thank you for your thorough review. We have stated that $p$ is a probability distribution, which by default satisfies the condition of integrating to 1. We will include this explicit condition in the revised PDF.
>
> >Section 2.1: The authors seem to introduce a new MDP framework with the rendering function in the background section. Either this is new, which means it should not be in the background section, or it is an existing framework from somewhere else, which means it is missing a citation.
>
> >The framework also seems reminiscent of a specific type of contextual MDP [10], perhaps it could be useful to frame this work within the CMDP framework.
>
> Thank you for your suggestion. We will include the relevant work in the references.
>
> >Table 2: It would be interesting to also include the training performance for this experiment.
>
> We will include the training performance in Table 2 in the revised PDF.

---

> ### Author Response · Authors · 2025-11-24
>
> >Figure 3: Does the x-axis for MDPO include the timesteps of both agents (since they collect data independently)? In other words, at timestep 50 million in the figure, the individual MDPO policies will only have trained on 25 million steps each? If not, I feel this figure is slightly misleading.
>
> Thank you for your comment. The two policies indeed double the actual number of interactions. However, when computing their respective losses, the two agents cannot directly access each other's training data for RL training. Specifically, assume that agent A (denote as $\pi_A$) collects a batch of training data $\mathcal{D}_A=((o_1^A,a_1^A,r_1^A),\dots,(o_k^A,a_k^A,r_k^A))$, agent B (denote as $\pi_B$) collects a same batch of training data $\mathcal{D}_B=((o_1^B,a_1^B,r_1^B),\dots,(o_k^B,a_k^B,r_k^B))$. Then, the total loss for agent A is:
>
> $$\mathcal{L}_A=\text{RL loss}(\mathcal{D}_A)+\alpha\cdot\frac{1}{K}\sum_i\text{KL}(\pi_B(\cdot|o_i^A)\Vert\pi_A(\cdot|o_i^A))$$
>
> which only involves agent A's **own** dataset $\mathcal{D}_A$, while the KL loss serves as a regularization term. Therefore, from each agent's own perspective, the number of interactions remains consistent with the baseline. In addition, **Figure 5** includes a comparison at **equal training cost**, showing that even when scaling the PPO baseline to match MDPO's total training cost, MDPO still achieves better generalization performance.
>
> **We will upload the revised PDF version as soon as possible.**
>
> ---
> **References**
>
> [1] A Zhang et al. Learning invariant representations for reinforcement learning without reconstruction.
>
> [2] B Mazoure et al. Cross-trajectory representation learning for zero-shot generalization in rl.
>
> [3] C Lyle et al. Learning dynamics and generalization in reinforcement learning.
>
> [4] M Weltevrede et al. How Ensembles of Distilled Policies Improve Generalisation in Reinforcement Learning.
>
> [5] KH Lai et al. Dual policy distillation.
>
> [6] Y Jiang et al. On the importance of exploration for generalization in reinforcement learning.
>
> [7] M Suau et al. Bad habits: Policy confounding and out-of-trajectory generalization in RL.
>
> [8] M Weltevrede et al. Exploration Implies Data Augmentation: Reachability and Generalisation in Contextual MDPs.
>
> [9] J Schulman et al. Trust region policy optimization.
>
> [10] R Kirk et al. A survey of zero-shot generalisation in deep reinforcement learning.
>
> [11] J Achiam et al. Constrained policy optimization.
>
> [12] J Schulman et al. Proximal policy optimization algorithms.
>
> [13] Z Xie et al. Simple policy optimization.
>
> [14] D Yarats et al. Image augmentation is all you need: Regularizing deep reinforcement learning from pixels.
>
> [15] M Laskin et al. Reinforcement learning with augmented data.
>
> [16] M Laskin et al. Curl: Contrastive unsupervised representations for reinforcement learning.
>
> [17] C Wang et al. Vrl3: A data-driven framework for visual deep reinforcement learning.
>
> [18] R Raileanu et al. Automatic data augmentation for generalization in reinforcement learning.
>
> [19] K Wang et al. Improving generalization in reinforcement learning with mixture regularization.

---

> > ### Comment · Reviewer_zv9o · 2025-11-27
> >
> > We thank the authors for their elaborate response. We will address some of them here:
> > - **Regarding positioning within existing literature:**
> > 	- Mentioning the provided works in the Related Work section unfortunately does not directly address my major concern. The lack of these related works in the positioning of the paper's contribution in the introduction, makes it somewhat difficult to validate the significance and novelty of these contributions.
> > 		- An example of this concern is in line 056: _"Invariant representation learning is a promising approach ... However, it relies on transformation correspondences, which are fundamentally inaccessible in the generalization scenarios of reinforcement learning ..._
> > 			- Isn't [1] an example of an invariant representation learning approach that uses transformation correspondences that are fundamentally _accessible_ in any generalization scenario? It relies only on the bisimulation metric, which can be defined for any MDP and does not require any prior knowledge of transformation correspondences?
> > 		- Related to this same paragraph in the introduction: I'm not sure why it is a limitation that existing approaches decouple the encoder from the policy? Specifically, when it comes to robustness of the policy to irrelevant features/distractions, doesn't showing that the encoder is robust also show the same for the policy that is based on that encoder?
> > 	- A small note: I would prefer it if (a summary of) the Related Works is in the main text, not just the appendix.
> > - **Regarding conceptual novelty of the motivation in Section 4:**
> > 	- As highlighted above, I do not yet understand why the representation learning perspective in [2] is an issue, nor do I understand yet how that makes their motivation irrelevant to the story in Section 4.
> > 	- I do not yet see how the listed papers share no conceptual similarity to your motivation in Section 4.1. From my perspective, they all argue that collecting more diverse (i.e., more than just the on-policy data from a single policy) training data from the training environments should increase generalisation, which seems to be the main motivation for MDPO?
> > - **Regarding the strength of the theoretical claim:**
> > 	- My point is not about the practical versus theoretical results. Instead, I'm referring to the fact that the theoretical bound could be completely vacuous. For example, if the rewards are strictly non-zero, then the test performance will also always be non-zero. Now, if the right-hand side of the lower bound from Corollary 3.8 is a large negative number, than this would be an example of a vacuous bound. A more robust policy might increase this lower bound, from -100 to -90 for example, but this would still have no relevance to the actual test performance (which is always > 0).  As such, a lower bound on the test performance, without proper discussion/analysis of whether it is vacuous or not, does not prove (not even theoretically) _"that enhancing the policy robustness to irrelevant features leads to improved generalization performance"_ (from the abstract).
> > 	- This is also why a discussion/analysis on the tightness of the bound would be very valuable.

---

> > > ### Comment · Reviewer_zv9o · 2025-11-27
> > >
> > > - **Clarity of theoretical framework:**
> > > 	- The fact that $\pi$ and $\tilde{\pi}$ can refer to any two policies hinders the clarity of the theoretical results in my opinion. Surely the choice of the two policies and how they relate to each other will have an impact on the different terms in the bound, and therefore on the tightness/relevance of the bounds?
> > > 	- The authors mention later on that $\pi$ and $\tilde{\pi}$ represent the policy of a single agent before and after each training iteration. But if that is the case, I don't see the connection between the theory and the proposed approach of mutual distillation?
> > > 	- On an additional note, the proof seems a little bit overly complex to me.
> > > 		- Isn't it somewhat trivial that if a policy is fully robust ($\mathcal{R} = 0$) to any rendering function $f$, it will achieve exactly the same performance regardless of rendering function $f$ (so it would have 0 generalisation gap)?
> > > 		- Furthermore, could we not (almost trivially) show that robustness decreases the generalisation gap by using something like Theorem 2 from [3] and using the definition of the rendering function $f$, policy $\mu_f$ and robustness $\mathcal{R}$?
> > > 		- It could very well be that this is not true, but in that case, I believe the theorem could benefit from a justification on why its not so trivial to prove this.
> > > 	- This also highlights an additional concern on the novelty of the proof, as I believe [4] prove something very related.
> > > 		- They first prove that a policy trained with the correct symmetric subgroup, can become robust to the full group. They then prove that a robust policy has bounded generalisation gap (which decreases as the policy becomes more robust). This second step is in principle independent from _how_ the policy has become robust or _what_ it is robust to (as long as the _what_ is relevant for the testing distribution). As such, they don't explicitly define robustness and then explicitly prove generalisation performance increases for more robust policies, but this _is_ an important substep of their overall proof. This might also be true for other proofs like this in the literature.
> > > - **Motivation for mutual distillation:**
> > > 	- I agree that a stochastic policy, rolled out twice, will likely collect different trajectories. However, I don't see how this motivates mutual distillation with two policies. Instead, doesn't this just mean you should collect more than one trajectory with the current policy (which is already standard practice in PPO and other on-policy approaches)?
> > > - **Number of environment interactions:**
> > > 	- I acknowledge that from the perspective of a single policy in MDPO, they only explicitly train on the data they collected themselves. However, I do think the fair comparison is to consider the total environment interactions of MDPO as a whole, compared to the total interactions of the baseline algorithms. Therefore, I would like to see the main results (Figures 3, 6 and 7) with total environment interactions as the metric for the x-axis.
> > >
> > > Overall, I believe the paper's approach and the empirical results are very interesting, and do constitute a worthwhile contribution. However, I still don't fully get the motivation for the approach. More importantly, I still have some significant concerns (clarity, novelty) about the theoretical results, and how it doesn't seem very related to the (motivation of) the proposed approach MDPO. As such, I will keep my score for now.
> > >
> > >
> > > [1] A Zhang et al. Learning invariant representations for reinforcement learning without reconstruction. \
> > > [2] M Suau et al. Bad habits: Policy confounding and out-of-trajectory generalization in RL. \
> > > [3] Maran et al. Tight Performance Guarantees of Imitator Policies with Continuous Actions. \
> > > [4] M Weltevrede et al. How Ensembles of Distilled Policies Improve Generalisation in Reinforcement Learning.

---

> > > > ### Author Response · Authors · 2025-11-28
> > > >
> > > > Reviewer zv9o, we appreciate your additional comments. We are greatly encouraged that you have begun to acknowledge that
> > > >
> > > > - _the paper's approach and empirical results are **very interesting** and indeed constitute a **worthwhile contribution**._
> > > >
> > > > We provide our responses to each point below.
> > > >
> > > > >Mentioning the provided works in the Related Work section unfortunately does not directly address my major concern. The lack of these related works in the positioning of the paper's contribution in the introduction, makes it somewhat difficult to validate the significance and novelty of these contributions.
> > > >
> > > > >An example of this concern is in line 056: "Invariant representation learning is a promising approach ... However, it relies on transformation correspondences, which are fundamentally inaccessible in the generalization scenarios of reinforcement learning ...
> > > >
> > > > >Isn't [1] an example of an invariant representation learning approach that uses transformation correspondences that are fundamentally accessible in any generalization scenario? It relies only on the bisimulation metric, which can be defined for any MDP and does not require any prior knowledge of transformation correspondences?
> > > >
> > > > >A small note: I would prefer it if (a summary of) the Related Works is in the main text, not just the appendix.
> > > >
> > > > Thank you for your question. We would first like to clarify what we specifically mean by **transformation correspondence** in our paper. Here, we mean that for any given observation $o_1 = f_1(s)$, it is impossible for us to infer the corresponding observation under a different rendering function $f_2\neq f_1$, i.e., $o_2 = f_2(s)$.
> > > >
> > > > Therefore, we cannot infer $o_2$ from a given observation $o_1$, since the underlying distribution of rendering functions is **unknown**. This is what we mean when we say that **transformation correspondences are fundamentally inaccessible in any generalization scenario**.
> > > >
> > > > In contrast, this issue does not arise in supervised learning, since the label of a cat can correspond to **many different images of cats**, which are readily available. Therefore, it is straightforward to learn an encoder that produces invariant representations for cats. We want to emphasize that in the context of reinforcement learning generalization, such transformation correspondences are fundamentally **inaccessible**.
> > > >
> > > > Now, let us go back to [1] and [2]. We would like to clarify further. While both [1] and [2] provide frameworks for learning invariant representations without relying on transformation correspondences, we believe that they introduce an additional encoder pretraining stage that is **separate** from the reinforcement learning process, potentially hindering sample efficiency and resulting in suboptimal downstream representations, which can further limit end-to-end adaptability.
> > > >
> > > > On the other hand, modern policy gradient algorithms such as TRPO [5], PPO [6], and SPO [7] typically formulate an end-to-end policy $\pi$. This further **motivates us** to develop a framework that is both theoretically and empirically end-to-end, while allowing easy integration into the reinforcement learning pipeline—exactly what MDPO achieves.
> > > >
> > > > Personally speaking, we **do believe** that this approach is more elegant, as we **do not** need to worry about how to learn an encoder with invariant representations. Instead, the two agents enhance generalization by mutually regularizing the spurious correspondences they learn from the environment.
> > > >
> > > > We find this perspective highly compelling, and thus we have extended our discussion of **representation learning** to the related work. Following your suggestion, we have also moved the related work into the **main text** to clearly situate our approach within the context of prior studies.
> > > >
> > > > >Related to this same paragraph in the introduction: I'm not sure why it is a limitation that existing approaches decouple the encoder from the policy? Specifically, when it comes to robustness of the policy to irrelevant features/distractions, doesn't showing that the encoder is robust also show the same for the policy that is based on that encoder?
> > > >
> > > > You are correct. If we learn an encoder with invariant representations, the resulting policy will also be more robust (which can be rigorously proven under the assumption that the downstream policy probe is Lipschitz continuous). However, it is worth noting that in our method, we **do not** explicitly optimize for an invariant representation objective (as in [1] and [2]), we only do mutual distillation, yet invariant representations **spontaneously emerge** in the encoders of the two policies (see Figure 4 and 9). We believe that this finding alone is remarkable.

---

> > > > > ### Author Response · Authors · 2025-11-28
> > > > >
> > > > > >As highlighted above, I do not yet understand why the representation learning perspective in [2] is an issue, nor do I understand yet how that makes their motivation irrelevant to the story in Section 4.
> > > > >
> > > > > >I do not yet see how the listed papers share no conceptual similarity to your motivation in Section 4.1. From my perspective, they all argue that collecting more diverse (i.e., more than just the on-policy data from a single policy) training data from the training environments should increase generalisation, which seems to be the main motivation for MDPO?
> > > > >
> > > > > >I agree that a stochastic policy, rolled out twice, will likely collect different trajectories. However, I don't see how this motivates mutual distillation with two policies. Instead, doesn't this just mean you should collect more than one trajectory with the current policy (which is already standard practice in PPO and other on-policy approaches)?
> > > > >
> > > > > We appreciate your question, but we respectfully disagree with your viewpoint. The core motivation of our **Distillation as Regularization** section is not to demonstrate that data diversity alone improves generalization. In fact, if data diversity were the primary factor, the scaled PPO in **Figure 5** would be expected to exhibit generalization performance comparable to MDPO. However, the truth is, MDPO still achieves significantly better generalization across these environments, suggesting that the improvement cannot be attributed simply to data diversity.
> > > > >
> > > > > The core motivation here is that the data collected by the two policies will inevitably exhibit their **own biases**, since the collected trajectories cannot exhaust all possibilities. Consequently, if the two policies are trained completely independently, they are likely to **exploit the biases** in their own data. Mutual distillation, on the other hand, forces them to learn representations that can **simultaneously** account for each other's biased datasets. To illustrate this point, we would like to provide a simpler and more intuitive explanation:
> > > > >
> > > > > For different rendering functions $f\in\mathcal{F}$, we can define the set of representations that **can solve the current task** as follows:
> > > > >
> > > > > $$\Phi^f:=(\phi_1^f,\dots,\phi_{k_f}^f)$$
> > > > >
> > > > > Clearly, the **optimal** representation $\phi^*$ must be the intersection of all $\Phi^f$, as this representation can absolutely solve the task under all $f\in\mathcal{F}$, i.e.,
> > > > >
> > > > > $$\phi^*=\cap_{f\in\mathcal{F}}\Phi^f$$
> > > > >
> > > > > Thus, assuming that the two policies correspond to $f_1\in\mathcal{F}$ and $f_2\in\mathcal{F}$ for their current observations, there exist many representations (in total $k_{f_1}$ and $k_{f_2}$, respectively) that can solve their **own** tasks.
> > > > >
> > > > > **(a) So what exactly does mutual distillation do?**
> > > > >
> > > > > It turns out that mutual distillation forces these two policies to output the **same** action distribution for the same observation. Therefore, mutual distillation essentially encourages them to learn a representation that can **simultaneously** solve the tasks under **both** $f_1$ and $f_2$, i.e.,
> > > > >
> > > > > $$\phi_{\mathrm{DML}}=\Phi^{f_1}\cap\Phi^{f_2}$$
> > > > >
> > > > > This process causes the representation space that the agents can learn to become **increasingly constrained** and eventually converge to the most fundamental representation:
> > > > >
> > > > > $$\phi_{\mathrm{DML}}\rightarrow\phi^*$$
> > > > >
> > > > > **(b) So how do we measure policy robustness?**
> > > > >
> > > > > Clearly, if the representation $\phi$ learned by the policy can solve the task under a more **diverse** set of $f$, then the policy is more robust. Conversely, if the representation $\phi$ can solve the task only under a **limited** set of $f$, that indicates the policy has learned some spurious features, since this representation cannot solve all tasks.
> > > > >
> > > > > As a result, from this perspective, the representation $\phi_{\mathrm{DML}}$ learned through mutual distillation will be more robust. We hope this provides you with some intuition.

---

> ### Author Response · Authors · 2025-11-28
>
> >My point is not about the practical versus theoretical results. Instead, I'm referring to the fact that the theoretical bound could be completely vacuous. For example, if the rewards are strictly non-zero, then the test performance will also always be non-zero. Now, if the right-hand side of the lower bound from Corollary 3.8 is a large negative number, than this would be an example of a vacuous bound. A more robust policy might increase this lower bound, from -100 to -90 for example, but this would still have no relevance to the actual test performance (which is always > 0). As such, a lower bound on the test performance, without proper discussion/analysis of whether it is vacuous or not, does not prove (not even theoretically) "that enhancing the policy robustness to irrelevant features leads to improved generalization performance" (from the abstract).
>
> >This is also why a discussion/analysis on the tightness of the bound would be very valuable.
>
> We appreciate your suggestion. We would like to clarify that in Definition 4.6, policy robustness is defined as
>
> $$\sup\_{s\in\mathcal{S},\tilde{f},f\in\mathcal{F}}D\_{\mathrm{TV}}(\mu\_f\Vert\mu\_{\tilde{f}})[s]=\mathcal{R}$$
>
> This definition itself is rather loose, as it involves the **maximum** over the **entire state space**. A simple modification would be to replace the maximum with the **expectation** over a given state distribution, which would immediately yield a tighter bound in our paper. As we emphasized after Definition 4.6:
>
> - _our intention in this definition is not to derive the tightest possible bound, but rather to demonstrate how policy robustness to irrelevant features can contribute to improved generalization._
>
> Therefore, while the range you suggested of –100 to –90 is possible, a range of –5 to 5 is also plausible. **Corollary 4.8** explicitly shows that the lower bound of generalization performance can be improved through policy robustness. Consequently, we believe this at least indicates that if the policy is sufficiently robust (for example, in the extreme case where $\mathcal{R}=0$), generalization performance will improve, which already constitutes a meaningful theoretical result.
>
> We would like to emphasize again that our goal is **not** to obtain the tightest possible bound, but rather to establish a connection between generalization performance and policy robustness, and our bound **can be easily improved**.

---

> ### Author Response · Authors · 2025-11-28
>
> >The fact that $\pi$ and $\tilde{\pi}$ can refer to any two policies hinders the clarity of the theoretical results in my opinion. Surely the choice of the two policies and how they relate to each other will have an impact on the different terms in the bound, and therefore on the tightness/relevance of the bounds?
>
> >The authors mention later on that $\pi$ and $\tilde{\pi}$ represent the policy of a single agent before and after each training iteration. But if that is the case, I don't see the connection between the theory and the proposed approach of mutual distillation?
>
> Thank you for your question. We first present our lower bound:
>
> $$\xi(\tilde{\pi})\geq L\_{\pi}(\tilde{\pi})-C\_{\mathrm{train}}\mathfrak{D}\_{\mathrm{train}}-C\_{\pi}\mathcal{R}\_{\pi}-C\_{\tilde{\pi}}\mathcal{R}\_{\tilde{\pi}}-C$$
>
> This bound can be further expressed as
>
> $$\xi(\tilde{\pi})\geq \eta(\pi)+\mathbb{E}\_{f\sim p\_{\mathrm{train}}(\cdot),s\sim d^{\mu\_f}(\cdot),a\sim\tilde{\mu}\_f(\cdot|s)}[A^{\mu\_f}(s,a)]-C\_{\mathrm{train}}\mathbb{E}\_{f\sim p\_{\mathrm{train}}(\cdot),s\sim d^{\mu\_f}(\cdot)}[D\_{\mathrm{TV}}(\tilde{\mu}\_f\Vert\mu\_f)[s]]-C\_{\pi}\mathcal{R}\_{\pi}-C\_{\tilde{\pi}}\mathcal{R}\_{\tilde{\pi}}-C$$
>
> Hence, using importance sampling, we have
>
> $$\xi(\tilde{\pi})\geq\eta(\pi)+\underbrace{\mathbb{E}\_{f\sim p\_{\mathrm{train}}(\cdot),s\sim d^{\mu\_f}(\cdot),a\sim\mu\_f(\cdot|s)}[\frac{\tilde{\mu}\_f(a|s)}{\mu\_f(a|s)}A^{\mu\_f}(s,a)]}\_{\text{surrogate objective}}-C\_{\mathrm{train}}\underbrace{\mathbb{E}\_{f\sim p\_{\mathrm{train}}(\cdot),s\sim d^{\mu\_f}(\cdot)}[D\_{\mathrm{TV}}(\tilde{\mu}\_f\Vert\mu\_f)[s]]}\_{\text{trust region}}-C\_{\pi}\mathcal{R}\_{\pi}-C\_{\tilde{\pi}}\underbrace{\mathcal{R}\_{\tilde{\pi}}}\_{\text{policy robustness}}-C$$
>
> Therefore, we aim to optimize the **surrogate objective** while constraining the **trust region**, which is achieved through the PPO **ratio clipping objectives** of the two policies in MDPO. The role of mutual distillation is to prevent the two policies from overfitting to the biases in their own training data. As a result, mutual distillation enhances the robustness of the new policy $\tilde{\mu}_f$, i.e., it **reduces** $\mathcal{R}\_{\tilde{\pi}}$ (which has **already been empirically demonstrated** both in the original paper and in our responses to other reviewers).
>
> All the results in our paper consistently support our claims from multiple perspectives, including improved generalization performance (**Figure 3**), more robust representations for adversarial examples (**Figure 4**), and lower empirical $\mathcal{R}$ values (**Table 1**).
>
> Furthermore, the ablation experiments on training cost in Figure 5 provide additional evidence that mutual distillation is not merely a simple effect of additional samples. We believe that our theoretical and empirical results **align well** from multiple perspectives.
>
> >Isn't it somewhat trivial that if a policy is fully robust $\mathcal{R}=0$ to any rendering function $f$, it will achieve exactly the same performance regardless of rendering function $f$ (so it would have 0 generalisation gap)?
>
> You are correct, but this trivial case does not imply that our theoretical results are trivial. The insights provided by our theory go far beyond this, as we explicitly establish a lower bound on generalization performance, which explicitly incorporates policy robustness, as we have consistently emphasized above.
>
> >Furthermore, could we not (almost trivially) show that robustness decreases the generalisation gap by using something like Theorem 2 from [3] and using the definition of the rendering function $f$, policy $\mu_f$ and robustness $\mathcal{R}$?
>
> We respectfully disagree with your viewpoint. The $J$ in Theorem 2 from [3] merely represents the expected return, which is the **simplest result** in TRPO [5] or CPO [8], and this expected return **does not** distinguish between training and generalization performance. We emphasize again that our theoretical results are **not trivial**; we leveraged many lemmas and proof techniques from policy optimization [5, 6, 7, 8] to obtain our final results. If possible, we welcome you to review the proofs in our appendix line by line.
>
> >It could very well be that this is not true, but in that case, I believe the theorem could benefit from a justification on why its not so trivial to prove this.
>
> We respectfully disagree with your viewpoint. In fact, if the reviewer believes that proving our results is trivial, you are welcome to provide the corresponding proof. We would like to emphasize again that Theorem 2 from [3] can only be regarded as the **most trivial result** in the field of policy optimization [5, 6, 7, 8], whereas Theorem 1 from [8] establishes a much more general result.

---

> > ### Author Response · Authors · 2025-11-28
> >
> > >This also highlights an additional concern on the novelty of the proof, as I believe [4] prove something very related.
> >
> > >They first prove that a policy trained with the correct symmetric subgroup, can become robust to the full group. They then prove that a robust policy has bounded generalisation gap (which decreases as the policy becomes more robust). This second step is in principle independent from how the policy has become robust or what it is robust to (as long as the what is relevant for the testing distribution). As such, they don't explicitly define robustness and then explicitly prove generalisation performance increases for more robust policies, but this is an important substep of their overall proof. This might also be true for other proofs like this in the literature.
> >
> > We believe there is **no** disagreement here, as you also acknowledge that they **don't explicitly define robustness and then explicitly prove generalization performance increases for more robust policies.** In contrast, we indeed establish **a clear connection** between generalization performance and policy robustness.
> >
> > >I acknowledge that from the perspective of a single policy in MDPO, they only explicitly train on the data they collected themselves. However, I do think the fair comparison is to consider the total environment interactions of MDPO as a whole, compared to the total interactions of the baseline algorithms. Therefore, I would like to see the main results (Figures 3, 6 and 7) with total environment interactions as the metric for the x-axis.
> >
> > Thank you for your suggestion. Other reviewers have also inquired about the overall training cost, so we have added a limitations section in the appendix. **Figure 5** clearly shows that even when **aligning the total training cost of the PPO baseline** with our method, MDPO still achieves better generalization performance. We will try to optimize the figures in the paper. In fact, if you align the final performance of PPO trained for 50M steps with MDPO trained for 25M steps, you will find that MDPO still achieves significant improvements in most environments.
> >
> > If you have any further questions, please do not hesitate to let us know. Otherwise, we sincerely ask you to reconsider the value of our work, as we firmly believe that **a score of 2** cannot objectively reflect the quality of our work or the potential impact it may have on the community. In addition, we would like to gently remind you that, as per your request, the **related work** has been moved to the **main text**.
> >
> > ---
> > **References**
> >
> > [1] A Zhang et al. Learning invariant representations for reinforcement learning without reconstruction.
> >
> > [2] M Suau et al. Bad habits: Policy confounding and out-of-trajectory generalization in RL.
> >
> > [3] Maran et al. Tight Performance Guarantees of Imitator Policies with Continuous Actions.
> >
> > [4] M Weltevrede et al. How Ensembles of Distilled Policies Improve Generalisation in Reinforcement Learning.
> >
> > [5] J Schulman et al. Trust region policy optimization.
> >
> > [6] J Schulman et al. Proximal policy optimization algorithms.
> >
> > [7] Z Xie et al. Simple policy optimization.
> >
> > [8] J Achiam et al. Constrained policy optimization.

---

> > > ### Author Response · Authors · 2025-11-28
> > >
> > > Finally, we would like to emphasize that we have uploaded our **source code** in the **Supplementary Material**, and we sincerely encourage you to try it.
> > >
> > > We believe that if you compare the generalization performance curves of PPO and MDPO in the `bigfish` environment, you will be impressed by the significant improvement in generalization achieved by MDPO.

---

### Official Review · Reviewer_xv4v · 2025-11-01

**Soundness:** 2
**Presentation:** 3
**Contribution:** 3
**Rating:** 6
**Confidence:** 4

**Summary:**

This paper attempts to prove the conjecture that the policy which is robust to irrelevant features would lead to improved generalization performance. In addition, they propose mutual distillation of policies to achieve such robustness and presents the intuition behind that. While they do not show state-of-the-arts performance, they present proof-of-concept for their approach with basic regularization baseline on all environments of Procgen benchmark.

**Strengths:**

### Strengths:

1. This work presents a formal proof of a long standing assumption that robustness of policy against irrelevant features improves generalization. In particular, they derive a lower bound for generalization performance that includes minimization of a robustness term, which is defined how a policy is influenced by two different rendering (perturbation) functions.

2. The paper is presented in a clear and well-organized way. Especially, Fig. 1 and 2 helps the readers to better understand the intuition and impact of DML in the discussed setting.

3. Legitimate ablations are conducted and the results validate the claims.

**Weaknesses:**

### Weaknesses:

1. The proposed method has been validated only on the ProcGen benchmark. Experiments on more diverse set up is needed to show the applicability of such methods.

2. While I understand that the target is not to outperform the state-of-the arts, but how DML stands against other data augmentation based approaches such as [1] are not evident. While the authors present result with SPO, it seems SPO performance itself is not upto the current standard.

3. The proposed method relies on multiple policies for distillation. However, the computational overhead compared to single policy methods is not discussed.

[1] Raileanu, Roberta, et al. "Automatic data augmentation for generalization in reinforcement learning." Advances in Neural Information Processing Systems 34 (2021): 5402-5415.

**Questions:**

I am wondering how MDPO will scale to large number of policies beyond only two policies as discussed. Can you share some insights on that?

---

> ### Author Response · Authors · 2025-11-27
>
> Dear Reviewer xv4v, thank you for your positive feedback of our paper. We are especially encouraged by your recognition that our work
>
> - _presents a formal proof of a long-standing assumption._
>
> - _Fig. 1 and Fig. 2 help readers better understand the intuition._
>
> Below, we provide our responses to your questions.
>
> >The proposed method has been validated only on the ProcGen benchmark. Experiments on more diverse set up is needed to show the applicability of such methods.
>
> Thank you for your suggestion. To further highlight the generalization ability of MDPO, we also experimented with incorporating DML into a robotic training scenario, and the final navigation success rates are shown in the table below:
>
> | RL training    | success rate % |
> |----------------|----------------|
> | LSTM w/o. DML  | 61.8           |
> | LSTM w. DML    | **63.5**           |
> | SRU w/o. DML   | 65.7           |
> | SRU w. DML     | **78.9**           |
>
> Here, SRU (Spatially-Enhanced Recurrent Units) is a novel type of RNN architecture [1]. This demonstrates that MDPO can be extended to more diverse and more challenging scenarios. In addition, we would like to clarify that Procgen [2] is already a highly representative benchmark for evaluating generalization in RL. For now, the original Procgen paper [2] has been cited **805** times. We also note that several related works have used Procgen as their sole benchmark [3, 4, 5, 6, 7].
>
> We believe that our insights (i.e., **distillation as regularization**) will inspire future research exploring mutual distillation and generalization. However, extending MDPO to more diverse tasks may be beyond the current scope of our work.
>
> >While I understand that the target is not to outperform the state-of-the arts, but how DML stands against other data augmentation based approaches such as [8] are not evident. While the authors present result with SPO, it seems SPO performance itself is not upto the current standard.
>
> Thank you for your valuable suggestion. We have added DrAC in [8] as an additional baseline, and the results are shown in the table below:
>
> | Algorithm | bigfish     | dodgeball   |
> |-------------|-------------|-------------|
> | PPO                | 0.26 ± 0.23 | 0.92 ± 0.46 |
> | DrAC         | 0.33 ± 0.19 | 1.04 ± 0.43 |
> | MDPO         | **16.11 ± 4.63**| **5.66 ± 1.98** |
>
> We found that DrAC in [8] primarily considers the `easy` setting of Procgen, whereas the generalization scenarios in our paper correspond to the `hard` setting. We observed that DrAC does not lead to a significant generalization improvement. This further suggests that mutual distillation may be a highly general technique for improving generalization, as it **does not** require manually incorporating data augmentation strategies and is **easy to implement**.
>
> >The proposed method relies on multiple policies for distillation. However, the computational overhead compared to single policy methods is not discussed.
>
> Thank you for your valuable feedback. MDPO indeed incurs additional computational overhead. Nevertheless, as shown in Figure 5, we found that even when scaling the PPO baseline to match the total computational cost of MDPO, MDPO still achieves significantly better generalization performance, showing that **mutual distillation is far more than simply increasing total experience**. We will add a discussion Section on computational overhead in the revised PDF version. Thank you again for your comments.

---

> > ### Author Response · Authors · 2025-11-27
> >
> > >I am wondering how MDPO will scale to large number of policies beyond only two policies as discussed. Can you share some insights on that?
> >
> > We appreciate this point, as it may involve large-scale parallelism and scaling in RL simulation environments. In fact, MDPO can be naturally extended to $N(N\geq 2)$ agents. Specifically, suppose we have $\pi_1,\pi_2,\dots,\pi_N$, for each policy $\pi_i$, suppose its RL loss is $\mathcal{L}_{\mathrm{RL}}^i$. Then, its KL loss term can be naturally extended as follows:
> >
> > $$\mathcal{L}\_{\mathrm{KL}}^i=\frac{1}{N-1}\sum\_{j\neq i}D\_{\mathrm{KL}}(\pi_j\Vert\pi_i)$$
> >
> > Therefore, the DML loss for each policy is
> >
> > $$\mathcal{L}\_{\mathrm{DML}}^i=\mathcal{L}\_{\mathrm{RL}}^i+\alpha\cdot\mathcal{L}\_{\mathrm{KL}}^i$$
> >
> > and the total loss is given by
> >
> > $$\mathcal{L}\_{\mathrm{MDPO}}=\frac{1}{N}\sum\_{i=1}^N\mathcal{L}\_{\mathrm{DML}}^i$$
> >
> > Accordingly, we have added experiments with $N=3$, as shown in the table below:
> >
> > | $N$  | dodgeball | starpilot |
> > |----------------|----------------|----------------|
> > | $N=1$ (baseline)  | 0.92 ± 0.46 | 3.99 ± 1.21 |
> > | $N=2$    | 5.66 ± 1.98 | 11.28 ± 3.04 |
> > | $N=3$   | **5.93 ± 1.95** | **11.43 ± 2.88** |
> >
> > However, we believe that increasing $N$ will quickly lead to diminishing marginal effects and higher computational overhead, so $N=2$ seems sufficient. We would be glad to continue exploring this direction in future work.
> >
> > **We will upload the revised PDF version as soon as possible.**
> >
> > ---
> > **References**
> >
> > [1] F Yang et al. Improving Long-Range Navigation with Spatially-Enhanced Recurrent Memory via End-to-End Reinforcement Learning.
> >
> > [2] K Cobbe et al. Leveraging procedural generation to benchmark reinforcement learning.
> >
> > [3] KW Cobbe et al. Phasic policy gradient.
> >
> > [4] K Wang et al. Improving generalization in reinforcement learning with mixture regularization.
> >
> > [5] K Wang et al. PPG reloaded: An empirical study on what matters in phasic policy gradient.
> >
> > [6] S Moon et al. Rethinking value function learning for generalization in reinforcement learning.
> >
> > [7] R Raileanu et al. Decoupling value and policy for generalization in reinforcement learning.
> >
> > [8] R Raileanu et al. Automatic data augmentation for generalization in reinforcement learning.

---

### Official Review · Reviewer_7ZFX · 2025-11-02

**Soundness:** 2
**Presentation:** 2
**Contribution:** 3
**Rating:** 6
**Confidence:** 3

**Summary:**

This paper presents a theoretical framework to demonstrate that improving the policy robustness to irrelevant features enhances its generalisation performance. The paper further shows that deep mutual learning forms an implicit regularisation and prevents policy from overfitting to irrelevant features. Empirical results are given on the ProcGen benchmark, designed to test generalisability under controlled environments, as well as for toy examples. The proposed method, mutual distillation policy optimisation, demonstrates benefits as compared to the selected baseline approaches.

**Strengths:**

The paper presents a new theoretical framework to investigate generalisation issues in deep RL. Generalisation in RL is a major and actively researched topic. The insights provided by the paper will have far reaching impact.

**Weaknesses:**

Although impactful, the experimental evaluation is limited. In the sense that, it doesn't demonstrate the phenomenon exists, beyond testing on the ProcGen benchmark and presenting performance. Also, apart from the toy example.

There are other methods focusing on distillation (mutual or peer). However, these papers seem not to be mentioned in the paper. It would be good to see a comparison, for example,

* Periodic Intra-Ensemble Knowledge Distillation for Reinforcement Learning, https://arxiv.org/pdf/2002.00149

* Robust Domain Randomised Reinforcement Learning through Peer-to-Peer Distillation, https://arxiv.org/pdf/2012.04839 (this has been cited in the paper)

* Online Policy Distillation with Decision-Attention, https://arxiv.org/pdf/2406.05488

While the theoretical framework seems to be strong, the paper lacks interpretability. More experiments would be helpful to understand the impact of mutual distillation on the representation space.

Running multi distillation may increase the computational overhead, which has not beed discussed in the paper.

Mutual distillation offers regularisation by reducing the reliance on irrelevant features. However, the generalisability of the approach hasn't been investigated.

**Questions:**

1) Have the authors analysed or discussed the additional cost introduced by mutual distillation?

2) Has the method been tested across tasks with different sources of irrelevant variation, or is it specific to the evaluated benchmarks?

3) Can the representational changes be visualised or quantified to support the theoretical claims?

4) Could the authors comment on how their approach differs from or relates to the works mentioned above?

---

> ### Author Response · Authors · 2025-11-19
>
> Dear Reviewer 7ZFX, thank you for your valuable suggestions and questions. We will address them one by one.
>
> >Although impactful, the experimental evaluation is limited. In the sense that, it doesn't demonstrate the phenomenon exists, beyond testing on the ProcGen benchmark and presenting performance. Also, apart from the toy example.
>
> >Can the representational changes be visualised or quantified to support the theoretical claims?
>
> >While the theoretical framework seems to be strong, the paper lacks interpretability. More experiments would be helpful to understand the impact of mutual distillation on the representation space.
>
> Thank you for your comments. First, to demonstrate the existence of the representation convergence phenomenon, we computed the **Centered Kernel Alignment (CKA)** between the encoders of the two policies of MDPO on the bigfish environment at different training stages, using the same batch of 100 adversarial samples as input:
>
> | training stage | 0% | 25% | 50% | 75% | 100% |
> |------|------|------|------|------|------|
> |   CKA   |   0.649   |   0.769   |   0.797   |   0.850   |   0.867   |
>
> We can observe that the two policies produce increasingly similar representations for the same batch of adversarial samples, serving as direct evidence of the representation convergence phenomenon.
>
> To further support our theoretical claims, we also evaluated the $\mathcal{R}$-robustness of PPO and MDPO. We proposed two different metrics: one is the maximum $\mathcal{R}$-robustness metric $\mathcal{R}_ {\mathrm{max}}$ from the original paper, and the other is the average $\mathcal{R}$-robustness metric $\mathcal{R}_ {\mathrm{average}}$, defined as follows:
>
> $$\mathcal{R}_ {\mathrm{max}}=\max_ {i=1,\dots,1000;j=1,\dots,100}D_ {\mathrm{TV}}(\pi_ {\theta}(\cdot|o_i)\Vert\pi_ {\theta}(\cdot|f_j^{(i)}(o_i)))$$
>
> $$\mathcal{R}_ {\mathrm{average}}=\frac{1}{100000}\sum_ {i=1}^{1000}\sum_ {j=1}^{100}D_ {\mathrm{TV}}(\pi_ {\theta}(\cdot|o_i)\Vert\pi_ {\theta}(\cdot|f_j^{(i)}(o_i)))$$
>
> Here, $\mathcal{R}_ {\mathrm{max}}$ is a direct measure of the policy robustness from the original paper, but it is relatively loose. We can easily extend the theorem results from the paper to the form of $\mathcal{R}_ {\mathrm{average}}$, which provides a tighter bound, simply by rewriting Definition 3.6 as an expectation over a specific set of underlying states $s\in\mathcal{S}$. The results are shown in the table below:
>
> | training stage | 0% | 25% | 50% | 75% | 100% |
> |------|------|------|------|------|------|
> |   PPO ($\mathcal{R}_ {\mathrm{max}}$)  |   0.1100   |   0.9989   |   0.9959   |   1.0   |   1.0   |
> |   MDPO ($\mathcal{R}_ {\mathrm{max}}$)  |   0.0781   |   0.9936   |   0.9987   |   0.9997   |   0.9993   |
>
> | training stage | 0% | 25% | 50% | 75% | 100% |
> |------|------|------|------|------|------|
> |   PPO ($\mathcal{R}_ {\mathrm{average}}$)  |   0.0239   |   0.3058   |   0.3853   |   0.7669   |   0.8953   |
> |   MDPO ($\mathcal{R}_ {\mathrm{average}}$)  |   0.0172   |   0.5949   |   0.5858   |   0.5576   |   0.4899   |
>
> We can see that under the tighter metric, MDPO's $\mathcal{R}_ {\mathrm{average}}$ gradually **decreases** during training, while PPO's **increases**, showing that mutual distillation effectively improves policy robustness.
>
> From the representation space perspective, we further evaluated the cosine similarity of the representations of the adversarial samples encoded by PPO and MDPO across different environments, as shown in the table below:
>
> |    environment   |   coinrun    |   dodgeball    |   fruitbot    |    starpilot   |
> |-------|-------|-------|-------|-------|
> |    PPO encoder   |    0.301   |    -0.006   |    0.18   |    0.027   |
> |   MDPO encoder   |   **0.781**    |    **0.585**   |    **0.547**   |    **0.718**   |
>
> It can be observed that MDPO consistently produces more similar representations among adversarial examples.
>
> >There are other methods focusing on distillation (mutual or peer). However, these papers seem not to be mentioned in the paper. It would be good to see a comparison.
>
> >Could the authors comment on how their approach differs from or relates to the works mentioned above?
>
> Thank you for the three references you cited. We will cite these works in the revised PDF version these days. The difference between our work and the above studies is as follows: [1] primarily discusses that mutual distillation benefits sample efficiency, whereas we focus on improving policy generalization. [2] uses a method similar to ours but concentrates on empirical studies. We further reveal, from both theoretical and empirical perspectives, how mutual distillation enhances generalization by improving the policy robustness to irrelevant features. [3] aims to avoid homogenization caused by mutual distillation, but we show that this is because the policies learn more fundamental representations.

---

> ### Author Response · Authors · 2025-11-19
>
> >Running multi distillation may increase the computational overhead, which has not beed discussed in the paper.
>
> >Have the authors analysed or discussed the additional cost introduced by mutual distillation?
>
> Mutual distillation does indeed introduce additional computational overhead, as both the total interaction steps and the trainable parameters are doubled. In Figure 5, we evaluated the generalization performance of the PPO baseline using same training cost, and MDPO still demonstrated better generalization. We will add a limitations section in Appendix in the revised PDF to discuss the overall computational overhead of MDPO these days.
>
> >Mutual distillation offers regularisation by reducing the reliance on irrelevant features. However, the generalisability of the approach hasn't been investigated.
>
> >Has the method been tested across tasks with different sources of irrelevant variation, or is it specific to the evaluated benchmarks?
>
> Thank you for your valuable suggestion. Although our evaluation is primarily based on the Procgen benchmark, we note that some recent works such as [4] have also achieved improved generalization in POMDP settings through mutual distillation, suggesting that mutual distillation can enhance the generalization of reinforcement learning policies across a variety of scenarios. Moreover, in the generalization tests of the Procgen benchmark [5], the training environments accessible to the agent are limited (set to the first 500 levels in this paper), while during evaluation, environments are sampled from a nearly infinite environments with unseen sources of irrelevant variation. This makes the generalization setup highly challenging and allows the method's inherent generalization ability to be properly reflected.
>
> **We will upload the revised PDF version as soon as possible.**
>
> ---
> **References**
>
> [1] ZW Hong et al. Periodic intra-ensemble knowledge distillation for reinforcement learning.
>
> [2] C Zhao et al. Robust domain randomised reinforcement learning through peer-to-peer distillation.
>
> [3] X Yu et al. Online Policy Distillation with Decision-Attention.
>
> [4] F Yang et al. Improving Long-Range Navigation with Spatially-Enhanced Recurrent Memory via End-to-End Reinforcement Learning.
>
> [5] K Cobbe et al. Leveraging procedural generation to benchmark reinforcement learning.

---

### Author Response · Authors · 2025-11-27
**We have uploaded the revised PDF**

To all reviewers,

we have uploaded the revised PDF, with the changes highlighted in **purple**. The main changes are as follows:

| # | Focus Area |Reviewer(s) | Sections | Our Actions |
|-------------------|-------------------|-------------------|-------------------|-------------------|
| 1 | Theoretical overstatement | `zv9o`, `kbQ7` | Abstract | We emphasize that our theoretical results are **end-to-end**, which fundamentally differs from previous representation learning approaches that require separating the encoder. |
| 2 | Missing words | `zv9o` | Introduction | We have added the (in)definite article in the corresponding part of the introduction to make the sentence complete. |
| 3 | Align language | `zv9o`, `kbQ7` | Preliminaries | We cited the original POMDP paper in the section where we introduce the rendering function. |
| 4 | Missing constraints | `zv9o` | Preliminaries | We added the constraint of integrating to 1 in the footnote on page 2. |
| 5 | Lack of definition | `kbQ7` | Theoretical Results | We provided the explicit form of $L_\pi$ before **Theorem 3.3**. |
| 6 | Missing details | `zv9o` | Experiments | We added in the caption of Figure 3 that the algorithm's mean and standard deviation were visualized using **5 random seeds**. |
| 7 | Inconsistency in robustness metric | `kbQ7` | Experiments | The TV divergence upper bound between discrete policies is 1, and we have re-tested the robustness metrics in **Table 1**. |
| 8 | Lack of literature review | `zv9o`, `kbQ7` | Related Work | We have expanded the related work section in the appendix and cited the relevant works mentioned by the reviewers. |
| 9 | Computational overhead | `7ZFX`, `xv4v`, `zv9o`, `kbQ7` | Limitations | We have added a Limitations section in the appendix, discussing the additional computational overhead introduced by MDPO. |
| 10 | Disconnect between theory and practice | `7ZFX`, `kbQ7` | The Representation Convergence Phenomenon | We have added a The Representation Convergence Phenomenon section in the appendix and included the experimental results **directly** demonstrating representation convergence phenomenon. |

We have done our best to respond to **every question** from all the reviewers and are currently exhausted. We sincerely hope that the reviewers can reassess the value of our work based on our additional experiments and the revised PDF. We note that all reviewers have recognized the positive impact of our work on the community:

- `7ZFX`: _Generalisation in RL is a major and actively researched topic. The insights provided by the paper will have **far reaching impact**._
- `xv4v`: _This work presents a formal proof of **a long standing assumption** that robustness of policy against irrelevant features improves generalization._
- `xv4v`: _The paper is presented in a **clear and well-organized** way. Especially, Fig. 1 and 2 helps the readers to better understand the intuition and impact of DML in the discussed setting._
- `xv4v`: _Legitimate ablations are conducted and the **results validate the claims**._
- `zv9o`: _The central idea of using **mutual distillation to induce robustness** to spurious correlations in the training data is **interesting**._
- `zv9o`: _The empirical results for **MDPO look promising**._
- `zv9o`: _The analytical experiments of the robustness of the MDPO policy to visual disturbances and the quality of the learned representations in Sections 5.3 and 5.4 are **very insightful**._
- `kbQ7`: _The paper is **easy to follow** for the most part and has a **clear motivated** given a **widespread interest in generalization** and representation learning in RL._
- `kbQ7`: _exploring generalization in RL through distillation is **well-motivated and interesting**. As far as I know, this technique **is not exhaustively researched / understood** in the context of RL._
- `kbQ7`: _The formalization of rendering families and the decomposition of generalization error into robustness and train-test divergence components is **intuitive** and provides a **neat mathematical framing**._
- `kbQ7`: _MDPO performs **consistently better** than PPO and other baselines on Procgen tasks, even when **controlling for model size or training budget**._

Given that generalization in reinforcement learning is indeed an **important and highly discussed** topic, we believe our insights can bring value to the community and both drive and inspire further research.

If you have any further questions, please feel free to reach out to us!

Best,

authors

---

### Author Response · Authors · 2025-11-28
**To the new AC (Part I)**

Dear AC, since the reviewers will no longer participate in the discussion, we summarize the reviewers' status below.

| Reviewer | Rating | Objectivity and Correctness |
|----------|----------|----------|
| `7ZFX`     | 6     | good      |
| `xv4v`      | 6      | good      |
| `kbQ7`      | 4      | fair      |

Since Reviewer `zv9o`'s comments contain **many factual errors**, we list them separately below.

| Reviewer | Rating | Objectivity and Correctness |
|----------|----------|----------|
| `zv9o`      | 2      | poor     |

Below, we would like to point out the factual errors made by the reviewer:

| Reviewer | Comment | Our Explanation |
|----------|----------|----------|
| `zv9o`      | The novelty of the motivation in Section 4 is difficult to judge, since it lacks comparison with related ideas in the existing literature (for example, see [1,2]).      | This is **incorrect**. The core motivation of our Section 4 (Section 5 in the latest PDF version) is to emphasize that mutual distillation between two policies serves as a **regularization** on irrelevant features. In contrast, [1] mainly analyzes generalization in reinforcement learning from a representation learning perspective, and [2] shows that introducing **exploration** before the RL training can improve generalization. We believe this shares **no conceptual similarity** with our motivation in Section 5, i.e., distillation as regularization.     |
| `zv9o`      | The core theoretical claim is formulated too strongly. Corollary 3.8 only shows an increase in the lower bound on generalization performance as robustness increases. However, increasing a lower bound does not guarantee an increase in actualized performance, which is what is promised in the abstract and introduction.      | This is **incorrect**. Our theoretical results clearly establish a direct connection between generalization and policy robustness. The reviewer claims that the theoretical results do not guarantee improvements in actual generalization performance. However, **no theory** can provide such a guarantee. In fact, although TRPO [3] proves monotonic improvement of the policy, we **cannot** ensure that the TRPO algorithm indeed achieves **actual** performance monotonic improvement. The reviewer's claim lacks objectivity. In addition, **Corollary 4.8** explicitly shows that the lower bound of generalization performance **can be improved** through policy robustness. Consequently, we believe this at least indicates that if the policy is sufficiently robust, generalization performance will improve, which already constitutes a meaningful theoretical result. |
| `zv9o`     | Clarity of the theoretical framework is lacking. For example, the proofs depend on two policies and $\pi$ and $\tilde{\pi}$, but there is no mention of what the significance of these two policies are or how they are related to the overall narrative of the paper.      | This is **incorrect**. This means that the reviewer is **not familiar** with the basic theoretical results of modern policy gradient algorithms such as TRPO [3] and CPO [4]. In this context, $\pi$ and $\tilde{\pi}$ typically represent the policies before and after an update, and they can be **any** two policies.      |
| `zv9o`      | Does it matter how $\pi$ and $\tilde{\pi}$ are related? Can we view them through the lens of the two policies used in mutual distillation somehow?      | This is a **fundamental misunderstanding** of our method. It indicates that the reviewer is **not only** unfamiliar with the commonly used theoretical results in policy optimization [3, 4] **but also** has not understood the motivation of our method. $\pi$ and $\tilde{\pi}$ represent the policy of a single agent before and after each training iteration, while the two agents regularize each other and improve robustness through mutual distillation. $\pi$ and $\tilde{\pi}$ **do not** represent the two agents in our MDPO algorithm.     |
| `zv9o`     | Line 200: "During the training process, we can only empirically bound $\mathfrak{D}\_{\mathrm{train}}$, why?     | Here, $\mathfrak{D}\_{\mathrm{train}}$ refers to the **commonly known** trust region, which in our algorithm is implemented via the **ratio clipping** in PPO [5]. The reviewer is **not even familiar** with the concept of a trust region.|
| `zv9o`      | The claims regarding theoretical novelty are strong, but difficult for me to verify due to insufficient discussion of existing work, and my own **lack of knowledge** on this specific topic.      | We have added the related work section in the discussion phase. However, based on the previous observations, the reviewer is **not familiar** with the theoretical results of policy optimization methods such as TRPO [3] and CPO [4], and also demonstrates **a lack of knowledge** in this specific area (generalization in reinforcement learning). Therefore, we **question** whether the reviewer's **score of 2** truly holds objective significance as a reference.      |

---

> ### Author Response · Authors · 2025-11-28
>
> | Reviewer | Comment | Our Explanation |
> |----------|----------|----------|
> | `kbQ7`      | I found that the novelty of the theoretical exposition is overstated in serveral places. One of the main theorems (Theorem 3.3) mirrors the first-order performance difference bound from TRPO [1] so closely that I believe it should be cited as a known existing bound.      | This is **incorrect**. First, we **did** cite TRPO [3] when introducing the first-order approximation term $L_\pi$ before Theorem 3.3. Second, although Theorem 3.3 is indeed related to TRPO, the lower bound in TRPO involves the **maximum** KL divergence over all states $s$, whereas our bound relies on the TV divergence **expectation** over states $s\sim d^{\mu_f}$, and we additionally introduce a rendering function $f\in\mathcal{F}$. Therefore, directly citing it as the original TRPO bound is not appropriate. Finally, Theorems 3.4, 3.5, 3.7, as well as Definition 3.6 and Corollary 3.8, constitute the main theoretical contributions of our paper, especially Corollary 3.8.  |
> | `kbQ7`      | Knowing that feature learning is difficult to treat theoretically in deep learning I would **suggest an alternative path** would be to support the argument **more empirically**.      | We have adopted the reviewer's suggestion and thus added **a large number** of empirical results to demonstrate that mutual distillation **indeed** helps the policy learn more robust representations. These results provide **strong evidence** supporting the claims of our paper, please refer to our responses for details. |
> | `kbQ7` | The paper furthermore claims to be the first to “ first to provide a rigorous proof of {the} intuition {that robustness to irrelevant features enhances generalization performance}", which is an overstatement in this phrasing in my view. | To avoid theoretical overstatement, we have **revised the wording** in the abstract to indicate that we are the first to provide an **end-to-end** generalization theory showing that improving policy robustness to irrelevant features helps generalization. Nevertheless, we **did not** find any theoretical results in the works cited by the reviewer that build an **explicit connection** between generalization performance and policy robustness. In addition, we have further supplemented the related work section and cited the studies listed by the reviewer. |
>
> In summary, we have reason to **doubt** the expertise and objectivity of Reviewer `zv9o`, since Reviewer `zv9o` gave our work the **lowest score of 2**. In addition, we have clarified some of Reviewer `kbQ7`'s comments and, as requested, supplemented **a large amount of empirical evidence** in our responses, effectively supporting the claims in our paper.
>
> For more details, please refer to our specific responses to the reviewers. Given that generalization in reinforcement learning is indeed an **important and highly discussed** topic, we believe our insights can bring value to the community and both drive and inspire further research.
>
> Thank you for your time and consideration.
>
> Best,
>
> authors
>
> ---
> **References**
>
> [1] M Suau et al. Bad habits: Policy confounding and out-of-trajectory generalization in RL.
>
> [2] M Weltevrede et al. Exploration Implies Data Augmentation: Reachability and Generalisation in Contextual MDPs.
>
> [3] J Schulman et al. Trust region policy optimization.
>
> [4] J Achiam et al. Constrained policy optimization.
>
> [5] J Schulman et al. Proximal policy optimization algorithms.

---

### Author Response · Authors · 2025-11-29
**To the new AC (Part II)**

During the discussion phase, reviewer `zv9o` responded to our rebuttal. However, we found that `zv9o`'s additional comments introduced **additional** factual errors, which we further summarize as follows:

| Reviewer | Comment | Our Explanation |
|----------|----------|----------|
| `zv9o`      |   I do not yet see how the listed papers share no conceptual similarity to your motivation in **Section 5 (revised PDF)**. From my perspective, they all argue that collecting more diverse (i.e., more than just the on-policy data from a single policy) training data from the training environments should increase generalisation, which seems to be the main motivation for MDPO?    |   This is **incorrect**. The reviewer **keeps insisting** that the core motivation of our paper is that more diverse data leads to better generalization. However, in the Distillation as Regularization section (**Section 5** of the revised PDF), we have clearly demonstrated that the core motivation is that mutual distillation between two online RL policies helps **prevent them from overfitting to irrelevant features** (i.e., improves policy robustness). In fact, if data diversity is the key factor, then the scaled PPO in **Figure 5** should exhibit the same generalization performance as MDPO, since it using the same batch size and the total interaction steps. Yet MDPO still achieves better generalization performance, which strongly demonstrates that mutual distillation **is not simply** sample aggregation/diversity.   |
| `zv9o`      |   My point is not about the practical versus theoretical results. Instead, I'm referring to the fact that the theoretical bound could be completely vacuous. For example, if the rewards are strictly non-zero, then the test performance will also always be non-zero. Now, if the right-hand side of the lower bound from Corollary 3.8 is a large negative number, than this would be an example of a vacuous bound. A more robust policy might increase this lower bound, from -100 to -90 for example, but this would still have no relevance to the actual test performance (which is always > 0).   |    The reviewer is trying so hard to **construct counterexamples** to our theoretical results. However, as long as we can make the policy sufficiently robust, the lower bound on generalization performance **will improve**. Corollary 4.8 explicitly shows that the lower bound of generalization performance can be improved through policy robustness, whether it is the reviewer's claimed range of **–100 to –90** or **–5 to 5**. Consequently, we believe this at least indicates that if the policy is sufficiently robust, generalization performance will improve, which already constitutes a **meaningful** theoretical result. In addition, after Definition 4.6, we emphasized that our intention in this definition **is not** to derive the tightest possible bound but rather to demonstrate how policy robustness to irrelevant features can contribute to improved generalization. Moreover, our theoretical results **can be easily improved** by modifying the definition of $\mathcal{R}$-robust to make it state-dependent.  |
| `zv9o`      |   This also highlights an additional concern on the novelty of the proof, as I believe [4] prove something very related. They first prove that a policy trained with the correct symmetric subgroup, can become robust to the full group. They then prove that a robust policy has bounded generalisation gap (which decreases as the policy becomes more robust). This second step is in principle independent from how the policy has become robust or what it is robust to (as long as the what is relevant for the testing distribution). As such, **they don't explicitly define robustness and then explicitly prove generalisation performance increases for more robust policies**, but this is an important substep of their overall proof. This might also be true for other proofs like this in the literature.    |   The reviewer insists that reference [4] is relevant. We acknowledge this and have **added a citation** to [4]. However, the experiments in [4] are **limited to toy environments** with symmetric group structures, such as Reacher with rotational symmetry, whereas our environments are the **much more challenging** hard-level Procgen benchmarks. Moreover, our theoretical framework **does not** require assuming that the rendering function has any symmetric group structure (we **do not even** explicitly define the the rendering function distribution $p:\mathcal{F}\mapsto[0,1]$) and **explicitly establishes a connection between generalization performance and policy robustness**. Therefore, we believe that our theoretical framework is more general, and our experimental setup is more challenging. Notably, the reviewer also acknowledge that **[4] does not explicitly define robustness nor explicitly prove that generalization performance increases for more robust policies**.   |

---

> ### Author Response · Authors · 2025-11-29
>
> | Reviewer | Comment | Our Explanation |
> |----------|----------|----------|
> | `zv9o`      |  Isn't it somewhat trivial that if a policy is fully robust $\mathcal{R}=0$ to any rendering function $f$, it will achieve exactly the same performance regardless of rendering function $f$ (so it would have 0 generalisation gap)? |  This is **incorrect**. We would like to point out the **logical error** in the reviewer's argument. The reviewer attempts to use this trivial example to suggest that our theoretical results are also trivial, but this trivial case **does not** imply that our theoretical results are trivial. The insights provided by our theory go far beyond this, as we explicitly establish a lower bound on generalization performance, which explicitly incorporates policy robustness, as we have consistently emphasized above. |
> | `zv9o`      |  Furthermore, could we not (almost trivially) show that robustness decreases the generalisation gap by using something like Theorem 2 from [3] and using the definition of the rendering function $f$, policy $\mu_f$ and robustness $\mathcal{R}$? It could very well be that this is not true, but in that case, I believe the theorem could benefit from a justification on why its not so trivial to prove this. |  This is **incorrect**. The reviewer insists that our theoretical results are trivial. However, the reviewer is indeed not very familiar with the field of policy optimization (see the evidence we provided in **Part I**). The reviewer cites Theorem 2 from [3] in **2023** here, but this theoretical result **had already been proved** in Theorem 1 of CPO [5] in **2017**, providing a very general and formally consistent result. This result only focuses on the lower bound of the performance difference between two policies under an **normal** MDP and **cannot distinguish** between the training performance and generalization performance as in our paper, which is why we introduced the rendering function $f\in\mathcal{F}$. The reviewer insists that our theoretical results are trivial and even **asks us to prove why it is not so trivial to prove this**. We believe this request is beyond the scope of the discussion phase. |
>
> In addition, we note that some of the references cited by reviewer `zv9o` overlap with those cited by reviewer `kbQ7`, even though we have repeatedly clarified that [4] and [6] are **not directly related** to our core contribution, i.e., **Distillation as Regularization**. We even implemented the algorithm from [6], PPO+Explore-Go, under our setting (i.e., the hard-level), with the results shown in the table below:
>
> | Algorithm | bigfish     | dodgeball   |
> |-------------|-------------|-------------|
> | PPO                | 0.26 ± 0.23 | 0.92 ± 0.46 |
> | PPO+Explore-Go [6]         | 0.33 ± 0.23 | 0.96 ± 0.45 |
> | MDPO         | **16.11 ± 4.63**| **5.66 ± 1.98** |
>
> It can be observed that MDPO still achieves significantly better generalization performance. While we have indeed **added citations to [4] and [6]**, we also note that `zv9o` and `kbQ7` gave our work the **lowest scores of 2 and 4**, respectively. We will not make any unfounded speculations, but we hope this fact draws your attention.
>
> Should this paper not be accepted, it may be a matter of differing research taste. We believe the key insight presented is highly aligned with the focus of the International Conference on **Learning Representations** (IC**LR**).
>
> Thank you again for your time and consideration.
>
> Best,
>
> authors
>
> ---
> **References**
>
> [1] A Zhang et al. Learning invariant representations for reinforcement learning without reconstruction.
>
> [2] M Suau et al. Bad habits: Policy confounding and out-of-trajectory generalization in RL.
>
> [3] Maran et al. Tight Performance Guarantees of Imitator Policies with Continuous Actions.
>
> [4] M Weltevrede et al. How Ensembles of Distilled Policies Improve Generalisation in Reinforcement Learning.
>
> [5] J Achiam et al. Constrained policy optimization.
>
> [6] M Weltevrede et al. Exploration Implies Data Augmentation: Reachability and Generalisation in Contextual MDPs.

---

### Note · Authors · 2026-01-26

I have read and agree with the venue's withdrawal policy on behalf of myself and my co-authors.

---

### Meta-Review · Area_Chair_xnxc · 2026-01-10

**Summary:**

The paper studies generalization in reinforcement learning by showing, both theoretically and empirically, that robustness to irrelevant (non-generalizable) features improves test performance. It proposes mutual distillation between policies as an implicit regularizer that encourages invariant representations, yielding improved generalization on Procgen benchmarks without aiming for state-of-the-art results.

This paper led to extended discussion between the reviewers and the authors. Three main concerns were raised by multiple reviewers: (1) presentation and clarity, (2) positioning with respect to related work, including missing citations, and (3) a perceived gap between the empirical results and the paper’s claims. Importantly, these issues were noted by more than one reviewer, rather than reflecting an isolated opinion of a reviewer that could be potentially wrong. In contrast, the reviews recommending acceptance were relatively muted. While the authors raised concerns about the reviews and provided a revised manuscript, the shortened discussion period did not allow reviewers sufficient time to fully assess whether the issues were adequately addressed. Overall, it seems to me that, despite the concerns raised by the authors about the reviews, the paper would still benefit from a clearer articulation of how its contributions differ from existing work and why those differences matter. I therefore recommend rejection.

**Reviewer Concerns:**

- Paper should better position itself in the literature
- Lack of interpretability

	"The paper has several issues with presentation and clariity"

- Evaluation limited to Procgen, and limited information on the experimental design.
- Results do not match what is being claimed in the paper

	"I found that the novelty of the theoretical exposition is overstated in serveral places."

	"The biggest weakness of this paper in my view is the strong disparity between the derived theoretical claims and the conjectured effect of mutual distillation."

**Reviewer Scores:**

- Reviewer 7ZFX: Initial rating of 6, would likely have kept it. The review is not necessarily that informative, though.
- Reviewer xv4v: Initial rating of 6, would likely have kept the score.
- Reviewer zv9o: Initial rating of 2. The review seems quite thorough and they were negative enough that it would surprise me if they were to increase their score.
- Reviewer kbQ7: Initial rating of 4. It seems unlikely they would have raised their score.

---

### Decision · Program_Chairs · 2026-01-26

Reject